**Technical Report**

# Accurate and sensitive mutational signature analysis with MuSiCal

Hu Jin [1,2], Doga C. Gulhan [1,2], Benedikt Geiger [1], Daniel Ben-Isvy [1], David Geng[1], Viktor Ljungström[1] & Peter J. Park [1] ✉

Mutational signature analysis is a recent computational approach for interpreting somatic mutations in the genome. Its application to cancer data has enhanced our understanding of mutational forces driving tumorigenesis and demonstrated its potential to inform prognosis and treatment decisions. However, methodological challenges remain for discovering new signatures and assigning proper weights to existing signatures, thereby hindering broader clinical applications. Here we present Mutational Signature Calculator (MuSiCal), a rigorous analytical framework with algorithms that solve major problems in the standard workflow. Our simulation studies demonstrate that MuSiCal outperforms state-of-the-art algorithms for both signature discovery and assignment. By reanalyzing more than 2,700 cancer genomes, we provide an improved catalog of signatures and their assignments, discover nine indel signatures absent in the current catalog, resolve long-standing issues with the ambiguous 'flat' signatures and give insights into signatures with unknown etiologies. We expect MuSiCal and the improved catalog to be a step towards establishing best practices for mutational signature analysis.

Mutational signature analysis has emerged as a powerful tool for unveiling the mutational processes underlying somatic DNA alterations[1–3]. Analysis of a large collection of tumors profiled with whole-genome and whole-exome sequencing has led to the discovery of a number of mutational signatures consisting of single-base substitutions (SBSs), insertion–deletions (indels; IDs) and structural variations (SVs)[4–6]. Some of these signatures have been associated with specific exogenous or endogenous mutational processes[7–9], although the majority still have unknown etiologies. Early evidence shows that application of mutational signature analysis may be informative for prognosis and treatment decisions for cancer patients[10–16].

To maximize the impact of mutational signature analysis, rigorous mathematical understanding and robust algorithms are needed to increase its accuracy and interpretability. The field has seen a rapid development of computational techniques in the past decade. Popular tools such as SigProfilerExtractor[17], SignatureAnalyzer[5,18,19], signature.tools.lib[20,21] and others have achieved considerable success[22]. However, several methodological difficulties remain[3,23]. One such difficulty is in

the signature assignment step, in which the set of active mutational signatures and their corresponding exposures are determined for either a dataset or a single sample. Misleading signature assignments are pervasive in the literature, as we demonstrate using the results from two recent pan-cancer studies[20,24] (Supplementary Fig. 1). Signatures with similar spectra—such as the relatively 'flat' ones—are particularly problematic because they are more difficult to separate from each other mathematically (Supplementary Fig. 2a,b). Inaccurate signature assignment can have meaningful implications in therapeutic decision-making. For example, observation of the homologous recombination deficiency signature SBS3—which suggests poly(ADP-ribose) polymerase inhibitors as a treatment option[25,26]—in a multiple myeloma cohort has been found to be a false positive caused by flawed signature assignment[11,23,27].

Another difficulty is in the signature discovery step (often termed de novo signature discovery), in which mutational signatures are directly extracted from a cohort. Numerous tools have been developed, most of which rely on nonnegative matrix factorization (NMF)[28].

[1]Department of Biomedical Informatics, Harvard Medical School, Boston, MA, USA. [2]These authors contributed equally: Hu Jin, Doga C. Gulhan. ✉e-mail: peter_park@hms.harvard.edu

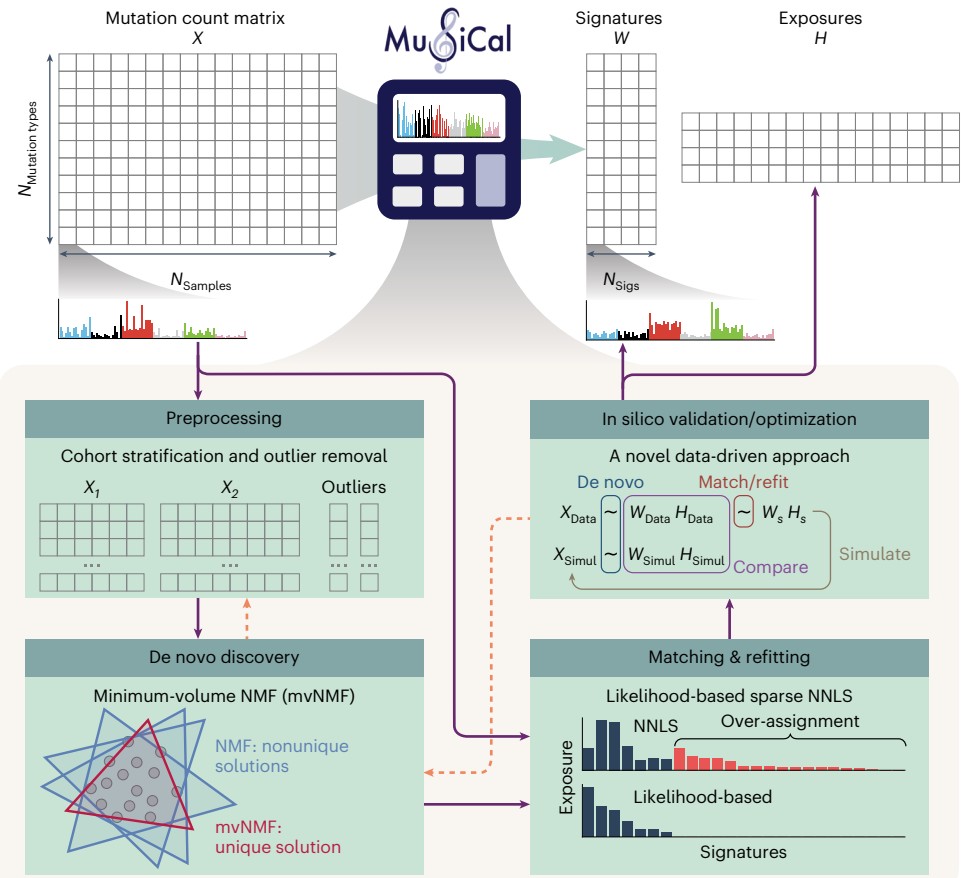

**Fig. 1 | Overview of MuSiCal.** MuSiCal decomposes a mutation count matrix $X$ into a signature matrix $W$ and an exposure matrix $H$ through four main modules: preprocessing, de novo discovery, matching and refitting, and in silico validation/optimization. The new methods implemented in each module are highlighted. Solid purple arrows represent a typical workflow of mutational signature analysis. Dashed orange arrows denote that some steps in the workflow potentially require repeating an earlier step ('Results').

However, they lead to highly variable results, even from the same dataset[5,22]. It is therefore difficult to compare signatures discovered across studies, especially to determine whether a signature is new or merely a variation of a known one due to algorithmic biases. This issue may underlie the large number of signatures that are overly similar or can be expressed as linear combinations of each other in the widely used catalog of known signatures from the Catalogue of Somatic Mutations in Cancer (COSMIC)[29] (Supplementary Fig. 2c). It may also confound the tissue specificity observed for some signatures[20].

Here, we present MuSiCal, a comprehensive framework that enables accurate signature assignment as well as robust and sensitive signature discovery (Fig. 1). To solve the problems described above, MuSiCal leverages several new methods, including minimum-volume NMF (mvNMF)[30–33], likelihood-based sparse nonnegative least squares (NNLS) and a data-driven approach for systematic parameter optimization and in silico validation. In addition, to facilitate future studies of mutational signatures in cancer, we present an improved catalog for both signatures and their assignments by reanalyzing more than 2,700 genomes from the Pan-Cancer Analysis of Whole Genomes (PCAWG) project[34]. Our new catalog of ID signatures is more complete, containing nine new ID signatures absent in the current COSMIC catalog. The derived signature assignments also resolve issues with ambiguous flat signatures, providing insights into signatures with unknown etiologies.

## Results

### De novo mutational signature discovery with mvNMF

Although NMF is commonly used, it has an intrinsic problem of producing nonunique solutions[35–37]. This can be illustrated by a geometric interpretation of NMF (Fig. 2a): any convex cone that encloses the data points is a valid NMF solution, and there exist infinitely many such convex cones for a fixed dataset in general. Indeed, given the same input data, multiple NMF runs converge to different solutions, none of which recover the true signatures accurately (Fig. 2a). Relatively flat signatures (for example, SigC in Fig. 2a) have especially large errors because of the lack of constraints. In Fig. 2b, we simulate profiles with real signatures SBS3, 8, 12 and 23, and try to recover them using NMF. Whereas SBS8, 12 and 23 are recovered accurately, the flatter SBS3 receives notable distortions. Specifically, the NMF solution for SBS3 has reduced weights in trinucleotide categories enriched in the other three signatures. As a result, it may be misidentified as a new signature or, when decomposed using COSMIC signatures by NNLS, false signatures (for example, SBS39) are introduced (Fig. 2b). Therefore, the nonuniqueness of NMF solutions can lead to algorithmic distortions that are difficult to distinguish from real biological signal, posing substantial challenges to accurate signature discovery and assignment.

We propose a new approach based on mvNMF, originally developed for audio-source separation and hyperspectral-image unmixing[30–33]. Compared with NMF, mvNMF adds a regularization term that penalizes the volume of the convex cone formed by discovered signatures (the area of the purple shaded triangle in Fig. 2a), promoting the unique solution with minimum volume among those that are equivalent based on NMF (Methods). Theoretically, mvNMF is guaranteed to recover the true underlying signatures under mild assumptions satisfied by tumor somatic mutations, whereas NMF requires much stronger assumptions that are often unrealistic[33,35,37] (Supplementary

Notes and Supplementary Figs. 3 and 4). In the examples above, mvNMF accurately recovers the true signatures without distortions (Fig. 2a,b).

For a more systematic comparison, we ran NMF and mvNMF on synthetic datasets generated using tumor type-specific SBS signatures and Dirichlet-distributed exposures (Fig. 2c and Methods). Consistent with the examples in Fig. 2a,b, mvNMF improves the accuracy of discovered signatures compared with NMF, reducing the cosine error by 67–98% when averaged across different signatures within the same tumor type (Extended Data Fig. 1). The improvement is especially notable for relatively flat signatures. Even sparse signatures with nonzero weights in only a few SBS categories can pose substantial challenges for NMF because of the interference received from other similar signatures, whereas mvNMF does not suffer from these distortions (see, for example, SBS7a in Extended Data Fig. 1d).

### Benchmarking MuSiCal for de novo signature discovery

We benchmarked the performance of MuSiCal for de novo signature discovery against the state-of-the-art algorithm SigProfilerExtractor[17]. To recapitulate the complexities of real-life data, synthetic datasets are simulated from tumor type-specific SBS signatures and real exposure matrices produced by PCAWG[5] (Methods). To infer the number of signatures from the dataset itself, MuSiCal employs a new parameter-free method (Methods and Extended Data Fig. 2).

To evaluate the quality of de novo signature discovery, we use the accuracy of signature assignment from the downstream matching step. We illustrate this metric with an example simulated from the 13 SBS signatures and real exposures produced by PCAWG for Skin-Melanoma (Fig. 3a and see Supplementary Data for examples based on other tumor types). SigProfilerExtractor discovers only three de novo signatures from this dataset, whereas MuSiCal discovers nine (Fig. 3a). Most de novo signatures are mixtures of a subset of the 13 true signatures, potentially because of their strong co-occurring patterns, making direct comparison between the de novo and the true signatures difficult. We therefore perform a downstream matching step in which each de novo signature is decomposed into a nonnegative mixture of potentially multiple COSMIC signatures. Precision, recall and their harmonic mean (the F1 score) can subsequently be calculated. Specifically, the likelihood-based sparse NNLS is used for this matching step at different likelihood thresholds (see 'Refitting with likelihood-based sparse NNLS' for more details), allowing the entire precision–recall curve (PRC) and the corresponding area under PRC (auPRC) to be obtained. For the Skin-Melanoma example, MuSiCal achieves an auPRC of 0.92, outperforming SigProfilerExtractor (auPRC = 0.84).

When we apply the same metric to all 25 tumor types (each with 10 independent simulations), MuSiCal achieves higher mean auPRC in 18 tumor types (9 are statistically significant) compared with SigProfilerExtractor (Fig. 3b). In the remaining seven tumor types, MuSiCal has similar mean auPRC (two equal, five lower and none statistically significant). In Fig. 3c, we show the PRC averaged across all tumor types,

and MuSiCal outperforms SigProfilerExtractor overall (auPRC of 0.929 versus 0.893). Specifically, MuSiCal achieves higher precision at the same recall (Fig. 3d) and higher recall at the same precision (Fig. 3e). MuSiCal outperforms SigProfilerExtractor even when SigProfilerExtractor's built-in input normalization is turned on (Supplementary Fig. 5). The performance of SigProfilerExtractor is improved—although still worse than MuSiCal—when it is forced to select the same number of signatures as MuSiCal (Supplementary Fig. 5). This observation suggests that the improved performance of MuSiCal relies on both reduced algorithmic bias powered by mvNMF as indicated by reduced cosine reconstruction errors of the de novo signatures when decomposed using the true signatures (Fig. 3f), and a better choice of the number of signatures, which in turn benefits from the uniqueness and stability of mvNMF solutions. Further, MuSiCal's improved performance is robust to random noise at different levels (Methods and Extended Data Fig. 3) and unknown spurious signatures present in the data (Methods and Supplementary Fig. 6). MuSiCal also significantly outperforms three additional tools (signature.tools.lib[20,21], SignatureAnalyzer[5,18,19,38] and SigneR[39]) representing different underlying approaches (Methods and Extended Data Fig. 4). Together, these results demonstrate that MuSiCal outperforms the state-of-the-art algorithms for de novo signature discovery in simulations emulating real-life datasets in diverse scenarios.

### Refitting with likelihood-based sparse NNLS

Another key improvement in MuSiCal is in the refitting step, in which an observed mutational spectrum is decomposed into a combination of existing signatures with nonnegative coefficients. Without additional constraints, NNLS attempts to utilize all signatures in the catalog to minimize the fitting error, which often results in over-assignment (that is, a signature that should not be present gets a weight) as illustrated with a simulated example in Fig. 4a. A common method to introduce sparsity in refitting is thresholded NNLS, where signatures that contribute less than a specified percentage to the total mutation burden are set to have zero exposures[40]. The problem with a fixed threshold is that different mutational signatures require different numbers of mutations to provide confidence in their presence. Sparse signatures with prominent weights in only a few mutation channels may be statistically significant even with a small number of mutations, whereas flatter signatures may not be significant even with many mutations. Indeed, in the simulated example above, there is no fixed threshold that recovers the true signatures without adding false positives (Fig. 4a).

We propose a sparse NNLS algorithm based on multinomial likelihoods (Methods and Extended Data Fig. 5a). Briefly, we first perform NNLS to obtain an initial exposure estimate and then refine the exposure vector in a bidirectional stepwise manner. During each step, a signature is considered for addition or subtraction based on its impact on the overall multinomial likelihood of the observed mutation count vector. The use of multinomial likelihoods provides a rigorous way

**Fig. 2 | mvNMF improves the accuracy of de novo signature discovery by inducing unique solutions. a**, An example demonstrating the nonuniqueness of NMF solutions and uniqueness of mvNMF solutions. Synthetic samples are simulated from signatures with three mutation channels (represented by the three axes $x$, $y$ and $z$). NMF and mvNMF are then applied to recover the signatures with three different initializations (shared between NMF and mvNMF). Both signatures and samples are normalized and subsequently visualized on the plane $x + y + z = 1$. **b**, A similar example with real SBS signatures. Example solutions are plotted for NMF and mvNMF. The NMF solution for SBS3 receives a relatively large cosine error, and when decomposed using COSMIC signatures by NNLS (with a relative exposure cutoff of 0.05), it is identified as a composite signature involving the false SBS39. By comparison, mvNMF is able to recover SBS3 accurately. **c**, Cosine errors of NMF and mvNMF solutions in tumor type-specific simulations with real SBS signatures. The number in parentheses after each signature name indicates the number of tumor types where the

corresponding signature is present. The number of data points in each box plot equals this number multiplied by 20 (independent simulations). mvNMF improves the accuracy of de novo signature discovery for most SBS signatures, especially for relatively flat ones characterized by small standard deviations (s.d., shown below the box plot). Although NMF outperforms mvNMF for some sparse signatures (marked by gray signature labels), the cosine errors for both algorithms are much smaller than 0.001 in those cases and thus negligible. Note that the apparent similar or larger spread of cosine errors for mvNMF is because of the log scale on the $y$ axis. Solutions from mvNMF are in fact more stable, producing smaller standard deviations in the cosine errors overall. In all panels, random exposures are generated from symmetric Dirichlet distributions with a concentration parameter of $\alpha = 0.1$, which is a representative value according to real exposure matrices obtained by the PCAWG Consortium[5] (Supplementary Fig. 3). sigs, signatures.

to assess the presence of a signature, regardless of its flatness. In the simulated example above, likelihood-based sparse NNLS is able to identify the set of active signatures correctly without any false positives if the likelihood threshold is within a proper range, alleviating the over-assignment problem (Fig. 4b, see 'Data-driven parameter optimization and in silico validation' for how to choose this threshold). Of note, the widely used cosine similarity is less powerful in separating similar signatures compared with multinomial likelihood (Extended Data Fig. 5b) and fails to discover the set of active signatures correctly at any threshold in the same example (Extended Data Fig. 5c).

When systematically benchmarked on synthetic datasets with tumor type-specific SBS signatures (Methods), likelihood-based sparse NNLS outperforms thresholded NNLS (auPRC of 0.872 versus 0.824) as well as two state-of-the-art algorithms—sigLASSO[41] and the Decomposition module of SigProfilerExtractor[17]—achieving higher precision at the same recall and higher recall at the same precision (Fig. 4c and Supplementary Fig. 7). Finally, MuSiCal uses the same likelihood-based sparse NNLS for the matching step, where de novo signatures are decomposed as nonnegative mixtures of known signatures in the catalog (Methods and Extended Data Fig. 5d).

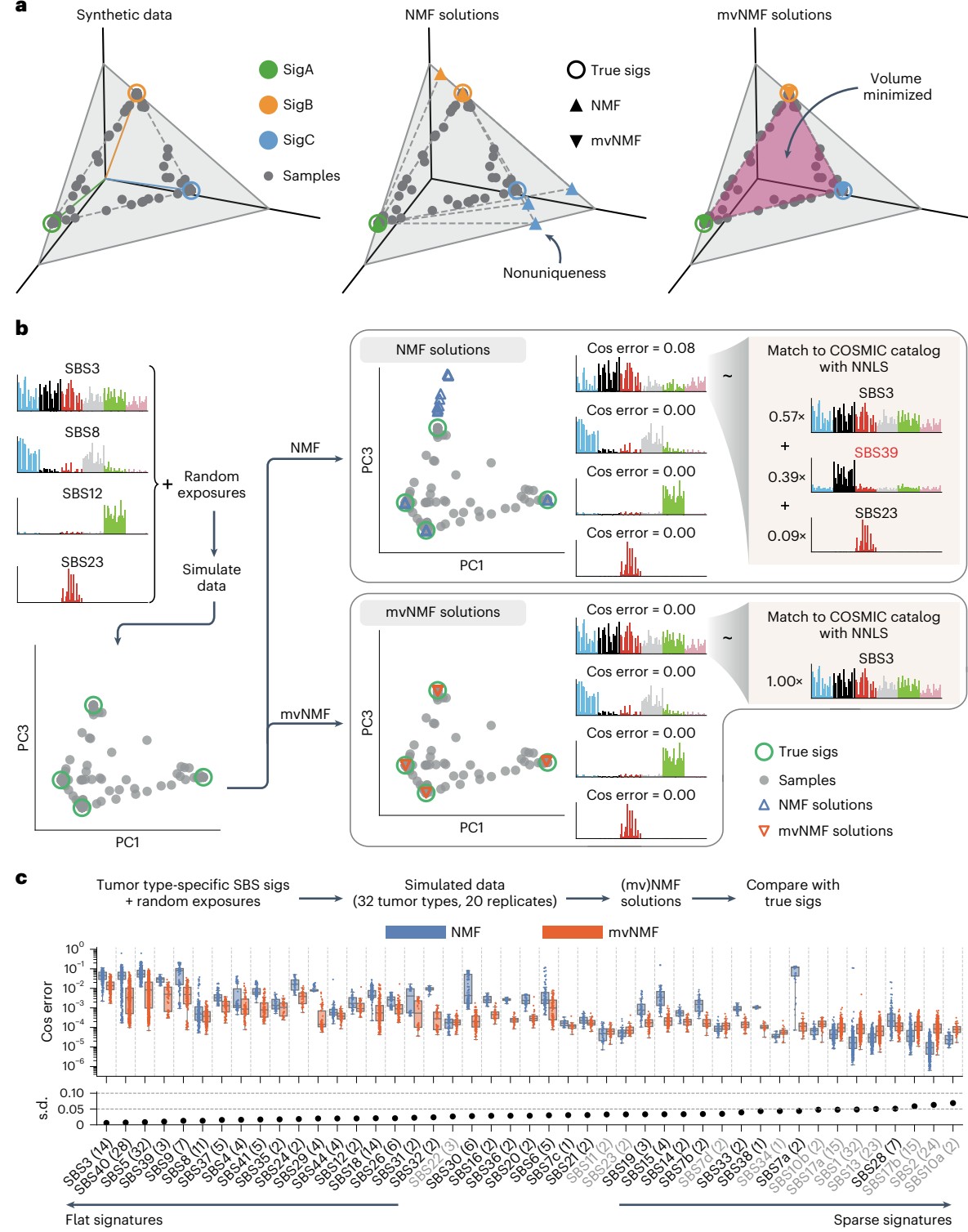

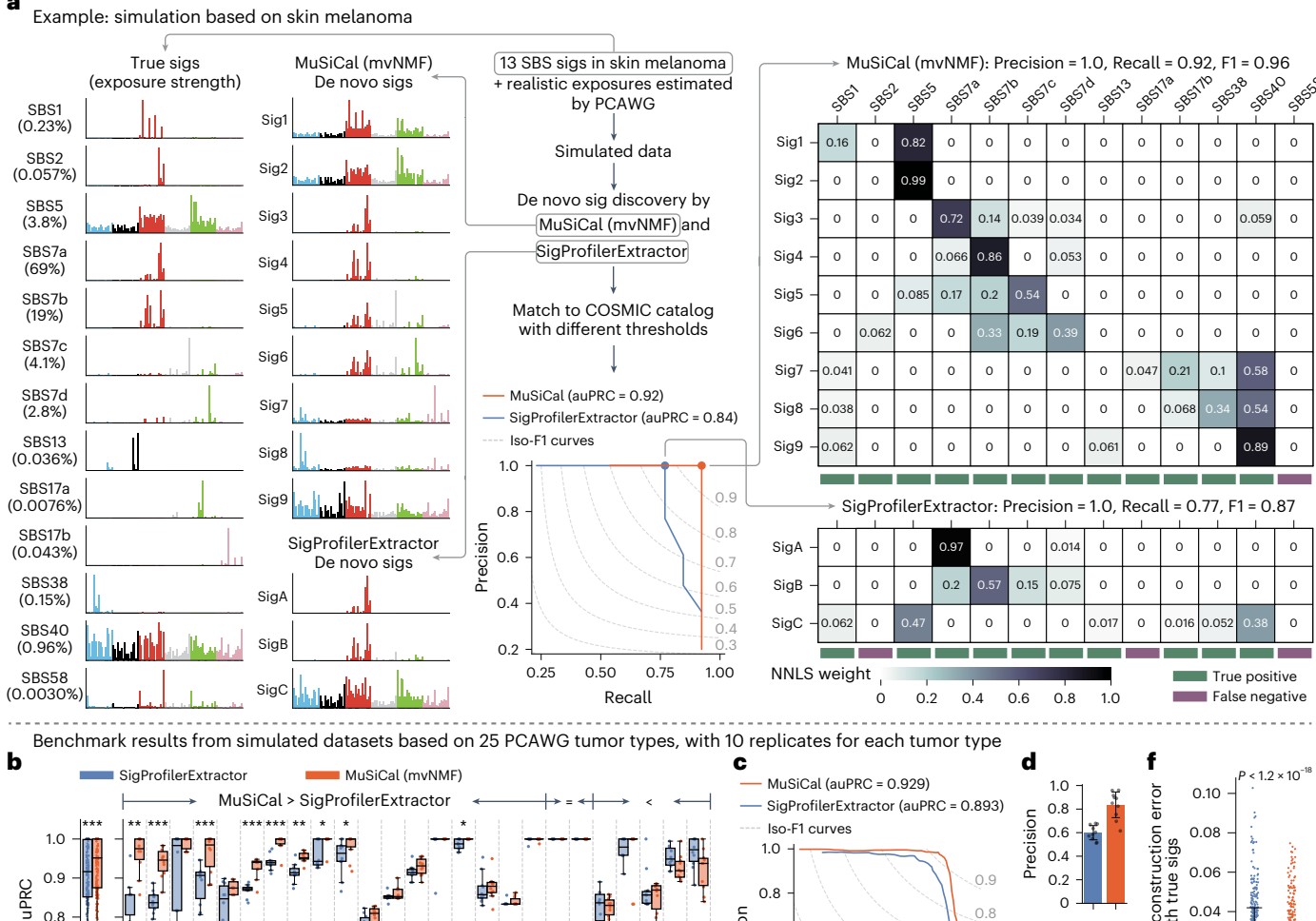

**Fig. 3 | MuSiCal outperforms the state-of-the-art algorithm SigProfilerExtractor for de novo signature discovery. a**, An example based on Skin-Melanoma demonstrating the metric for evaluating the quality of de novo signature discovery. A synthetic dataset is simulated from 13 SBS signatures specific to Skin-Melanoma (percentages below signature names denote exposure strengths) and the exposure matrix produced by the PCAWG Consortium. MuSiCal and SigProfilerExtractor are applied to derive de novo signatures, which are subsequently decomposed as nonnegative mixtures of COSMIC signatures with likelihood-based sparse NNLS at different likelihood thresholds (see 'Refitting with likelihood-based sparse NNLS' for more details). Precision and recall are then calculated at each threshold by comparing matched COSMIC signatures with the 13 true signatures, and auPRC is obtained. The matching result corresponding to the largest achieved F1 score is shown in the heatmap on the right. **b**, Comparison of MuSiCal and SigProfilerExtractor for de novo signature discovery from synthetic datasets based on 25 PCAWG tumor types with at least 20 samples. Ten independent simulations are performed for each

tumor type. The box plot shows auPRC for each individual dataset. Tumor types are sorted according to the mean auPRC gain by MuSiCal. *$P < 0.05$, **$P < 0.005$, ***$P < 0.0005$. $P$ values are calculated with two-sided paired $t$-tests. Raw $P$ values from left to right (significant ones only): $7.5 \times 10^{-10}$, 0.0034, $8.4 \times 10^{-5}$, $1.0 \times 10^{-4}$, $8.3 \times 10^{-5}$, $5.6 \times 10^{-5}$, 0.0012, 0.044, 0.032, 0.012. **c**, PRC of MuSiCal and SigProfilerExtractor averaged across all tumor types. **d**, Precision of MuSiCal and SigProfilerExtractor averaged across all tumor types. Recall is fixed at 0.9. Error bars indicate the standard deviation from ten independent simulations. **e**, Recall of MuSiCal and SigProfilerExtractor averaged across all tumor types. Precision is fixed at 0.98, corresponding to a false discovery rate of 2%. Error bars indicate the standard deviation from ten independent simulations. **f**, Cosine reconstruction errors for MuSiCal- and SigProfilerExtractor-derived de novo signatures ($n = 1,798$ and 1,564, respectively). Each de novo signature is decomposed into a nonnegative mixture of the true underlying signatures with NNLS. Cosine distance is then calculated between the reconstructed and the original de novo signature. The $P$ value is calculated with a two-sided $t$-test.

## Data-driven parameter optimization and in silico validation

To improve the interpretability of final results, de novo signature discovery is usually followed by post-processing steps, including matching and refitting. Currently, the post-processing steps are highly variable across different tools, leading to vastly different

final signature assignments even from the same de novo discovery results. In particular, methods are lacking for systematically selecting parameters involved in post-processing and quantifying the consistency between final signature assignment and data. To tackle this, we propose a new data-driven approach for in silico validation (Methods

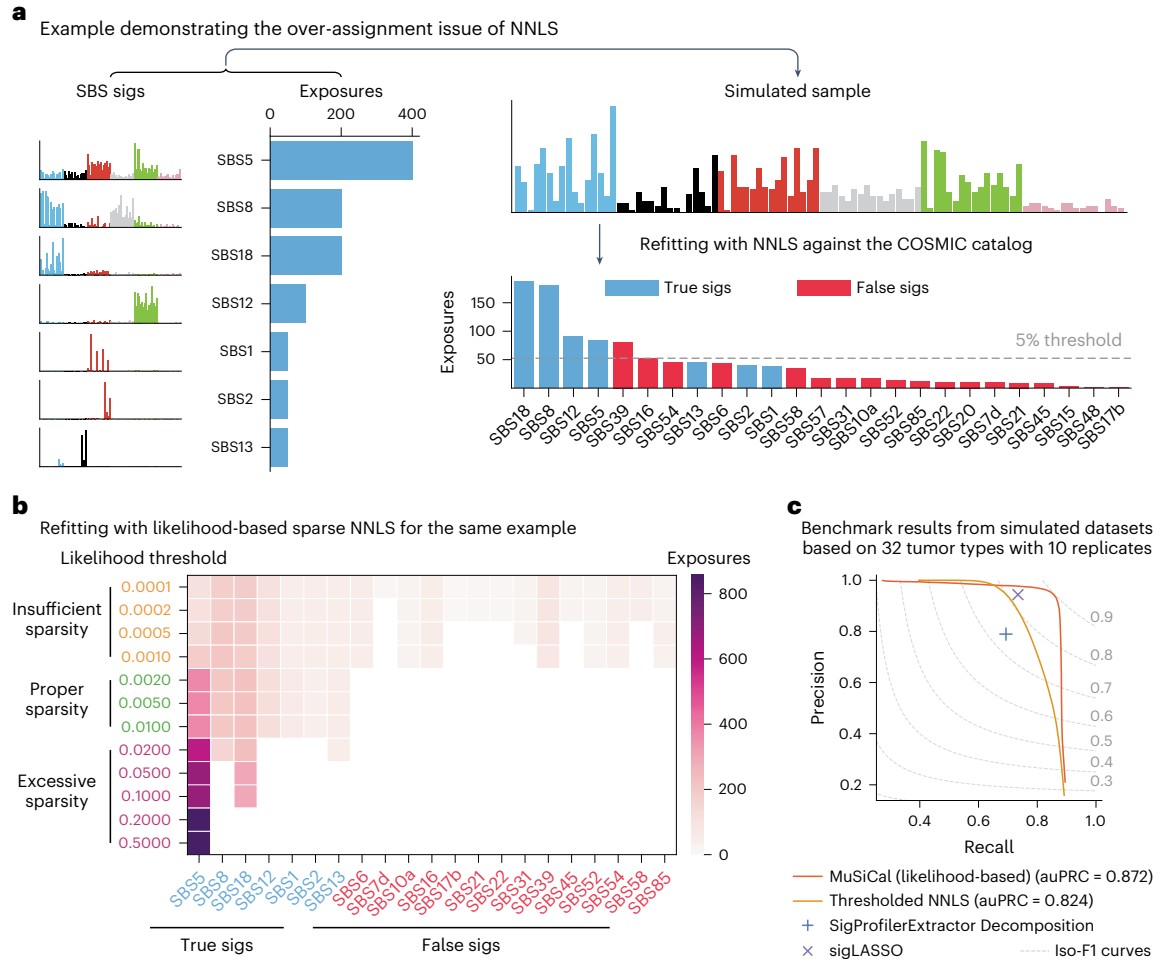

**Fig. 4 | MuSiCal solves the over-assignment problem of refitting using multinomial likelihood-based sparse NNLS. a**, An example demonstrating the over-assignment problem of NNLS. A synthetic sample is simulated from seven SBS signatures and a fixed exposure vector, and then fed into NNLS for refitting against the COSMIC catalog. As a result, 18 additional false-positive signatures receive nonzero exposures. Even with a sparsity-inducing threshold, NNLS fails to discover the set of active signatures correctly. **b**, Solutions of likelihood-based sparse NNLS for the same example with different likelihood thresholds. With a properly chosen threshold, likelihood-based sparse NNLS identifies the set of active signatures correctly. The solutions also show a desired property of continuity with varying thresholds: when the threshold is too small, all seven true signatures are discovered, as well as additional false positives; when the threshold is too large, only the strongest true signatures are discovered. **c**, Performance of likelihood-based sparse NNLS and three other algorithms for refitting on simulated datasets (Methods). For MuSiCal and thresholded NNLS, the entire PRC is shown by scanning the corresponding threshold. For SigProfilerExtractor Decomposition and sigLASSO, a PRC cannot be generated because multiple parameters are involved in each algorithm, and thus the precision and recall achieved with default parameters are shown.

and Fig. 5a). We denote the de novo discovery results as ($W_{data}$, $H_{data}$) and the final signature assignments as ($W_s$, $H_s$) (where s stands for sparse), which are obtained through post-processing steps from ($W_{data}$, $H_{data}$). We first simulate a synthetic dataset $X_{simul}$ from ($W_s$, $H_s$). The simulated data $X_{simul}$ then undergoes de novo signature discovery with exactly the same settings (including the selected number of signatures, $r$) as used for the original data $X_{data}$ to derive ($W_{simul}$, $H_{simul}$). We reason that if ($W_s$, $H_s$) faithfully describes $X_{data}$, the de novo discovery results from simulations and data should be consistent with each other. We thus compare ($W_{simul}$, $H_{simul}$) with ($W_{data}$, $H_{data}$) and use the resulting distances to quantify the consistency between the signature assignments ($W_s$, $H_s$) and the original data $X_{data}$. These distances provide a systematic way to validate the final signature assignments in silico. Compared with the difference between $X_{simul}$ and $X_{data}$, which merely reflects reconstruction errors, the difference between ($W_{simul}$, $H_{simul}$) and ($W_{data}$, $H_{data}$) captures more intricate structures of the dataset as learned through the de novo discovery step. Doing the comparison immediately after de novo discovery also avoids any influence from potential biases in the post-processing steps, allowing us to assess the quality of post-processing and thus optimize the parameters therein (see below).

To illustrate that the validation scheme described above is able to assess the quality of signature assignment and reveal potential issues, we study the PCAWG glioblastoma dataset as an example and compare the final signature assignments from MuSiCal and the PCAWG Consortium[5] (Fig. 5b–f). MuSiCal discovers four de novo signatures in this dataset and, with matching and refitting, assigns eight COSMIC signatures (SBS1, 2, 5, 8, 12, 13, 30 and 31), whereas PCAWG assigns four COSMIC signatures (SBS1, 5, 30 and 40) (Fig. 5b). Following the validation scheme, synthetic datasets are simulated from both the MuSiCal and PCAWG assignments and undergo de novo signature discovery again. We find that the signature assignments from MuSiCal show higher consistency between simulation and data than those from PCAWG (Fig. 5c,d), suggesting improved accuracy of MuSiCal assignments for describing the original data. To investigate potential issues with the PCAWG assignment, de novo signatures from simulations and data are decomposed using the corresponding assigned signatures, and the resultant NNLS weights are compared (Fig. 5e,f). As expected,

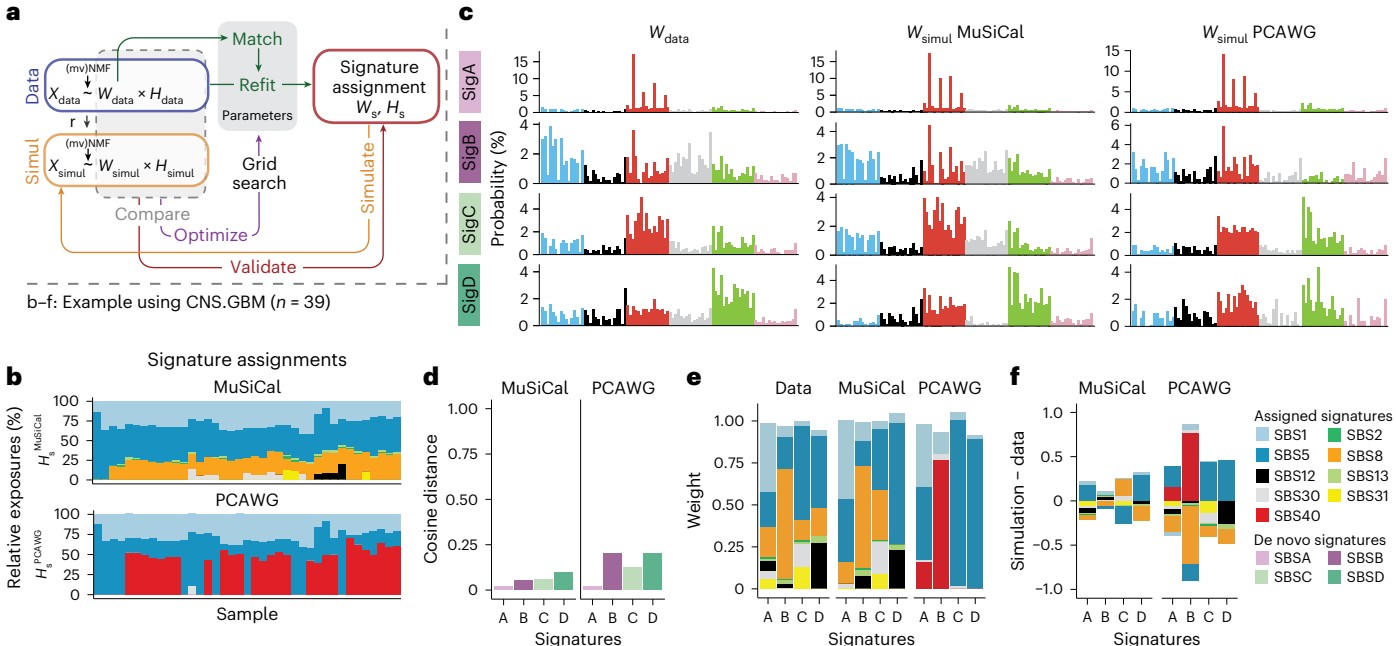

**Fig. 5 | In silico validation quantifies the consistency between signature assignments and data. a**, Diagram of the data-driven approach for parameter optimization and in silico validation of post-processing steps after de novo signature discovery. See the main text for details. **b**, Signature assignments by MuSiCal (upper, $H_s^{MuSiCal}$) and the PCAWG Consortium (lower, $H_s^{PCAWG}$)[5] for the PCAWG glioblastoma dataset, showing discrepancies in both assigned signatures and the corresponding exposures. For simplicity, samples with MMRD or hypermutation because of temozolomide treatment are removed. **c**, Four de novo signatures discovered from the original data (left, $W_{data}$) are compared with those discovered from simulated data based on MuSiCal (middle, $W_{simul}^{MuSiCal}$) and PCAWG (right, $W_{simul}^{PCAWG}$) assignments. **d**, Cosine distances between $W_{data}$ and $W_{simul}^{MuSiCal}$ (left) are compared with those between $W_{data}$ and $W_{simul}^{PCAWG}$ (right), showing that MuSiCal assignments achieve better consistency between

simulation and data in terms of de novo signatures. **e**, NNLS weights of de novo signatures for both simulations and data. De novo signatures from data ($W_{data}$) and MuSiCal-derived simulations ($W_{simul}^{MuSiCal}$) are matched to the MuSiCal-assigned signatures; that is, SBS1, 2, 5, 8, 12, 13, 30 and 31. De novo signatures from PCAWG-derived simulations ($W_{simul}^{PCAWG}$) are matched to the PCAWG-assigned signatures; that is, SBS1, 5, 30 and 40. Standard NNLS is used and the obtained weights are plotted. **f**, Differences between the NNLS weights in **e** for the original data and those for MuSiCal- and PCAWG-derived simulations. Positive and negative differences are plotted separately in opposite directions in a cumulative manner. The results are indicative of over-assignment of SBS40 and under-assignment of SBS8 by PCAWG as well as much improved signature assignment by MuSiCal.

a large discrepancy is observed for the decomposition results between data and PCAWG-derived simulations. In particular, the de novo signatures from PCAWG-derived simulations contain strong components of SBS40, which lacks any evidence from data, but find no matches to SBS8, which has strong evidence from data. These results are indicative of over-assignment of SBS40 and under-assignment of SBS8 by PCAWG for the glioblastoma dataset. In fact, as we demonstrate later, similar misassignment problems, especially of SBS40, are prevalent in the results produced by the PCAWG Consortium[5].

Finally, we use the same data-driven approach to optimize post-processing parameters. In MuSiCal, these parameters are the two likelihood thresholds for likelihood-based sparse NNLS in matching and refitting, respectively. The optimization is achieved through a grid search. Specifically, among the pairs of thresholds that minimize the distance between $W_{data}$ and $W_{simul}$, the pair with the sparsest signature assignment is selected (Methods and Supplementary Fig. 8).

## PCAWG reanalysis with MuSiCal

We reanalyzed more than 2,700 cancer genomes from PCAWG for both SBS and ID signatures to derive a refined catalog of signatures and their assignments (Methods). Tumor type-specific de novo signature discovery is performed after stratifying heterogeneous datasets and removing outliers with the preprocessing module (Methods and Extended Data Fig. 6). Samples with mismatch repair deficiency (MMRD) and/or polymerase epsilon exonuclease (POLE-exo) domain mutations are isolated with a dedicated procedure beforehand and analyzed separately in a tissue-independent manner (Methods).

ID signatures have been under-studied until recently, in part because of the relatively low mutational burden of IDs compared with SBSs[5,7,9,42–44]. Our reanalysis results in a substantial revision and expansion of the COSMIC catalog of ID signatures (Fig. 6a). Among the 18 ID signatures in the current COSMIC catalog, 16 are discovered with modified spectra in our reanalysis (ID1–14, 17 and 18). The other two (ID15 and ID16) are not discovered and likely reflect artifactual IDs upon further inspection (Supplementary Notes and Extended Data Fig. 7a–d). In addition, our reanalysis discovers nine new ID signatures. Specifically, eight of the nine have a cosine similarity <0.8 when decomposed using COSMIC signatures, and we named them ID19–26. The remaining signature is similar to COSMIC ID11 (cosine similarity = 0.94) and we refer to it as ID11b. Compared with COSMIC ID11, which is characterized by 1-bp insertions of C and T at their respective mononucleotide repeats, ID11b is dominated by T insertions and thus included as a new signature. Together, we constructed an updated catalog of ID signatures by replacing COSMIC ID1–14, 17 and 18 with the corresponding signatures from our reanalysis (ID11 was renamed ID11a to distinguish it from ID11b), removing the likely artifactual COSMIC ID15 and 16, and adding the new ID11b and ID19–26 (Fig. 6a and Supplementary Table 1). We note that it is necessary to replace the existing COSMIC spectra with MuSiCal-derived results because several COSMIC ID signatures are resolved into multiple signatures in our updated catalog. For example, COSMIC ID4 is separated into MuSiCal ID4, 19 and 24, and importantly, our updated MuSiCal version of ID4 is more similar to the experimentally derived signature of topoisomerase 1-mediated deletions[42] (Supplementary Notes and Extended Data Fig. 7e,f).

For SBSs, most signatures discovered from our reanalysis are successfully matched to the current COSMIC catalog (SBS1–60 and SBS84–94). The six new SBS signatures with relatively poor matches are mainly (four of the six) discovered in MMRD tumors (potentially benefiting from a separate analysis for them), likely representing non-linear interactions between DNA damage and repair[45] (Extended Data Fig. 8). We thus constructed an updated catalog of SBS signatures by adding the six new ones named SBS95–100, while keeping the COSMIC SBS signatures unchanged (Supplementary Table 2). A global update (as for ID signatures) was not performed because no substantial difference was observed between COSMIC and MuSiCal spectra. One exception is SBS40, a relatively flat signature annotated as having unknown etiology and as being correlated with patient age in some tumor types. The MuSiCal-derived SBS40 has higher proportions of T>N mutations (57%) compared with the COSMIC SBS40 (41%) (Supplementary Fig. 9). Therefore, we replaced COSMIC SBS40 with this modified spectrum in the updated SBS catalog.

With the updated catalogs of ID and SBS signatures, tumor type-specific signature assignments display marked differences compared with results obtained by the PCAWG Consortium[5] (Fig. 6b, Supplementary Fig. 10 and Supplementary Tables 3 and 4).

### SBS–ID associations inform unknown etiologies
Of the 25 ID signatures we identified with MuSiCal, 16 (other than the 9 with annotations from COSMIC) have unknown etiologies. To infer their etiologies, we investigate the association between ID and SBS signatures (Fig. 6c), utilizing the large number of SBS signatures with annotated etiologies (41 from COSMIC). Because many mutagenic processes produce IDs and SBSs simultaneously in experimental studies[7,9], co-occurrence of ID and SBS signatures can indicate a common origin.

Apart from expected correlations between ID and SBS signatures with known and related etiologies (Supplementary Notes), new correlations are observed for both newly added ID signatures in our updated catalog and existing signatures with revised spectra (Fig. 6c). ID11b−unlike COSMIC ID11−correlates with both SBS (SBS4 and SBS92 (ref. 17)) and ID (ID3) signatures known to be associated with tobacco smoking (Fig. 6c and Extended Data Fig. 9a,b), suggesting a potential origin of tobacco smoking-related mutagenesis. ID11b is composed predominantly of 1-bp T insertions following a T, with additional contributions from T insertions following TT or TTT (Fig. 6a). Similar T insertions, as well as the C deletions at CC or CCC observed in ID3, are seen in cells exposed to tobacco carcinogens dibenz[*a*,*h*]anthracene and its diol-epoxide metabolite dibenz[*a*,*h*]anthracene diol-epoxide (Extended Data Fig. 9c)[7]. By comparison, another two components of tobacco smoke−benzo[*a*]pyrene and its diol-epoxide metabolite benzo[*a*]pyrene-7,8-dihydrodiol-9,10-epoxide−result in only ID3 and not ID11b (Extended Data Fig. 9c)[7]. Interestingly, all four chemicals produce SBS signatures similar to SBS4 (Extended Data Fig. 9c)[7]. In smoking-related (SBS4- or SBS92-positive) lung, liver, head-and-neck and bladder cancers, ID11b is indeed required to explain the observed abundance of 1-bp T insertions following a T (Extended Data Fig. 9d). Together, these results suggest that ID11b represents another component of tobacco smoking-related mutagenesis and is likely specific

to DNA damage produced by dibenz[*a*,*h*]anthracene and dibenz[*a*,*h*]anthracene diol-epoxide.

ID14, dominated by 1-bp C insertions at nonrepetitive regions and 1-bp T insertions at long (≥5 bp) mononucleotide T repeats, correlates with SBS10 and 28, indicating that ID14 is associated with POLE-exo mutations (Fig. 6a,c). ID26, characterized by 1-bp T deletions at 4- and 5-bp mononucleotide T repeats and potentially resolved from COSMIC ID7, correlates with SBS14 and 15, suggesting an MMRD-related origin (Fig. 6a,c). Finally, ID25, consisting of mostly 1-bp indels at nonrepetitive regions and T indels at long mononucleotide T repeats, correlates with SBS7 and contributes to relatively large numbers of mutations in skin melanomas, suggesting that DNA damage by ultraviolet (UV) light exposure or related repair processes may underlie ID25 (Fig. 6a–c).

Some of the SBS–ID correlations mentioned above are also observed with exposure matrices produced by the PCAWG Consortium. However, the results from MuSiCal show stronger correlations overall and reveal new ones that are not detected by PCAWG (Supplementary Fig. 11).

### Improved signature assignment resolves ambiguities with flat signatures
To demonstrate the improved signature assignments by MuSiCal (Fig. 6b and Supplementary Fig. 10), we focused on SBS40. SBS40 is relatively flat, similar to SBS3 and SBS5 (cosine similarities of 0.88 and 0.83, respectively), and overlaps with many other signatures. Thus distinguishing SBS40 from other signatures and quantifying its contribution in different samples are mathematically difficult. PCAWG assigns SBS40 to nearly every tumor type (Supplementary Fig. 10)[5]. However, we believe this result is artifactual, caused by the over-assignment problem described earlier. Some studies have noticed this problem and have manually combined SBS40 with other flat signatures[46,47]. Indeed, our in silico validation using data-driven simulations (described in Fig. 5a) confirms this over-assignment issue (Fig. 7a), because de novo signatures extracted from simulated datasets based on PCAWG assignments ($W_{simul}^{PCAWG}$) contain an excess amount of SBS40 (red bars in Fig. 7a) in most tumor types when compared with de novo signatures extracted from the original data ($W_{data}$). By comparison, MuSiCal assigns SBS40 only to kidney and bladder cancers (Fig. 6b), achieving better consistency between simulation ($W_{simul}^{MuSiCal}$) and data ($W_{data}$) in in silico validation (Fig. 7a). The assignments of other SBS and ID signatures are also much improved with MuSiCal, as indicated by the overall reduced bar heights in Fig. 7a and Supplementary Fig. 12, and the smaller cosine distances between $W_{simul}$ and $W_{data}$ in Fig. 7b. Of note, SBS5 appears to be over-assigned by both MuSiCal and PCAWG (blue bars in Fig. 7a), suggesting that the SBS5 spectrum in its current form may be inaccurate, or that it is in fact a composite signature corresponding to multiple mutational processes operative in different tumor types.

Previous studies[5] and our own analysis (Fig. 6c) suggest that SBS40 is correlated with ID5, characterized by 1-bp T deletions at 1−5-bp mononucleotide T repeats (Fig. 6a). If SBS40 is truly a ubiquitous signature, this correlation should be observed in most tumor types. However, a strong correlation between SBS40 and ID5 exists only in kidney and bladder cancers (for both MuSiCal and PCAWG assignments) and not

---

**Fig. 6 | Reanalysis of PCAWG data with MuSiCal uncovers new signatures and SBS–ID associations. a**, The catalog of ID signatures obtained from our reanalysis of PCAWG data with MuSiCal. An asterisk indicates newly added signatures not present in the current COSMIC catalog. Other signatures also have modified spectra. Each of the nine new signatures is decomposed using COSMIC signatures by NNLS. The resulting cosine similarities between the original and the reconstructed signatures are annotated next to each spectrum. **b**, MuSiCal-derived signature assignments in PCAWG samples for SBS (upper) and ID (lower) signatures. Marker size represents the prevalence of a signature; that is, the proportion of samples with nonzero exposures of the corresponding signature

within a tumor type. Color indicates the median number of mutations per Mb contributed from the corresponding signature among samples with nonzero exposures. **c**, Heatmap of Pearson correlation coefficients between the per-sample exposures of SBS and ID signatures. The correlations are calculated for MMRP samples and samples with MMRD and/or POLE-exo mutations separately. Only the samples with at least 100 indels are used and only statistically significant associations are shown (adjusted $P < 0.05$). The correlations are also examined separately in each tumor type where the corresponding ID and SBS signatures are both active, and are excluded if they are significant in only a small fraction (<20%) of these tumor types.

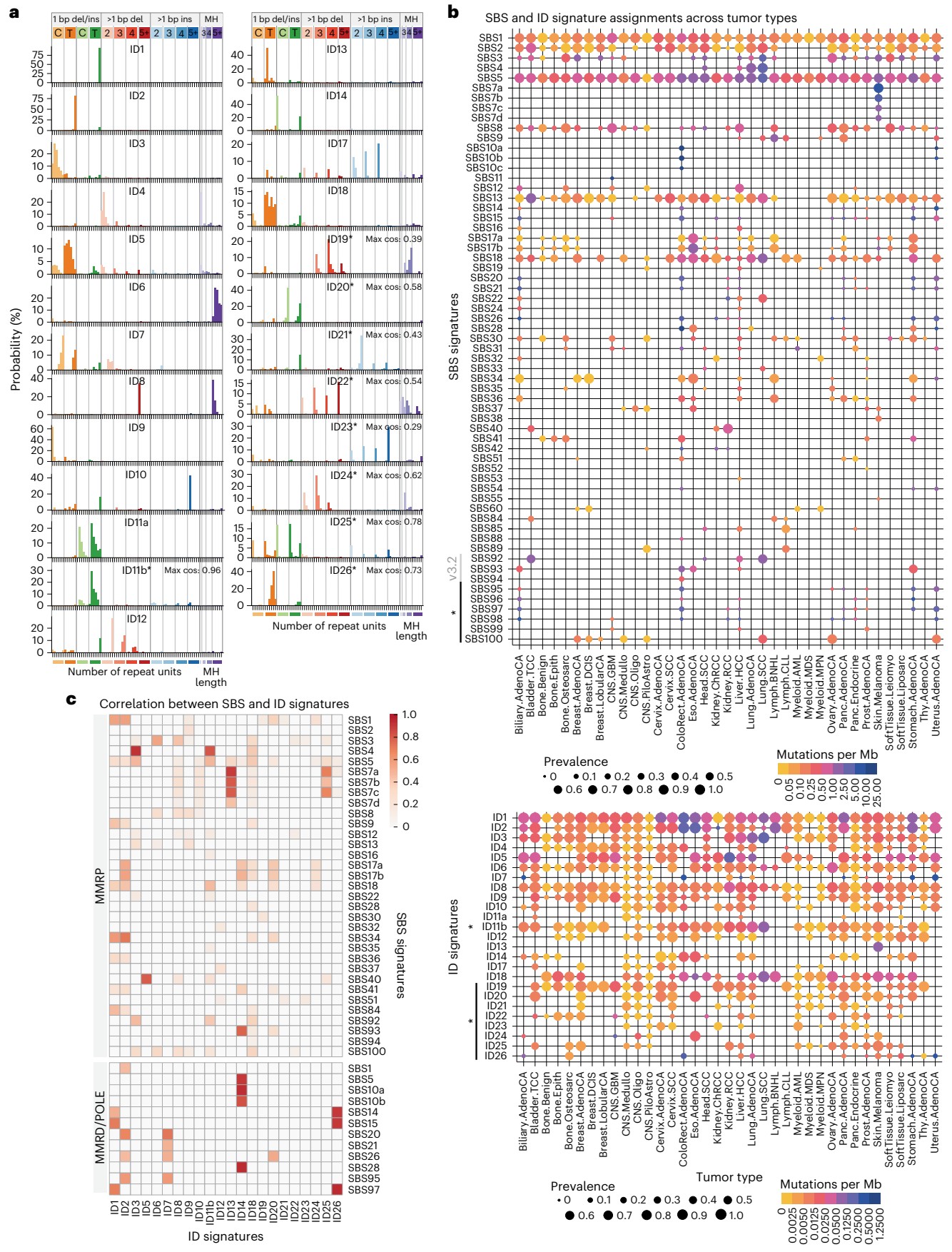

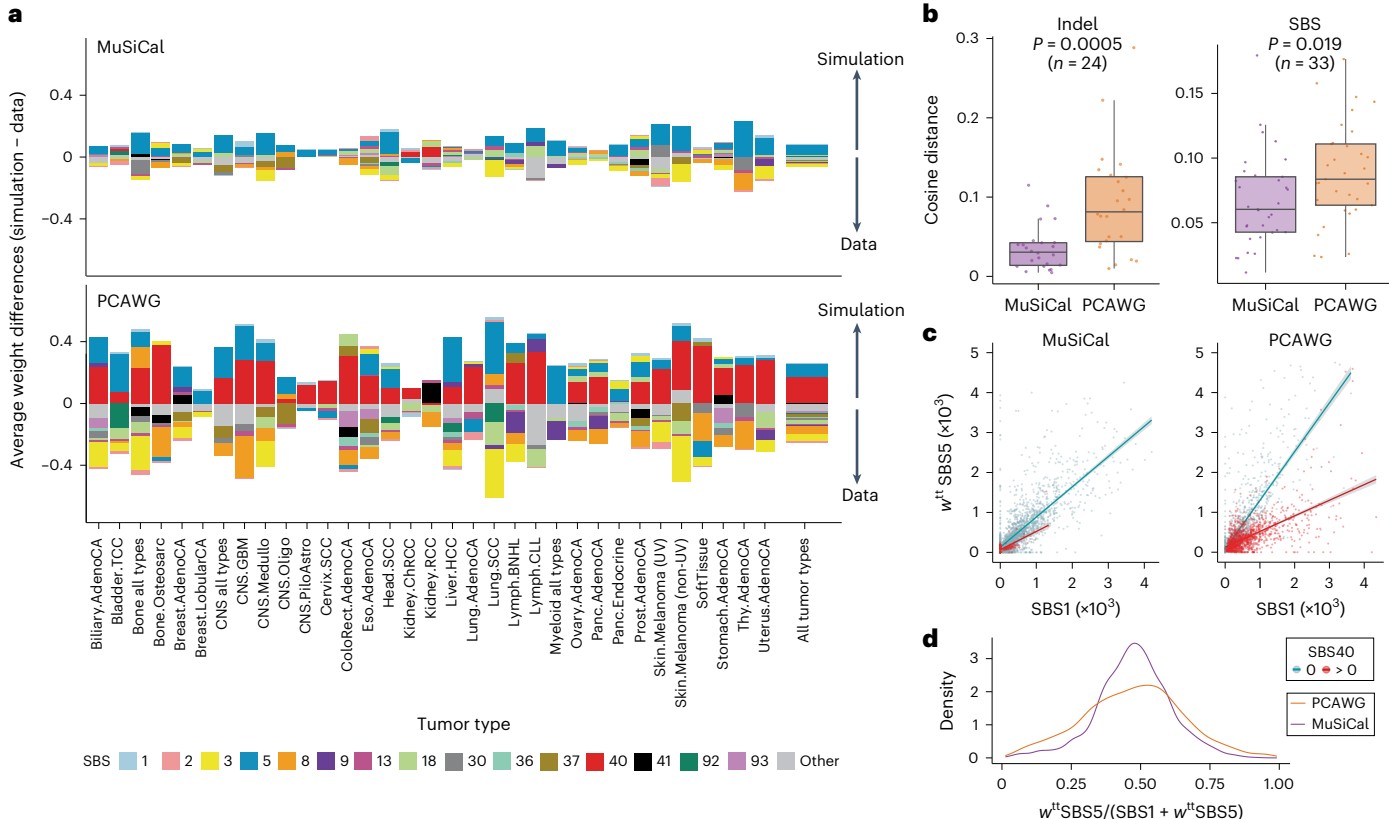

**Fig. 7 | MuSiCal improves the accuracy of signature assignments for PCAWG data and specifically solves the over-assignment problem of SBS40.**
**a**, Differences between the de novo signatures extracted from simulation ($W_{simul}$) and from data ($W_{data}$) in terms of NNLS weights obtained by matching the corresponding de novo signatures to assigned signatures. The differences are calculated in the same way as in Fig. 5f, except that the average differences across de novo signatures from each tumor type are plotted. Positive (excess in simulation, indicating over-assignment) and negative (excess in data, indicating under-assignment) differences are plotted separately in opposite directions in a cumulative manner. PCAWG assignments show widespread over-assignment of SBS40 across diverse tumor types (red bars), whereas MuSiCal assignments demonstrate much improved consistency between simulation and data, as indicated by the overall reduced bar heights. **b**, Mean cosine distances between $W_{simul}$ and $W_{data}$ for ID (left) and SBS (right) signature assignments across different

tumor types. MuSiCal assignments demonstrate improved consistency between simulation and data, as indicated by the smaller cosine distances overall. $P$ values are calculated with two-sided $t$-tests. **c**, Correlation between exposures of SBS1 and SBS5 for MuSiCal (left) and PCAWG (right) assignments. Samples with zero and nonzero exposures of SBS40 are plotted separately. To account for different accumulation rates of the two clock-like signatures in different tumor types, the SBS5 exposure is multiplied by a normalization factor $w^{tt}$, corresponding to the ratio between average SBS1 and SBS5 exposures for a given tumor type (tt) (Supplementary Fig. 13a). Error bands (shaded areas) indicate 95% confidence intervals of the linear regressions. **d**, Distribution of the fraction of weighted SBS5 exposure over the sum of SBS1 and weighted SBS5 exposures. MuSiCal assignments demonstrate improved correlation between SBS1 and SBS5, as indicated by the tighter distribution.

in the other tumor types (for PCAWG assignments) (Extended Data Fig. 10). Therefore, the assignment of SBS40 in the other tumor types by PCAWG is unlikely to be reliable. The impact of the over-assignment of SBS40 cannot be overstated, because it also results in the widespread under-assignment of other similar or overlapping signatures and thus confounds the interpretation of these signatures. For example, signature assignments from the PCAWG Consortium show distinct ratios between the exposures of SBS1 and SBS5 (clock-like signatures) in samples with and without SBS40, even when tumor type-specific accumulation rates are factored out (Supplementary Notes, Fig. 7c,d and Supplementary Fig. 13). By comparison, this confounding effect is largely alleviated in MuSiCal-derived signature assignments (Fig. 7c,d and Supplementary Fig. 13).

## Discussion

Mutational signature analysis has become routine in cancer genome analysis[48]. In this work, we developed MuSiCal that outperforms state-of-the-art algorithms for both signature discovery and assignment. MuSiCal's improved performance is further demonstrated by the improved consistency of MuSiCal-derived signature assignments with

biological ground truth in real data when such ground truth is known, as is the case for homologous recombination deficiency-associated SBS3 (Supplementary Fig. 14) and platinum-associated SBS31/35 (Supplementary Fig. 15). Although MuSiCal solves several problems in the standard workflow, other methods that perform specific tasks—such as joint discovery of different signature types[49], incorporation of epigenetic data[50] and utilization of clinical information[51]—are complementary and valuable additions to the signature analysis toolbox. There are also many other outstanding methodological challenges requiring future developments, which could potentially benefit from recent advances in machine learning and artificial intelligence[52].

The number of whole genomes of cancer and other diseases continues to grow rapidly, especially by consortium projects such as Genomics England[21,53] and the Hartwig Medical Foundation's metastatic tumor project[54,55]. Applications of MuSiCal to these datasets will further refine the set of mutational signatures and facilitate comparison of signatures from different contexts, such as tumor types, metastatic status and treatments received.

Mutational signature catalogs (such as those from COSMIC) serve as a crucial resource for the interpretation of mutational spectra.

Our reanalysis of PCAWG data with MuSiCal provides a revision and expansion of the catalog of ID signatures. For SBS signatures, our reanalysis demonstrates that there are still new signatures to be discovered even from existing data and that the spectra of some known signatures—such as SBS40 (Supplementary Fig. 9)—need to be revised. Our analysis is based on the standard signature definitions (the 96-channel trinucleotide signatures for SBSs and the 83-channel signatures for IDs), which lack more detailed context information. As a result, some of the identified signatures may correspond to multiple mutational processes, thus leading to uncertainties in the interpretation of some signature assignment results (Supplementary Fig. 16). Application of MuSiCal to alternative signature definitions should further disentangle mutational processes and help gain insights into signatures with unknown etiologies. Finally, signatures of other mutation types, including copy number variations and SVs, are less well studied[6,56–58]. We expect that MuSiCal will facilitate future developments of these signatures when their proper definitions are developed.

## Online content

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

## Methods

### Ethical approval

This study is based on the reanalysis of previously published data and complies with ethical regulations within the respective cohorts (The Cancer Genome Atlas and International Cancer Genome Consortium).

### The MuSiCal algorithm

The input of MuSiCal is a $p$ by $n$ matrix $X$ of mutation counts, where $p$ denotes the number of features (for example, $p = 96$ for SBS signatures with trinucleotide contexts) and $n$ denotes the number of samples.

**De novo signature discovery.** In de novo signature discovery, MuSiCal decomposes $X$ into the product of a $p$ by $r$ signature matrix $W$ and a $r$ by $n$ exposure matrix $H$, where $r$ denotes the number of signatures.

 **Minimum-volume NMF.** Standard NMF tries to factorize $X$ as a product of $W$ and $H$ by minimizing the reconstruction error $L = D(X|WH)$, where $D$ denotes a distance measure, such as the Frobenius norm or the generalized Kullback–Leibler (KL) divergence. However, any invertible matrix $Q$ provides an equivalent pair of solutions $(WQ, Q^{-1}H)$ as long as the resulting matrices are nonnegative, resulting in NMF solutions being nonunique. MuSiCal employs mvNMF to solve this nonuniqueness problem[30–33]. In mvNMF, the following optimization problem is solved[33]:

$$\min_{W \geq 0, H \geq 0} \left( D(X|WH) + \lambda \log \det(W^T W + \delta I) \right), \tag{1}$$

where $\lambda$ is the penalty parameter controlling the strength of volume regularization, $\delta$ is a positive constant preventing $\log \det(W^T W)$ from going to $-\infty$ and $I$ is the identity matrix. In MuSiCal, we use KL divergence for the distance measure and set $\delta$ to 1. We follow the algorithm in Leplat et al.[33] to solve the optimization problem above. Specifically, $W$ and $H$ are randomly initialized and alternately updated while keeping the other one fixed. When $W$ is fixed, the problem is equivalent to standard NMF, and thus the well-known multiplicative update is used to update $H$ (ref. [28]). When $H$ is fixed, a revised multiplicative update is used to update $W$, combined with a backtracking line-search procedure to guarantee that columns of $W$ are $L_1$ normalized[33]. The updates are iterated until the relative change of objective falls below a tolerance threshold.

 **Selecting the regularization parameter.** In mvNMF, the regularization parameter $\lambda$ needs to be carefully chosen for optimal performance. To make $\lambda$ less dependent on the number of samples or mutations and thus more generalizable across different datasets, we set $\lambda = \tilde{\lambda} \frac{D(X|W_{ini}H_{ini})}{\log \det(W_{ini}^T W_{ini} + \delta I)}$, where $W_{ini}$ and $H_{ini}$ are initializations for $W$ and $H$, respectively, and tune $\tilde{\lambda}$ instead[32]. To tune $\tilde{\lambda}$, mvNMF is run for a logarithmic grid (by default, from $10^{-10}$ to 2) of $\tilde{\lambda}$ values separately. The solutions for different $\tilde{\lambda}$ values are then compared with the one for the smallest $\tilde{\lambda}$ (by default, $10^{-10}$), which is nearly equivalent to the standard NMF solution without any regularization. We reason that, as long as the solution is not significantly worse than the NMF solution in reconstructing $X$, the strongest volume regularization should be favored. Therefore, we choose the largest $\tilde{\lambda}$ value such that the reconstruction error $D(X|WH)$ is not significantly worse when compared with the solution with the smallest $\tilde{\lambda}$. Specifically for comparing reconstruction errors, two statistical tests are jointly used. First, the Mann–Whitney $U$-test is used to compare sample-wise reconstruction errors. Second, to better capture potential outliers that receive exceptionally large errors, a test for differences in distribution tails is used[59]. A solution is said to be significantly worse if either of the two tests gives a $P$ value smaller than 0.05. This procedure ensures that the regularization does not significantly sacrifice reconstruction accuracy, while still promoting uniqueness of the solutions (Supplementary Fig. 17).

 **Solution filtering and clustering.** For a fixed choice of $r$ (number of signatures), mvNMF is run multiple times (by default, 20) with

different initializations. Of note, $\tilde{\lambda}$ tuning is performed only once, and the selected $\tilde{\lambda}$ is used for the rest of the mvNMF runs, because we have observed that the optimal $\tilde{\lambda}$ is stable with a fixed $r$ and $X$, even when $X$ is bootstrapped before each run. To get rid of occasional bad solutions potentially caused by failed convergence, solutions with reconstruction errors greater than five median absolute deviations above the median are filtered out. De novo signatures (columns of $W$) of the remaining solutions are clustered via hierarchical clustering, and cluster means are taken as the final solutions for de novo signatures.

 **Selecting the number of signatures.** In practice, the true number of signatures $r$ is not known a priori and needs to be inferred from the data itself. MuSiCal employs a new method for automatic selection of $r$ (Extended Data Fig. 2). After running mvNMF multiple times with different initializations for each candidate $r$ and filtering out potential bad solutions (as described above), previous methods usually proceed by clustering the signatures from solutions with the same $r$ into $r$ clusters, and choosing the best $r$ such that the reconstruction error is reasonably small and the solutions are reasonably stable, as quantified by the silhouette score of the clustering[2,17,20]. However, the exact choice of $r$ is either manual or heavily dependent on predefined thresholds. More importantly, forcing the number of clusters to be equal to $r$ seems artificial because multiple mvNMF/NMF runs may in fact produce different signatures, especially when $r$ is misspecified (see, for example, Extended Data Fig. 2c). We therefore propose to cluster the signatures from solutions with the same $r$ into $k$ clusters, allowing $k \neq r$, where the optimal $k$ is determined by the well-established gap statistic[60,61]. We reason that, at the true underlying $r$, solutions from multiple mvNMF/NMF runs should be stable and comprise the same signatures, leading to $k = r$. By contrast, when $r$ is over-specified, mvNMF/NMF should start discovering redundant or split signatures, resulting in $k \neq r$ (most likely, $k < r$) (see, for example, Extended Data Fig. 2g,h). We thus select the largest $r$ among those with $k = r$ as the inferred number of signatures (Extended Data Fig. 2b). This method is parameter-free and more robust. In practice, $k = r$ can happen accidentally when $r$ is over-specified. We therefore additionally check the per-signature silhouette scores to ensure that these occasional cases are not selected. By default, the largest $r$ with $k = r$ as well as a mean silhouette score $\geq 0.7$ and a minimum silhouette score $\geq 0.2$ is selected.

**Matching and refitting.** Following mvNMF-based de novo signature discovery, matching and refitting are performed. In matching, each de novo signature is matched to a known catalog, for example, from COSMIC, and decomposed as a nonnegative mixture of potentially multiple signatures from the catalog. In refitting, the original data $X$ is refit against matched signatures from the catalog to recalculate the corresponding exposures. MuSiCal uses a new multinomial likelihood-based sparse NNLS algorithm for both matching and refitting.

 **Multinomial likelihood-based sparse NNLS.** We describe a generic problem of fitting a sample spectrum, denoted as a $p$-dimensional column vector $\mathbf{x}$ of mutation counts (for example, a column of $X$), against a set of signatures, represented as columns of the $p$ by $r$ matrix $W$. $W$ can be the set of matched signatures from the catalog if refitting is performed after de novo discovery and matching, or it can be any user-specified signatures if refitting is performed as a standalone task.

 In standard NNLS, the $r$-dimensional column vector $\mathbf{h}$ of exposures is estimated by solving the following optimization problem:

$$\min_{\mathbf{h} \geq 0} \| \mathbf{x} - W\mathbf{h} \|_2^2, \tag{2}$$

where $\| \cdot \|_2$ denotes the $L_2$ norm. NNLS lacks explicit sparsity constraints and leads to the over-assignment problem. We thus propose a new multinomial likelihood-based sparse NNLS algorithm (Extended Data Fig. 5a). In this algorithm, we first perform NNLS to obtain an

initial exposure estimate $\mathbf{h}^{(0)}$ and then refine the exposure vector in a bidirectional stepwise manner.

Let $\mathbf{h}^{(m)}$ be the exposure estimate at the $m$th iteration. The multinomial log likelihood of observing $\mathbf{x}$ according to $\mathbf{h}^{(m)}$ is

$$\log L^{(m)}(\mathbf{h}^{(m)}|\mathbf{x}) = \sum_{i=1}^{p} x_i \log w_i^{(m)}, \qquad (3)$$

where $\mathbf{w}^{(m)} = \frac{W\mathbf{h}^{(m)}}{\sum_{i=1}^{p}(W\mathbf{h}^{(m)})_i}$ is the normalized probability vector of the composite signature corresponding to $\mathbf{h}^{(m)}$, and the constant combinatorial factor is dropped because it is independent of $\mathbf{h}^{(m)}$ and depends only on the fixed $\mathbf{x}$. Let $R^{(m)} = \{j = 1, 2, \dots, r | h_j^{(m)} > 0\}$ be the set of indices of nonzero exposures in $\mathbf{h}^{(m)}$; that is, the support of $\mathbf{h}^{(m)}$ or the set of indices corresponding to signatures believed to be active at the $m$th iteration. Let $\bar{R}^{(m)} = \{j = 1, 2, \dots, r | h_j^{(m)} = 0\}$ be the complement of $R^{(m)}$.

During the backward step, a currently active signature is excluded if removing it does not substantially reduce the multinomial likelihood of the observed mutation count vector $\mathbf{x}$. Specifically, we attempt to exclude each active signature in turn and calculate the revised multinomial log likelihood $\log L^{(m)(-j)}(\mathbf{h}^{(m)(-j)}|\mathbf{x}), \forall j \in R^{(m)}$, where $\mathbf{h}^{(m)(-j)}$ is the exposure vector returned by NNLS when $\mathbf{x}$ is fit against the signatures with indices $R^{(m)} \backslash \{j\}$, while all other signatures corresponding to $R^{(m)} \cup \{j\}$ are forced to have zero exposures. The revised log likelihoods $\log L^{(m)(-j)}$ are then compared with the original $\log L^{(m)}$; the signature $\hat{j}$ with the smallest drop in log likelihood when its exposure is additionally forced to be zero is selected; signature $\hat{j}$ is excluded if the drop in log likelihood is smaller than a predefined positive threshold $\epsilon > 0$; otherwise, $\mathbf{h}^{(m)}$ is kept unchanged. That is,

$$\mathbf{h}^{(m+1)} = \mathbf{h}^{(m)(-\hat{j})}, \text{ if } \log L^{(m)} - \log L^{(m)(-\hat{j})} < \epsilon,$$

$$\mathbf{h}^{(m+1)} = \mathbf{h}^{(m)}, \text{ if } \log L^{(m)} - \log L^{(m)(-\hat{j})} \geq \epsilon, \qquad (4)$$

$$\hat{j} = \arg\min_{j \in R^{(m)}}(\log L^{(m)} - \log L^{(m)(-j)}).$$

Similarly, during the forward step, a currently inactive signature is included if adding it substantially increases the multinomial likelihood of observing $\mathbf{x}$,

$$\mathbf{h}^{(m+1)} = \mathbf{h}^{(m)(+\hat{j})}, \text{ if } \log L^{(m)(+\hat{j})} - \log L^{(m)} > \epsilon,$$

$$\mathbf{h}^{(m+1)} = \mathbf{h}^{(m)}, \text{ if } \log L^{(m)(+\hat{j})} - \log L^{(m)} \leq \epsilon, \qquad (5)$$

$$\hat{j} = \arg\max_{j \in \bar{R}^{(m)}}(\log L^{(m)(+j)} - \log L^{(m)}),$$

where $\mathbf{h}^{(m)(+j)}$ is the exposure vector returned by NNLS when $\mathbf{x}$ is fit against the signatures with indices $R^{(m)} \cup \{j\}, \forall j \in \bar{R}^{(m)}$, while all other signatures $\bar{R}^{(m)} \backslash \{j\}$ are forced to have zero exposures.

The backward and forward steps are alternately iterated until no signature is removed or added. The same likelihood threshold $\epsilon$ is used for both steps to reduce the number of parameters and ensure that successive backward and forward steps do not remove or add the same signature. We have observed in simulation studies that even when different thresholds are allowed for backward and forward steps, the global optimum is usually achieved with the same threshold.

Of note, the log likelihood in equation (3) is divided by the total mutation count $\sum_{i=1}^{p} x_i$ in practice, leading to a per-mutation log likelihood. This normalization makes the likelihood threshold $\epsilon$ independent of the total mutation count (thus more generalizable) and turns out to be crucial for adapting the algorithm for matching (see below).

**Adapting the likelihood-based sparse NNLS for matching.** Matching and refitting are essentially the same mathematical problem, except that in matching, $\mathbf{x}$ is an $L_1$-normalized probability vector corresponding to the de novo signature instead of an integer vector of mutation counts. The multinomial log likelihood of observing a probability vector $\mathbf{x}$ as in equation (3) may seem ill-defined at first glance.

However, observe that equation (3) is equivalent to the KL divergence between $\mathbf{x}$ and $\mathbf{w}^{(m)}$ if the constant term independent of $\mathbf{h}^{(m)}$ is dropped. Note that both $\mathbf{x}$ and $\mathbf{w}^{(m)}$ are well-defined probability vectors—$\mathbf{w}^{(m)}$ is $L_1$-normalized by definition; $\mathbf{x}$ is $L_1$-normalized in matching because it is in fact a de novo signature; and in refitting, the normalization by total mutation count effectively turns $\mathbf{x}$ into a probability vector. Therefore, the multinomial likelihood-based sparse NNLS can be alternatively interpreted as a bidirectional stepwise NNLS based on KL divergence. In fact, this correspondence is natural given the keen relation between likelihood and KL divergence. Consequently, the same algorithm can be directly used for matching without any modifications.

**Data-driven parameter optimization and in silico validation.** MuSiCal employs a new data-driven approach for optimizing parameters involved in post-processing steps after de novo signature discovery, as well as validating the consistency between the final signature assignment and the original data in silico (Fig. 5a).

Let $X_{\text{data}}$ be the original data matrix of mutation counts, and $(W_{\text{data}}, H_{\text{data}})$ the corresponding de novo discovery results. After matching and refitting with properly chosen parameters (see below), the final signature assignments $(W_s, H_s)$ are obtained, where s stands for sparse. To assess the quality of $(W_s, H_s)$, we simulate a synthetic mutation count matrix $X_{\text{simul}}$ from $(W_s, H_s)$ through multinomial sampling, and perform de novo signature discovery from $X_{\text{simul}}$ with exactly the same settings as used for $X_{\text{data}}$. In particular, the selected number of signatures $r$ and mvNMF regularization parameter $\tilde{\lambda}$ are kept the same. Let $(W_{\text{simul}}, H_{\text{simul}})$ be the de novo discovery results from $X_{\text{simul}}$. The distances between $(W_{\text{simul}}, H_{\text{simul}})$ and $(W_{\text{data}}, H_{\text{data}})$ are then used to quantify the consistency between $(W_s, H_s)$ and $X_{\text{data}}$. In more detail, a one-to-one correspondence between signatures in $W_{\text{simul}}$ and $W_{\text{data}}$ is determined to minimize the mean cosine distance, and the resulting mean cosine distance is used to quantify the overall quality of $(W_s, H_s)$.

MuSiCal also applies the same scheme for optimizing post-processing parameters (Supplementary Fig. 8). Two likelihood thresholds, corresponding to the likelihood-based sparse NNLS used for matching and refitting, respectively, are optimized through a two-dimensional grid search. Specifically, the threshold pair that minimizes the mean cosine distance between $W_{\text{simul}}$ and $W_{\text{data}}$ (as described above) is selected as a candidate solution. Additional threshold pairs are further included in the candidate list if the corresponding difference between $W_{\text{simul}}$ and $W_{\text{data}}$ is not significantly larger (if both the Mann–Whitney $U$-test and the test for differences in distribution tails[59] result in $P$ values >0.05 for the element-wise $L_1$ error). Then, among these candidate threshold pairs, the one with the smallest number of assigned signatures is selected, and if there are multiple such pairs, the one with the largest thresholds (the sparsest signature assignment) is selected. Alternatively, to reduce computation time, two separate one-dimensional grid searches can be performed (Supplementary Fig. 8): the matching threshold is first optimized while the refitting threshold is fixed to a small value (0.0001), and the refitting threshold is subsequently optimized while the matching threshold is fixed to its optimal value.

It is worth pointing out that the application of this data-driven approach is not limited to post-processing steps in MuSiCal or MuSiCal-derived signature assignments. Indeed, the quality of signature assignments obtained by the PCAWG Consortium can be assessed within the same framework, and potential issues can be detected (for example, see Fig. 5). Parameters used in post-processing steps of other signature analysis tools can also be optimized with the same approach, although a high-dimensional grid search can be prohibitive when many parameters are involved.

**Preprocessing.** MuSiCal provides two complementary methods for preprocessing to improve the sensitivity of de novo signature discovery (Extended Data Fig. 6).

**Automatic cohort stratification**. In automatic cohort stratification, samples in the input matrix $X$ are clustered into potential subsets with distinct signature compositions; for example, MMRD versus mismatch repair proficient (MMRP), samples with versus without aristolochic acid exposure (Extended Data Fig. 6a–c). Such stratifications may benefit de novo signature discovery by allowing subtle signatures within each subset to be discovered cleanly. Specifically, $X$ is subject to hierarchical clustering, where the optimal number of clusters $k$ is determined by the gap statistic[60,61]. Compared with other metrics (such as the silhouette score) that are ill-defined when the number of clusters is 1, the gap statistic handles $k = 1$ gracefully and is able to select $k = 1$ when there is in fact no subset structure within the dataset. When $k > 1$ is selected, de novo signature discovery is run on the corresponding subsets of samples separately. Alternatively, the clustering is performed on the de novo exposure matrix $H$ after an initial de novo signature discovery on $X$ with all samples. We have observed that the latter approach of clustering $H$ better captures inherent subset structures within the dataset and is thus recommended. Of note, cohort stratification can lead to subsets with small numbers of samples, reducing the power of signature discovery. Therefore, the suggested stratification needs to be investigated in a case-specific manner to maximize the benefit for signature discovery (see below for detailed approach taken for reanalysis of PCAWG data).

**Outlier removal based on the Gini coefficient**. When the dataset contains a small number of samples with strong exposures of signatures that are not present in other samples, these strong signatures can reduce the sensitivity of detecting other more prevalent signatures, or introduce biases in them, during de novo signature discovery. We therefore detect such outlier samples and remove them to increase the sensitivity and accuracy of signature discovery (Extended Data Fig. 6d–f). Specifically, after an initial signature discovery with the entire dataset, we investigate the distribution of de novo exposures in $H$ with the Gini coefficient for each de novo signature. The Gini coefficient quantifies the inequality among the per-sample exposures; a small Gini coefficient is expected for a homogeneous dataset with similar signature compositions, whereas strong outliers are indicated by a large Gini coefficient. We thus first select signatures for which the exposures have a Gini coefficient greater than a baseline threshold (by default 0.65). Then, for each selected signature, samples are inspected one by one in descending order of the corresponding relative exposure, and a sample is excluded if removing it results in a decrease in the Gini coefficient greater than a threshold $\delta_{Gini}$ (by default, 0.05).

**Computational cost.** For de novo signature discovery, MuSiCal (with mvNMF) requires similar but slightly more computational time compared with SigProfilerExtractor (with NMF), and considerably less memory (Supplementary Fig. 18). The computational costs of matching and refitting steps in MuSiCal are negligible compared with de novo signature discovery. The in silico parameter optimization step requires rerunning de novo signature discovery for a grid of thresholds. But during this grid search, there is no need to select the regularization parameter in mvNMF or the number of signatures (because they are both fixed), which are the most time-consuming calculations. Thus, the computational cost for in silico optimization is also small compared with de novo signature discovery when parallelized properly.

### Simulation studies
**Simulation studies comparing mvNMF and NMF for de novo signature discovery.** To compare mvNMF and NMF for de novo signature discovery (Fig. 2c and Extended Data Fig. 1), synthetic datasets were simulated from tumor type-specific SBS signatures and Dirichlet-distributed exposures. In more detail, SBS signatures specifically present in 32 PCAWG tumor types were read out from Fig. 3 in the PCAWG study[5]. Random exposures were generated from symmetric Dirichlet distributions with a concentration parameter of $\alpha = 0.1$, which

is a representative value according to real exposure matrices obtained by the PCAWG Consortium[5] (Supplementary Fig. 3). Mutation count matrices were then simulated through multinomial sampling, and 20 independent simulations were performed for each tumor type. To focus on algorithmic differences between NMF and mvNMF, each synthetic dataset was allowed to contain 200 samples and on average 5,000 mutations per sample (Poisson-distributed), such that there was enough power to discover each signature involved. The number of signatures was also assumed to be known. NMF and mvNMF were then applied to derive de novo signatures, and the same random initialization was used for both algorithms for a given dataset to facilitate comparison. For NMF, the implementation within MuSiCal according to the multiplicative update algorithm with KL divergence was used[28]. Simulation was performed with the function simulate_LDA() from the simulation module in MuSiCal (musical.simulation).

**Simulation studies benchmarking MuSiCal and SigProfilerExtractor for de novo signature discovery.** To benchmark the performance of MuSiCal and SigProfilerExtractor for de novo signature discovery (Fig. 3 and Supplementary Fig. 5), more realistic synthetic datasets were simulated to capture the complexity present in real-life data. Specifically, tumor type-specific SBS signatures were combined with real exposure matrices obtained by the PCAWG Consortium[5] to simulate mutation count matrices through multinomial sampling (using the function simulate_count_matrix() from musical.utils). Ten independent simulations were performed for each of the 25 tumor types with at least 20 samples remaining after removing hypermutated samples with signatures associated with MMRD or DNA polymerase mutations (SBS6, 9, 10, 14, 15, 20, 21, 26 and 44). Samples with MMRD or DNA polymerase mutations are usually analyzed as a separate group for improved sensitivity of signature discovery[5] and thus excluded. MuSiCal and SigProfilerExtractor were then applied to identify de novo signatures. For SigProfilerExtractor, v1.1.3 was used with default parameters (-i random -b True -nr 100 -min 10000 -max 1000000 -conv 10000 -tol 1e-15). Input normalization was turned off (-nx none) except in the results denoted by SigProfilerExtractor-norm in Supplementary Fig. 5, where it was turned on with -nx gmm. For MuSiCal, the preprocessing module was skipped. In Extended Data Fig. 3, we added random Gaussian noise at different levels (1%, 2.5%, 5% and 10%) to the simulated genomes described above (which already included random Poisson sampling noises). In more detail, we followed[17] and resampled each element of the mutation count matrix from a Gaussian distribution where the mean was equal to the original count and the standard deviation was equal to the original count multiplied by the noise level. The resulting matrix was then rounded to the closest integers, and negative values were set to 0. In Supplementary Fig. 6, we spiked in mutations from COSMIC SBS48 to the simulated genomes. The number of spike-in mutations was equal to 5% of the mutational burden for each sample. SBS48 was chosen to represent a spurious signature because it is not present in the original synthetic dataset and is annotated in COSMIC as a possible sequencing artifact found in cancer samples that were subsequently blacklisted for poor quality of sequencing data. To pretend that SBS48 was unknown, we removed SBS48 from the COSMIC catalog when matching the de novo signatures and calculating precision and recall. The auPRC scores were calculated with sklearn.metrics.average_precision_score() from scikit-learn v0.24.1 (ref. 62), which is more accurate than using the trapezoidal rule.

**Simulation studies benchmarking other existing tools for de novo signature discovery.** We benchmarked the performance of MuSiCal for de novo signature discovery mainly against SigProfilerExtractor because it has been the most popular tool in the field and was shown to outperform a number of other existing tools in a recent benchmark study[17]. COSMIC signatures and the PCAWG study also largely relied on SigProfilerExtractor. To be more comprehensive in the benchmark

analysis (Extended Data Fig. 4), we selected three additional tools based on the consideration that different underlying algorithms should be represented and popular tools (especially those used in large landmark studies) should be included. Specifically, signature.tools.lib[20,21] is based on an R implementation of NMF (SigProfilerExtractor is based on a Python implementation of NMF), included several methodological improvements[20] and was used in the recent landmark study of the Genomics England Cohort[21]; SignatureAnalyzer[18,19,38] is based on a Python implementation of Bayesian NMF and was used in the PCAWG study as well[5]; and SigneR is based on an R implementation of Bayesian NMF[39]. Compared with the four included in our benchmark analysis, most other existing tools differ only in implementation details or specific functionalities[17,22].

For signature.tools.lib, v2.1.2 was used with recommended parameters (nrepeats=200, nboots=20, clusteringMethod="MC", filterBestOfEachBootstrap=TRUE, filterBest_RTOL=0.001, filterBest_nmaxtokeep=10, nmfmethod="brunet"). Because signature.tools.lib does not select the optimal number of signatures automatically, we followed the recommendation described in https://github.com/Nik-Zainal-Group/signature.tools.lib and performed manual selection based on norm.Error (orig. cat.) and Ave.SilWid.MC by visually inspecting the produced plots. For SignatureAnalyzer, v0.0.7 was used with default parameters, except that more iterations than default were used (-n 20). For SigneR, v1.22.0 was used with default parameters.

### Simulation studies benchmarking different refitting algorithms.
To benchmark the performance of different algorithms for refitting (Fig. 4c and Supplementary Fig. 7), synthetic datasets were simulated from tumor type-specific SBS signatures and Dirichlet-distributed ($\alpha = 0.1$) exposures, similar to those in Fig. 2c and Extended Data Fig. 1, using the function simulate_LDA() from the simulation module in MuSiCal (musical.simulation). Specifically, 100 samples were simulated for each of the 32 tumor types and combined as a single dataset. Each sample contained 5,000 mutations on average (Poisson-distributed), and 10 independent simulations were performed. To further introduce sparsity in the exposure matrices, signatures with relative exposures <0.01 as sampled from the Dirichlet distribution were set to have zero exposures. Four different algorithms were then applied to refit these synthetic samples against the COSMIC catalog. For sigLASSO, v1.1 was used with default parameters and flat priors. For SigProfilerExtractor, the Decomposition module from v1.1.3 was used with default parameters, except that connected_sigs = False was set to avoid the use of empirical rules. In Fig. 4c and Supplementary Fig. 7a–c, precision and recall were calculated by comparing the zero and nonzero entries between the true exposure matrix and the exposure matrix obtained from refitting. The auPRC scores were calculated with sklearn.metrics.average_precision_score() from scikit-learn v0.24.1 (ref. [62]), which is more accurate than using the trapezoidal rule. For likelihood-based sparse NNLS and thresholded NNLS (also implemented in MuSiCal) in Supplementary Fig. 7d,e, the optimal threshold was chosen through a grid search to achieve the maximum correct support discovery rate when averaged across the ten independent simulations.

## Reanalysis of PCAWG data
### SBS signatures
**De novo signature discovery.** Samples with MMRD and/or POLE-exo mutations were isolated and analyzed separately in a tissue-independent manner. To identify these samples, we inspected the exposures of SBS signatures known to be associated with MMRD and/or POLE-exo mutations (MMRD: SBS6, 14, 15, 20, 21, 26 and 44; POLE-exo: SBS10a–d and 28), together with the microsatellite mutation rate as obtained from Fujimoto et al.[63] (Supplementary Fig. 19a,b). MMRD/POLE tumors showed a clear separation from the MMRP ones based on these metrics and were subsequently identified (40 tumors with MMRD, 9 with POLE-exo mutations and 1 with concurrent MMRD and

POLE-exo mutations) (Supplementary Fig. 19a,b). Specifically for this identification step, exposures of relevant signatures were determined simply by standard NNLS using all signatures in the COSMIC catalog. Then, de novo signature discovery followed by matching and refitting was performed to determine the final signature assignments. During the initial signature discovery, we found that signatures also present in MMRP samples (such as SBS1, 2, 4, 7a, 7b, 17b and 30) were mixed with signatures specific to MMRD/POLE-exo mutant tumors. To facilitate the accurate extraction of these MMRD/POLE-specific signatures, we included spectra of the discovered MMRP signatures (scaled by the median SBS count in this cohort) as additional samples and performed de novo signature discovery again.

The remaining samples were grouped into 37 tumor types as defined by the PCAWG Consortium. De novo signature discovery was performed separately for each of the 26 tumor types with at least 20 samples (100 on average). The remaining 11 tumor types had too few samples for independent signature discoveries. For example, there were only two samples for Cervix.AdenoCA, three for Breast.DCIS and four for Myeloid.MDS. These 11 tumor types were of 6 different tissue origins (breast, cervix, myeloid, bone, soft tissue and central nervous system) and thus combined with tumor types with the same tissue origin, respectively, for de novo signature discovery. For example, samples from Breast.DCIS and Breast.LobularCA were combined with those from Breast.AdenoCA, and the resulting larger cohort was used for signature discovery. Of note, the combined cohorts were only used to inform the 11 tumor types with a small number of samples; the 26 tumor types with a sufficient number of samples were still analyzed separately.

In addition to the large variation in number of samples per tumor type, the number of mutations per sample also varied greatly within each tumor type. In particular, the presence of hypermutated samples can substantially influence the signature discovery results, because a single highly mutated sample can dominate the objective function optimized by mvNMF and thus be picked up as a distinct signature even when it is actually a composite spectrum. We therefore performed $L_1$ normalization for columns of the input matrix $X$ before running de novo signature discovery. We also excluded samples with fewer than 500 SBSs, because samples with few mutations can cause biases after $L_1$ normalization. As a result, 5.6% of samples were removed in total, and fewer than 21% of samples were removed from all tumor types except CNS.PiloAstro, in which 91% of samples had fewer than 500 SBSs. CNS.PiloAstro was thus combined with other samples with CNS origin for de novo signature discovery, resulting in 25 tumor types processed through tumor type-specific analysis and 12 tumor types processed through tissue-of-origin-specific analysis in the end.

For each tumor type, the optimal $\tilde{\lambda}$ for mvNMF varied between 0.01 and 0.5 with an average of 0.05. The selected number of signatures $r$ varied between 2 and 15 with an average of 6. The dependence of $\tilde{\lambda}$ and $r$ on the number of samples and SBSs is shown in Supplementary Fig. 19c.

**Implementation of preprocessing.** After an initial step of de novo signature discovery within each tumor type, the Gini-based outlier removal was applied, removing on average 2.3% of samples (ranging between 0% and 7.0% in different tumor types). For Kidney.RCC, Liver.HCC and Skin.Melanoma, we further stratified samples into subsets with distinct signature compositions using hierarchical clustering on the exposure matrix $H$ obtained from the initial de novo signature discovery. The distinct clusters within these tumor types were defined by several hypermutator processes including aristolochic acid, aflatoxin and UV exposures. Note that we separated the MMRD and POLE-exo mutant samples before this stage, although preprocessing can be used to stratify these samples if they had not been isolated beforehand (Extended Data Fig. 6a–c).

**Matching, refitting and validation.** Matching of de novo signatures to the COSMIC catalog and refitting were performed with likelihood-based sparse NNLS. The two likelihood thresholds were

optimized with a grid search using data-driven simulations as in the in silico validation/optimization module of MuSiCal. In the matching step, three signatures—SBS25, 39 and 86—were excluded. These signatures contributed small weights (<0.3) in matching and were usually identified only when other overlapping signatures were present (SBS8, 22 and 35 for SBS25 with overlapping T>A mutations, SBS3 and 13 for SBS39 and 86 with overlapping C>G mutations). Excluding these signatures had minimal effect, resulting in a difference of ~0.02 in the cosine reconstruction errors of de novo signatures. Further, signatures associated with MMRD and/or POLE-exo mutations were only used in matching when analyzing samples with MMRD and/or POLE-exo mutations, and UV-related signatures (SBS7a–d) were limited to the melanoma cohort. Finally, an additional cleaning step was performed after matching to avoid flat signatures being picked up because of tiny backgrounds present in the de novo signatures. Specifically, a matched signature was considered to be a consequence of overfitting to the background if: (1) the sum of its probabilities at the 96 trinucleotide types where it contributed at least 1% was less than 15%; and (2) at the same time, its maximum contribution to any of the trinucleotide types was five times smaller than the maximum probability of the de novo signature.

**ID signatures.** Similar to the analysis of SBS signatures, samples with MMRD and/or POLE-exo mutations were analyzed separately for ID signatures. However, a tumor type-specific de novo signature discovery for ID signatures was more challenging because of the lower number of IDs per sample. We therefore performed joint analysis of multiple tumor types to obtain well-separated signatures. Specifically, we first ran de novo signature discovery with all samples having at least 100 IDs ($n = 2,241$) combined together. We then performed hierarchical clustering with the obtained exposure matrix $H$, which resulted in two major clusters. Each tumor type was subsequently examined to determine which cluster contained the majority of samples from the tumor type. Accordingly, all tumor types were stratified into two groups. De novo signature discovery was performed again on these two groups separately. The resulting signatures from these two groups were combined with those from the initial run with all tumor types to derive the updated ID signature catalog. Signature assignments were still performed for each tumor type separately, where tumor type-specific thresholds were used for the minimum number of IDs per sample to maximize the number of samples retained while reducing samples with low ID counts that deteriorate the result.

### Statistics and reproducibility
This study was designed to be a retrospective analysis of previously published data. No statistical method was used to predetermine sample size. The Investigators were not blinded to allocation during experiments and outcome assessment. For de novo signature discovery in the PCAWG reanalysis, samples with low SBS or indel counts were excluded, as described in detail above. All box plots indicate median (center line), upper and lower quartiles (box limits) and 1.5× interquartile range (whiskers).

### Reporting summary
Further information on research design is available in the Nature Portfolio Reporting Summary linked to this article.

### Data availability
PCAWG data, including mutation count matrices of SBSs and IDs, as well as exposure matrices obtained by Alexandrov et al.[5], were downloaded from https://www.synapse.org/#!Synapse:syn11726601/files/. Additional PCAWG data were downloaded from https://dcc.icgc.org/releases/PCAWG. All PCAWG data used in this paper are in the open tier. COSMIC signatures were downloaded from https://cancer.sanger.ac.uk/signatures/. Source data, including MuSiCal-derived signature catalog from the PCAWG reanalysis, are provided in Supplementary Tables 1–4.

### Code availability
MuSiCal is implemented in Python and available at https://github.com/parklab/MuSiCal (ref. 64). SigProfilerExtractor (v1.1.3) was downloaded from https://github.com/AlexandrovLab/SigProfilerExtractor. sigLASSO (v1.1) was downloaded from https://github.com/gersteinlab/siglasso. signature.tools.lib (v2.1.2) was downloaded from https://github.com/Nik-Zainal-Group/signature.tools.lib. SignatureAnalyzer (v0.0.7) was downloaded from https://github.com/getzlab/SignatureAnalyzer. SigneR (v1.22.0) was obtained from https://www.bioconductor.org/packages/release/bioc/html/signeR.html. Custom scripts and analysis notebooks for reproducing results in the paper are available on Zenodo[65].

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

### Acknowledgements

This work was funded by grants from the National Institutes of Health (grant number R01CA269805 to P.J.P.), Ludwig Center at Harvard (P.J.P.), Cancer Grand Challenges partnership by Cancer Research UK and the Mark Foundation for Cancer Research to the SPECIFICANCER team (P.J.P.), and the Swedish Research Council (grant number 2020-00583 to V.L.). The funders had no role in study design, data collection and analysis, decision to publish or preparation of the paper. We would like to thank D. Glodzik for careful reading of the paper. We gratefully acknowledge the contributions of the many clinical networks across the International Cancer Genome Consortium, The Cance Genome Atlas and other consortia, who provided samples and data. We thank the patients and their families for their participation in the individual projects.

### Author contributions

H.J., D.C.G. and P.J.P. conceived the study. P.J.P. supervised the study. H.J. and D.C.G. developed the algorithms and carried out data analysis, with assistance from B.G., D.B.-I., D.G. and V.L. B.G. helped with implementing the algorithms. B.G. and V.L. helped with software testing. D.B.-I. helped with early tests of the mvNMF algorithm. D.G. helped with early tests of the in silico validation algorithm. H.J., D.C.G. and P.J.P. wrote the paper with input from the other authors. All authors read and approved the final paper.

**Competing interests**

The authors declare no competing interests.

**Additional information**

**Extended data** is available for this paper at

**Supplementary information** The online version contains supplementary
material available at https://doi.org/10.1038/s41588-024-01659-0.

**Correspondence and requests for materials** should be addressed
to Peter J. Park.

**Peer review information** *Nature Genetics* thanks the anonymous
reviewer(s) for their contribution to the peer review of this work. Peer
reviewer reports are available.

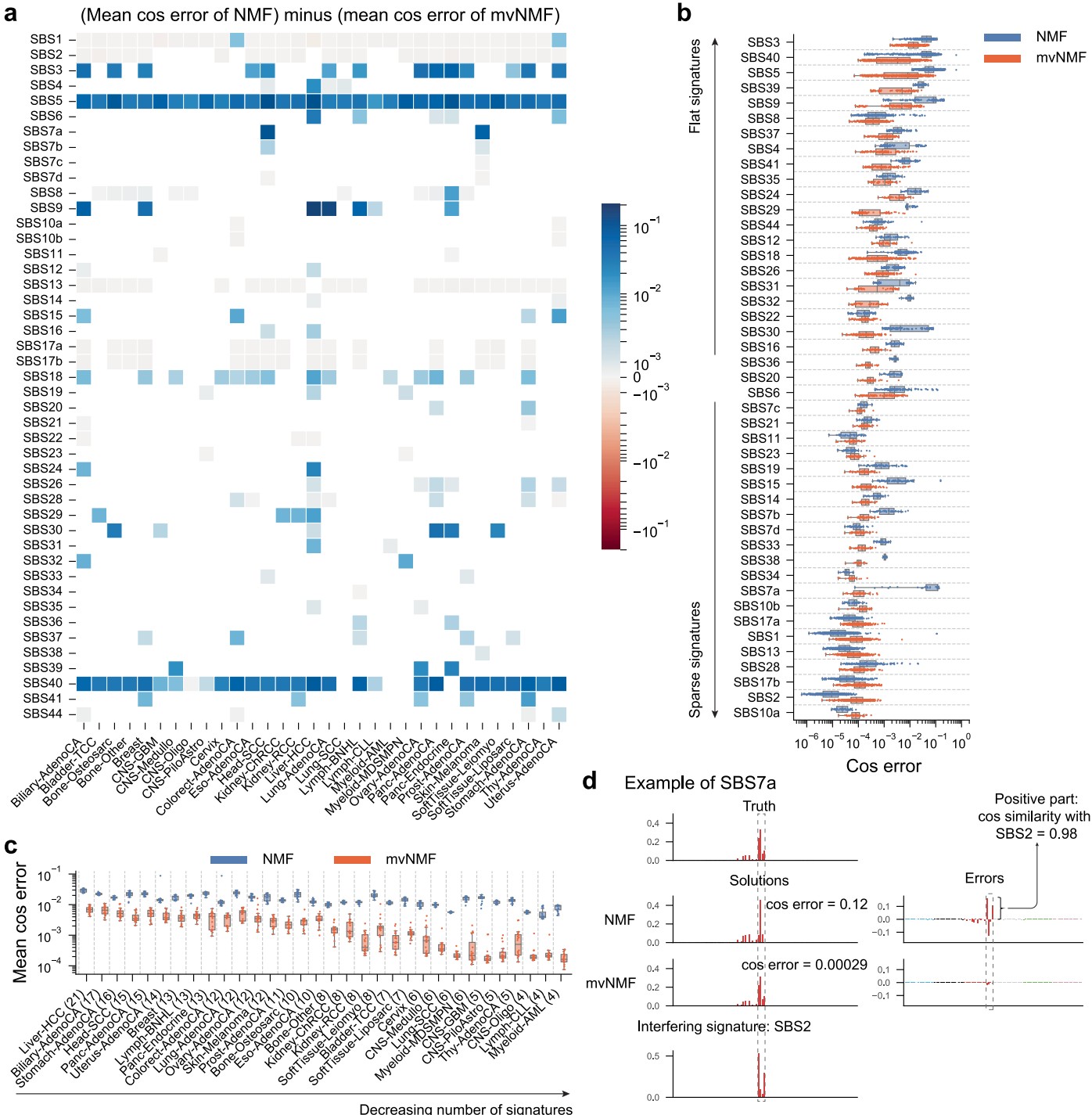

**Extended Data Fig. 1 | mvNMF improves the accuracy of de novo signature discovery in simulated datasets with SBS signatures.** Synthetic datasets are simulated from tumor type-specific SBS signatures and Dirichlet-distributed exposures for 32 tumor types with 20 replicates. Each dataset contains 200 samples and on average 5,000 mutations per sample (Poisson-distributed). NMF and mvNMF are then applied for de novo signature discovery, assuming that the number of signatures is known. Finally, the discovered signatures are compared to the true ones, and their discrepancies are quantified by cosine errors. **a.** Heatmap of the difference between NMF- and mvNMF-derived cosine errors. Each element represents the mean of 20 independent simulations. **b.** NMF- and mvNMF-derived cosine errors for different SBS signatures, sorted by standard deviation of the corresponding signature spectrum. Data from different tumor types are collapsed. Same as Fig. 2c and included for completeness. **c.** NMF- and mvNMF-derived cosine errors for different tumor types, sorted by the number of signatures present in the corresponding tumor type. Data from different signatures are averaged within each tumor type. $n$ = 20 independent simulations for each box plot. **d.** An example comparing the performance of NMF and mvNMF on identifying SBS7a. The NMF solution of SBS7a receives a large cosine error. The error spectrum indicates interference from SBS2 coexisting in the dataset. By comparison, mvNMF does not suffer from the SBS2 interference and is able to discover SBS7a accurately.

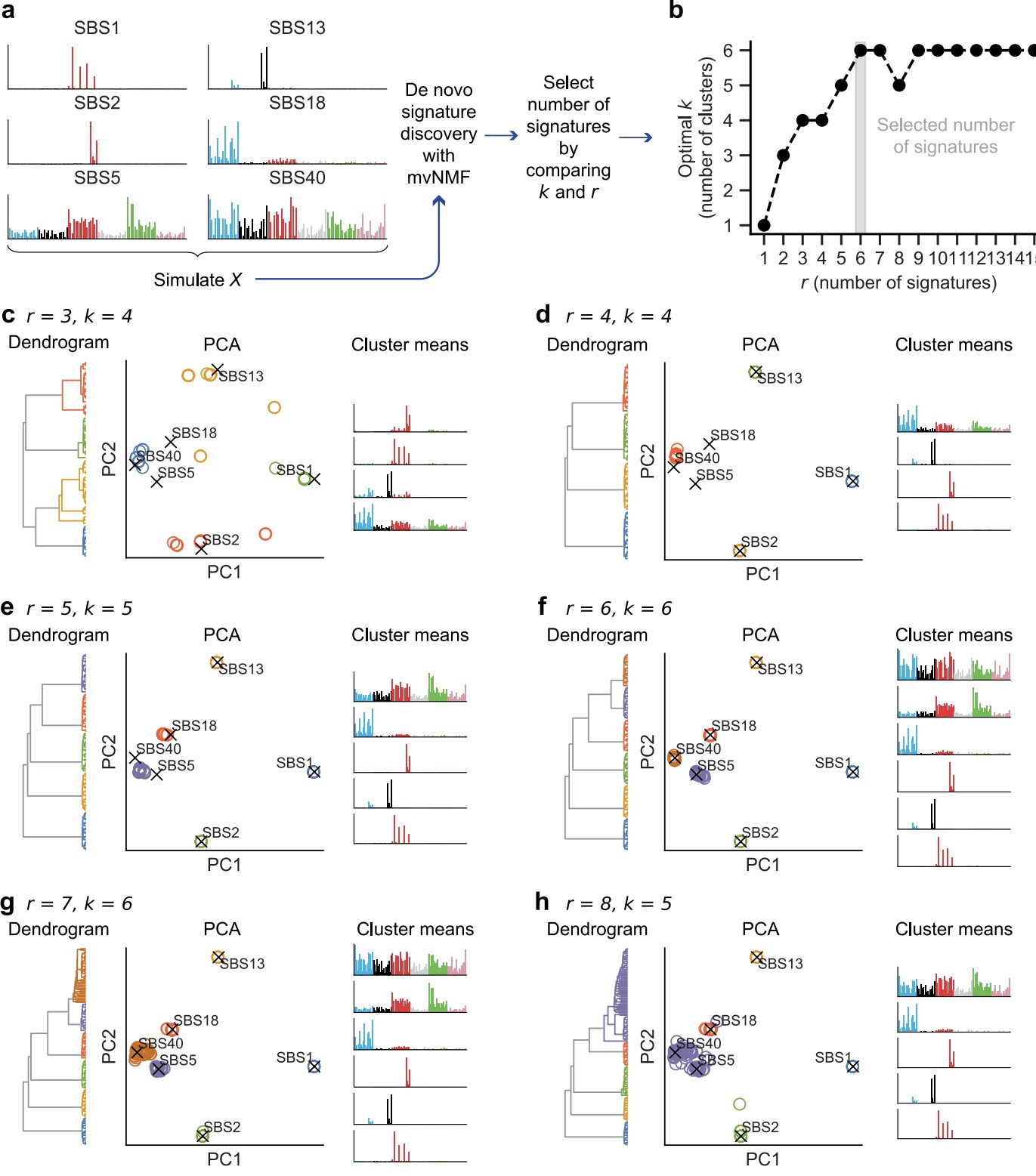

**Extended Data Fig. 2 | An example illustrating the method for selecting the number of signatures $r$. a.** Data simulation scheme. A synthetic dataset is simulated from Dirichlet-distributed exposures and 6 SBS signatures (SBS1, 2, 5, 13, 18, and 40) believed to be present in cervical cancers according to PCAWG results. The dataset contains 200 samples and on average 5000 mutations per sample. mvNMF is then run for $r = 1$ to 15 separately, with 20 independent replicates from different initializations for each $r$ value. **b.** Selecting the number of signatures by comparing $k$ and $r$. De novo signatures from solutions with the same $r$ are clustered into $k$ clusters, where $k$ is determined by the gap statistic[60,61]. The greatest $r$ such that $k = r$ is chosen as the final number of signatures. In this

example, the correct number of signatures ($r = 6$) is selected. **c-h.** Details of mvNMF solutions at different $r$ values. For each $r$ value, the mvNMF-derived de novo signatures from 20 independent runs are clustered and visualized as dendrograms and PCA plots. The signatures corresponding to cluster means are also shown. When $r$ is smaller than the true number of signatures (for example, $r = 3$), the mvNMF solutions can be unstable, resulting in $k > r$. When $r$ is greater than the true number of signatures (for example, $r = 7$), mvNMF may produce redundant signatures, resulting in $k < r$. Together, these observations suggest that the greatest $r$ with $k = r$ is a reasonable estimate of the true number of signatures.

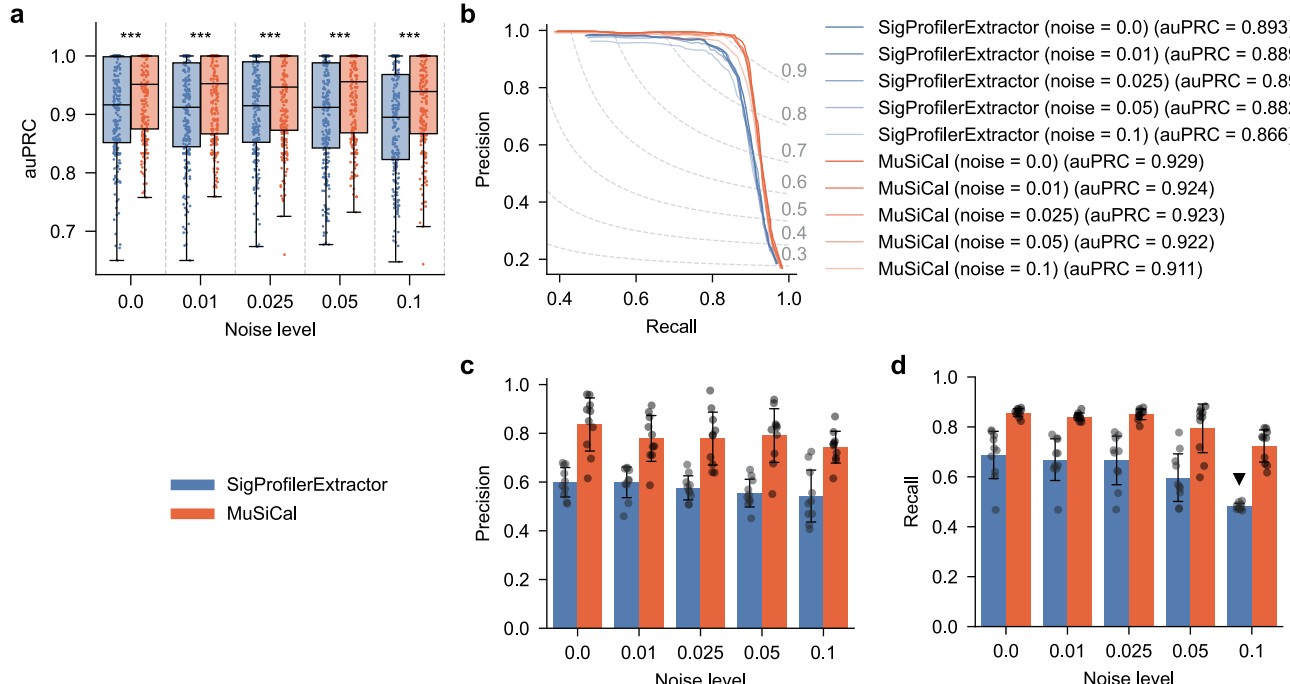

**Extended Data Fig. 3 | MuSiCal outperforms SigProfilerExtractor for de novo signature discovery at different noise levels.** See Methods for details on how different levels of random noise were added. **a.** Area under precision-recall curve (auPRC) for MuSiCal and SigProfilerExtractor at different noise levels. Each box in the box plot represents 250 synthetic datasets (25 tumor types × 10 replicates). auPRC was calculated for each dataset separately, as in Fig. 3b. ***: $p < 0.0005$. $p$-values were calculated with two-sided paired t-tests. Raw $p$-values from left to right: $7.5 \times 10^{-10}$, $1.5 \times 10^{-8}$, $1.7 \times 10^{-6}$, $3.1 \times 10^{-11}$, $5.4 \times 10^{-12}$. **b.** Precision-recall curve (PRC) for MuSiCal and SigProfilerExtractor at different noise levels. Each PRC represents the average result of 250 synthetic datasets (25 tumor types × 10 replicates), as in Fig. 3c. **c.** Precision of MuSiCal and SigProfilerExtractor averaged across all tumor types at different noise levels. Recall was fixed at 0.9. Error bars indicate standard deviation over 10 replicates. **d.** Recall of MuSiCal and SigProfilerExtractor averaged across all tumor types at different noise levels. Precision was fixed at 0.98, corresponding to a false discovery rate (FDR) of 2%. The black triangle indicates the case where a precision of 0.98 was never achieved and the recall at the highest achieved precision was shown. Error bars indicate standard deviation over 10 replicates.

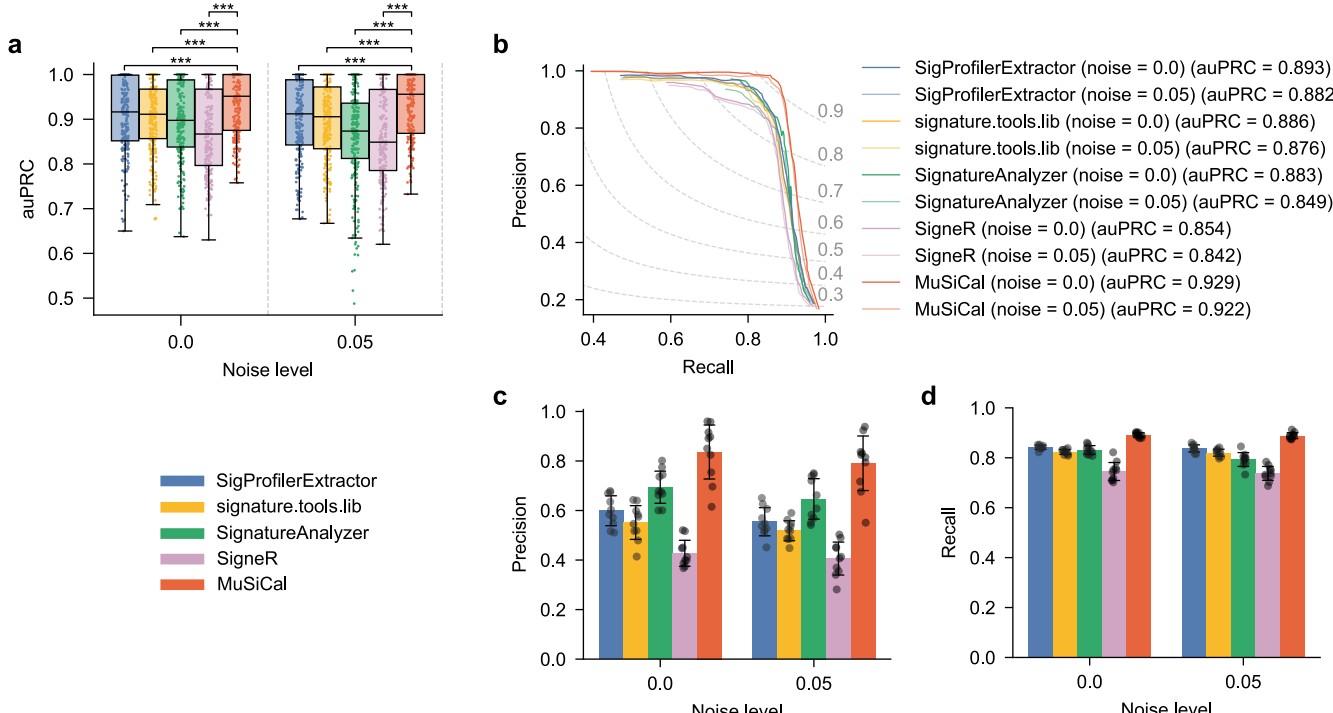

**Extended Data Fig. 4 | MuSiCal outperforms SigProfilerExtractor and three additional existing tools – signature.tools.lib, SignatureAnalyzer, and SigneR – for de novo signature discovery at different noise levels. a**. Area under precision-recall curve (auPRC) for all five tools at different noise levels. Random noise were added as in Extended Data Fig. 3. Each box in the box plot represents 250 synthetic datasets (25 tumor types × 10 replicates). auPRC was calculated for each dataset separately, as in Fig. 3b. ***: $p < 0.0005$. $p$-values were calculated with two-sided paired t-tests. Raw $p$-values from top to bottom at noise level 0.0: $1.2 \times 10^{-26}$, $1.4 \times 10^{-15}$, $1.1 \times 10^{-14}$, $7.5 \times 10^{-10}$. Raw $p$-values from top to bottom at noise level 0.05: $1.7 \times 10^{-25}$, $1.3 \times 10^{-20}$, $5.8 \times 10^{-16}$, $3.1 \times 10^{-11}$. **b**. Precision-

recall curve (PRC) for all five tools at different noise levels. Each PRC represents the average result of 250 synthetic datasets (25 tumor types × 10 replicates), as in Fig. 3c. **c**. Precision of all five tools averaged across all tumor types at different noise levels. Recall was fixed at 0.9. Error bars indicate standard deviation over 10 replicates. **d**. Recall of all five tools averaged across all tumor types at different noise levels. Precision was fixed at 0.9, corresponding to a false discovery rate (FDR) of 10%. Here, precision was fixed at a smaller value compared to Fig. 3e, Supplementary Fig. 5d, and Extended Data Fig. 3d, because a precision of 0.98 was never achieved by SignatureAnalyzer or SigneR in many cases. Error bars indicate standard deviation over 10 replicates.

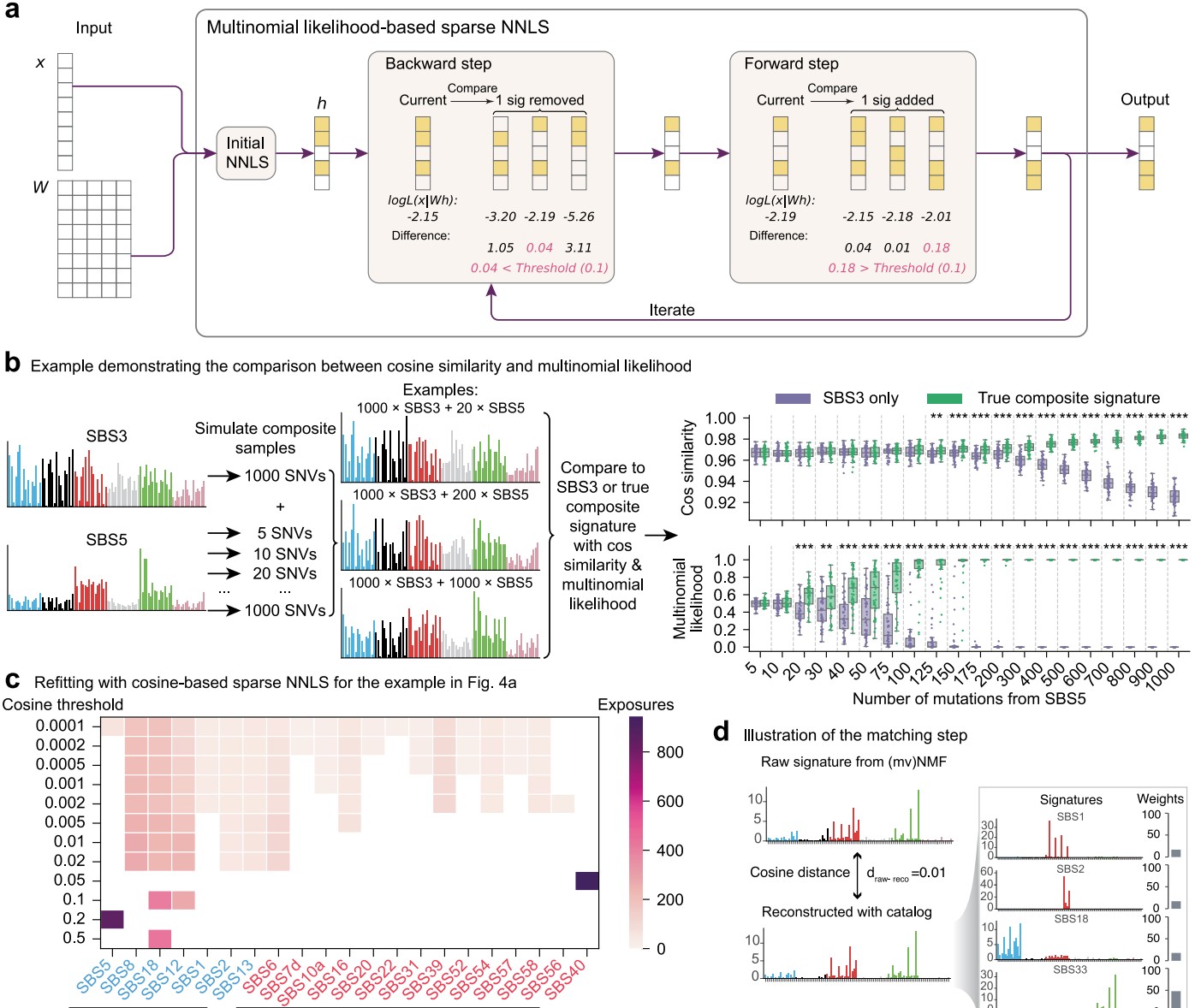

**Extended Data Fig. 5 | Matching and refitting with multinomial likelihood-based sparse NNLS. a**. Diagram of the likelihood-based sparse NNLS algorithm. See Methods for details. **b**. An example demonstrating that multinomial likelihood is more powerful than cosine similarity for separating similar signatures. SBS3 and SBS5 are used to simulate synthetic samples. All samples contain 1000 SNVs contributed by SBS3 as well as varying numbers of SNVs from SBS5. Multinomial likelihood and cosine similarity are then applied to distinguish whether the sample spectra are generated from the correct (SBS3 + SBS5 with appropriate weights) or incorrect (pure SBS3) underlying signatures. The problem is expected to be difficult with few SNVs from SBS5, as the sample spectra will be dominated by SBS3. Indeed, cosine similarity fails to separate the two underlying signatures when there are less than 100 SNVs from SBS5, corresponding to an SBS5 exposure of 100/(100 + 1000) = 9%. By comparison,

multinomial likelihood achieves statistically significant separation down to 20 SNVs from SBS5, corresponding to an SBS5 exposure of 2%. *: $p < 0.05$. **: $p < 0.005$. ***: $p < 0.0005$. $p$-values are calculated with two-sided t-tests. **c**. Same as Fig. 4b, but refitting is performed using cosine similarity combined with the same bidirectional stepwise algorithm as in (a). Notably, there is no threshold with which the set of active signatures is identified correctly. Also, solutions with cosine similarity do not possess the desired property of continuity as in Fig. 4b. For example, even when the threshold is overly small, true signatures (for example, SBS5) can be missed, while when the threshold is overly large, false signatures (for example, SBS40) can be discovered instead of the strongest true signatures. **d**. Illustration of the matching step. MuSiCal uses the same likelihood-based sparse NNLS for the matching step, where a de novo signature is decomposed as a non-negative mixture of known signatures in the catalog.

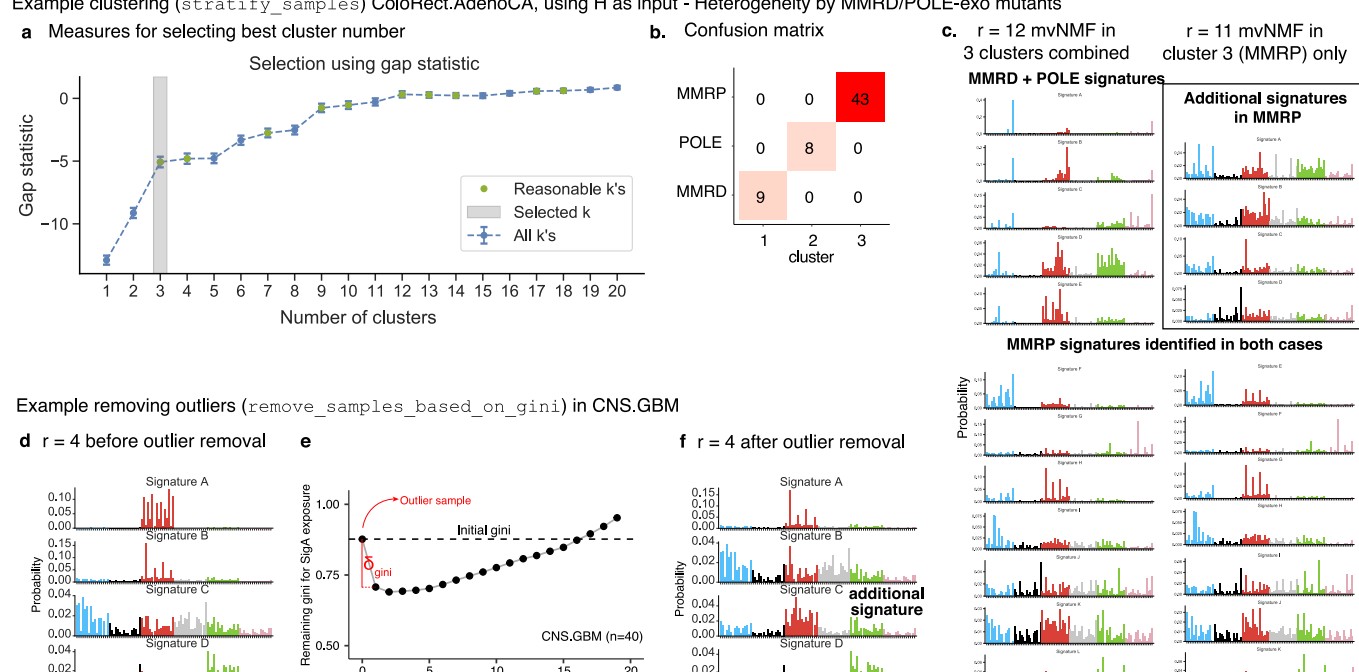

Example clustering (`stratify_samples`) ColoRect.AdenoCA, using H as input - Heterogeneity by MMRD/POLE-exo mutants

**a** Measures for selecting best cluster number

**b.** Confusion matrix

**c.** r = 12 mvNMF in 3 clusters combined / r = 11 mvNMF in cluster 3 (MMRP) only

Example removing outliers (`remove_samples_based_on_gini`) in CNS.GBM

**d** r = 4 before outlier removal

**e**

**f** r = 4 after outlier removal

**Extended Data Fig. 6 | Examples illustrating cohort stratification and outlier removal with the preprocessing module. a-c.** An example with the PCAWG ColoRect.AdenoCA dataset to demonstrate that cohort stratification can benefit de novo signature discovery. After an initial signature discovery with the entire dataset of 60 samples, hierarchical clustering is performed using the de novo exposure matrix $H$, and 3 clusters are selected by the gap statistic, as shown in panel (a). The 3 clusters correspond to samples with MMRP, MMRD, and POLE mutations, respectively, as shown in panel (b). In panel (c), de novo signatures discovered from the entire dataset of 60 samples and the MMRP cluster (43 samples) are plotted, demonstrating improved sensitivity of signature discovery after cohort stratification. Specifically, 12 signatures are discovered when de novo discovery is performed with the entire dataset of 60 samples, while 11 are discovered with the MMRP cluster alone. Out of the 11 MMRP-specific signatures, only 7 are discovered before cohort stratification. Therefore, a separate run of de novo discovery with the MMRP cluster allows 4 more MMRP-specific

signatures to be discovered. In panel (a), reasonable $k$ means any $k$ satisfying $Gap(k) \geq Gap(k+1) - s_{k+1}$, where $Gap(k)$ denotes the gap statistic (indicated by dots in the plot) and $s_k$ the standard deviation of 50 independent simulations after accounting for simulation errors (indicated by error bars in the plot). The smallest reasonable $k$ is chosen as the optimal $k$. See[60,61] for more details. **d-f.** An example with the PCAWG CNS.GBM dataset to demonstrate that outlier removal can benefit de novo signature discovery. When signature discovery is performed with the entire dataset, 4 de novo signatures are discovered, as shown in panel (d), where Signature A corresponds to SBS11 associated with temozolomide treatment. When the de novo exposures are inspected with the Gini coefficient, a single outlier sample with an exceptionally strong exposure of Signature A is detected, as shown in panel (e). After removing this outlier and rerunning signature discovery with the remaining samples, an additional signature is discovered, as shown in panel (f), demonstrating improved sensitivity.

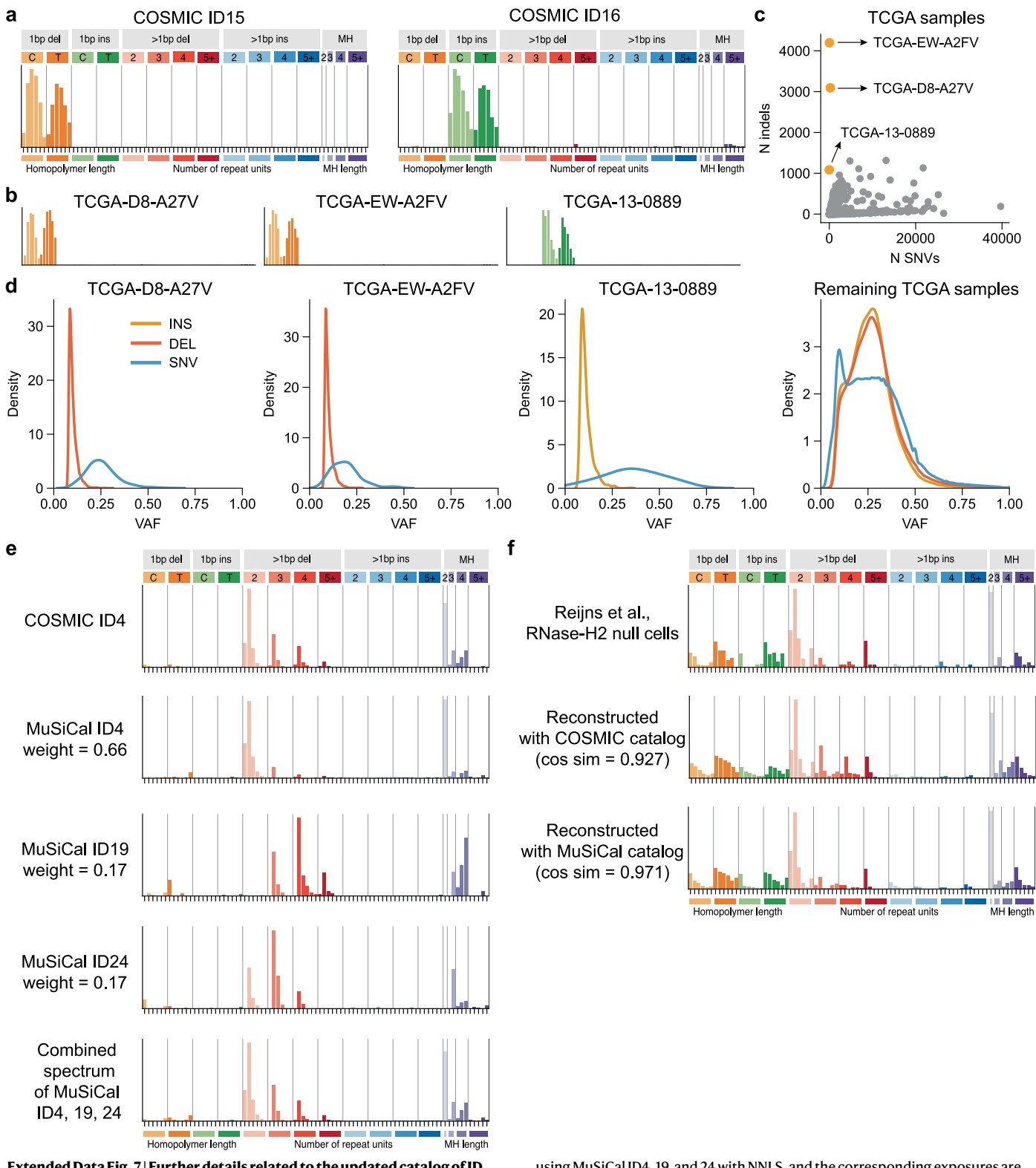

**Extended Data Fig. 7 | Further details related to the updated catalog of ID signatures. a**. Spectra of COSMIC ID15 and 16. These two signatures are not discovered from our PCAWG reanalysis with MuSiCal. **b**. Indel spectra of the three TCGA whole-exome sequenced samples from which COSMIC ID15 and 16 are discovered by the PCAWG consortium[5]. **c**. Number of indels vs. number of SBSs for all TCGA samples, highlighting that these three samples have exceptionally high indel counts but low SBS counts. **d**. Variant allele frequency (VAF) distributions of indels and SBSs for these three samples are compared to those for the other TCGA samples. Indels in these three samples have particularly low VAF, suggesting that they are likely artifactual. **e**. COSMIC ID4 is resolved into multiple signatures in our updated catalog. COSMIC ID4 is decomposed

using MuSiCal ID4, 19, and 24 with NNLS, and the corresponding exposures are annotated next to each of the MuSiCal signatures. The reconstructed signature has cosine similarity of 0.996 with COSMIC ID4 and is shown at the very bottom. **f**. The TOP1-associated ID spectrum observed in RNase-H2-null cells from[42] (top) is compared to the reconstructed spectra using the COSMIC (middle) and the MuSiCal catalog (bottom). The MuSiCal catalog better reconstructs the experimentally derived TOP1 signature. The TOP1 signature is more similar to MuSiCal ID4 (cosine similarity = 0.87) than COSMIC ID4 (cosine similarity = 0.83). Specifically, COSMIC ID4 contains longer (3- and 4-bp) deletions that are not observed in the TOP1 signature.

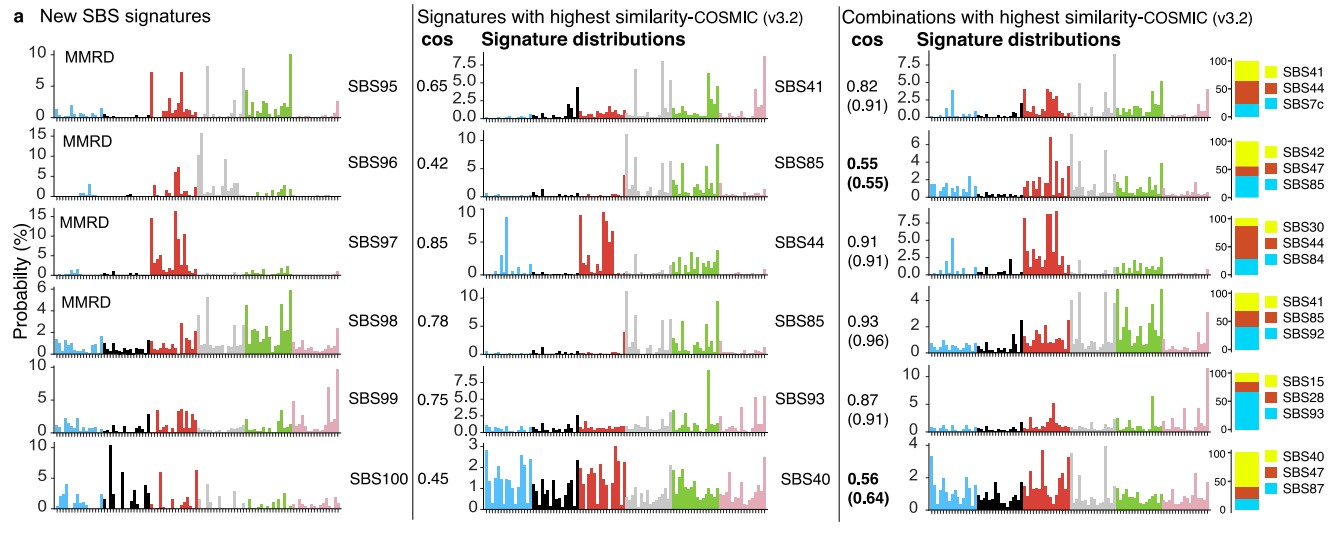

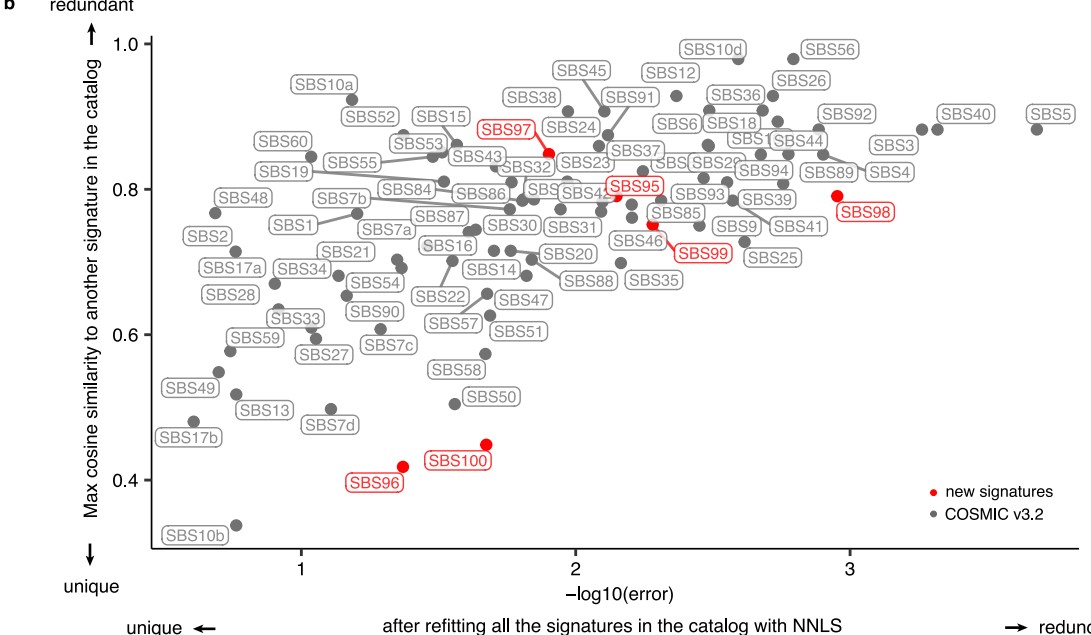

**Extended Data Fig. 8 | MuSiCal discovers 6 new SBS signatures from the PCAWG dataset. a.** Spectra of the 6 new SBS signatures and comparison to the current COSMIC catalog. The 6 new SBS signatures discovered by MuSiCal are plotted to the left. In the middle, signatures from the COSMIC catalog with the highest cosine similarities to the 6 new ones, respectively, are plotted, and the corresponding cosine similarities are annotated next to the spectra. To the right, each of the 6 new SBS signatures is matched to the COSMIC catalog as a combination of at most 3 signatures through NNLS, and the reconstructed signature with the highest cosine similarity is plotted, with the corresponding cosine similarities annotated. The cosine similarities inside the parentheses are obtained from matching the new SBS signatures to the entire COSMIC catalog through NNLS without constraining to at most 3 signatures. Two new signatures, SBS96 and 100, are especially poorly reconstructed by COSMIC signatures and thus highlighted. **b.** Comparison of the 6 new signatures with those from the COSMIC catalog in terms of their uniqueness. To put the cosine similarities in (a) in context, two statistics are calculated for each signature in the COSMIC catalog as well as for the new ones – the maximum cosine similarity to another signature in the COSMIC catalog (y-axis), and the residual error after matching to all other signatures in the COSMIC catalog through NNLS (x-axis). Indeed, the 6 new signatures discovered by MuSiCal are not overly redundant compared to signatures already present in the COSMIC catalog. Two new signatures, SBS96 and 100, are especially unique compared to the majority of COSMIC signatures.

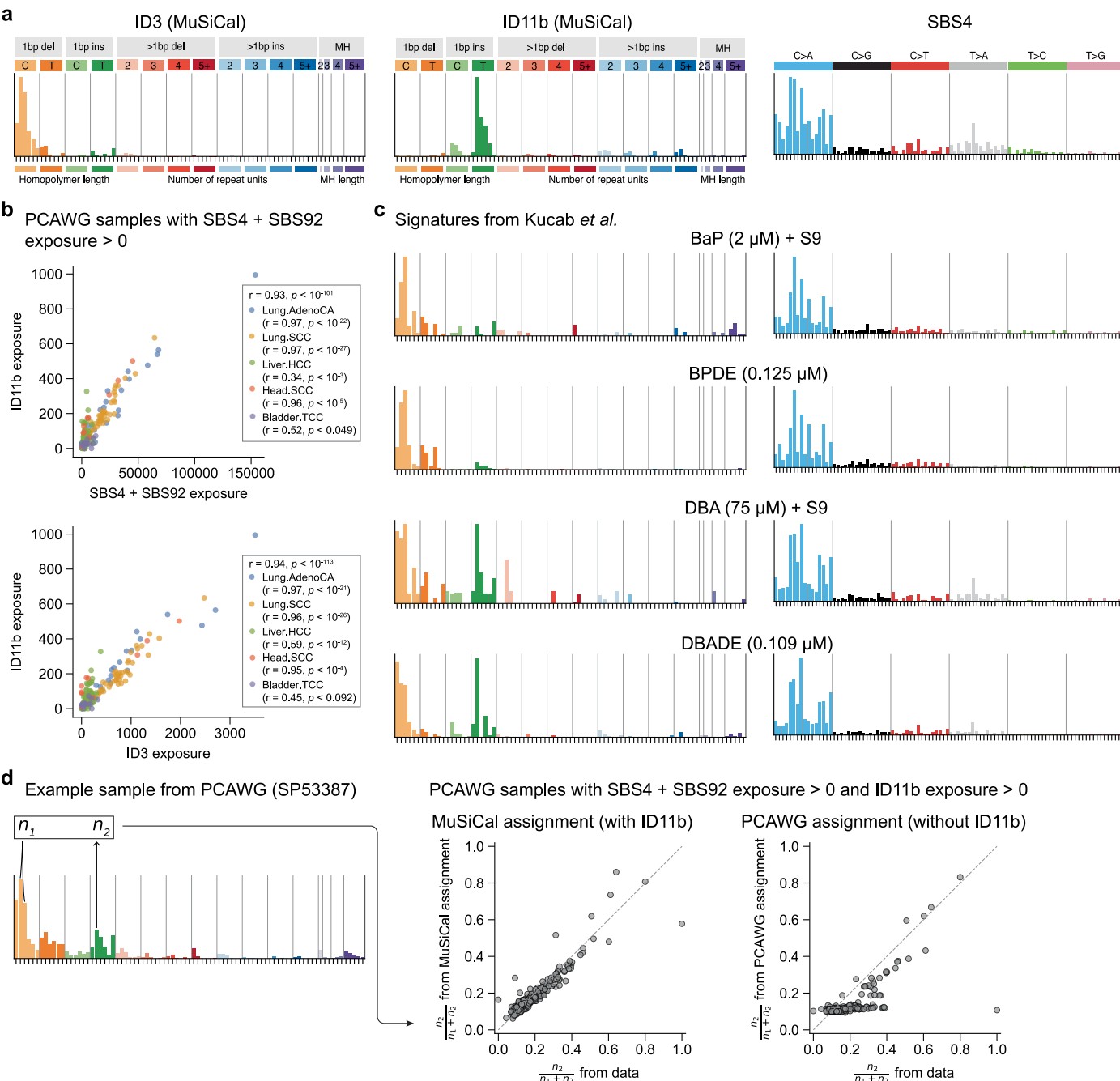

**Extended Data Fig. 9 | ID11b is potentially associated with tobacco smoking.**
**a.** Spectra of signatures known to be associated with tobacco smoking (ID3 and SBS4) and ID11b. SBS92 (not shown) is a recently discovered signature also associated with tobacco smoking[17]. **b.** ID11b exposure is correlated with both SBS4 + SBS92 and ID3 exposures. PCAWG samples related to tobacco smoking, as indicated by nonzero exposures of SBS4 + SBS92, are selected. The per-sample exposures are then plotted for ID11b vs. SBS4 + SBS92, and ID11b vs. ID3. Pearson correlation coefficients and the corresponding p-values are annotated. **c.** ID (left) and SBS (right) signatures observed in cells exposed to different chemicals present in tobacco smoke. Mutation data is obtained from[7]. Background ID and

SBS signatures averaged from control clones are removed. **d.** ID11b is required to explain the observed abundance of T insertions in PCAWG samples related to tobacco smoking. $n_1$ is defined as the number of C deletions at CC or CCC, characteristic of ID3, and $n_2$ is defined as the number of T insertions following a T, characteristic of ID11b (left). The ratio $\frac{n_2}{n_1+n_2}$ is then plotted for reconstructed spectra with MuSiCal assignment vs. observed spectra (middle), as well as for reconstructed spectra with PCAWG assignment vs. observed spectra (right). Without ID11b, the PCAWG assignment fails to explain the observed T insertions following a T.

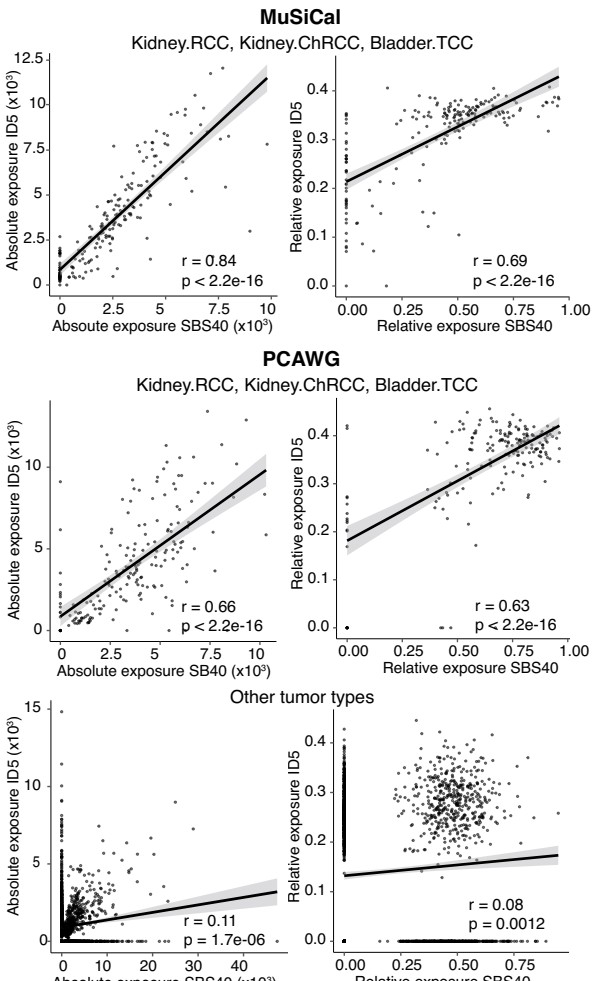

**Extended Data Fig. 10 | Association between SBS40 and ID5.** Exposures of SBS40 and ID5 are correlated for both MuSiCal- (top) and PCAWG-derived (middle) signature assignments in kidney and bladder cancers, which are the only tumor types where SBS40 is assigned by MuSiCal. PCAWG also assigns SBS40 in many other tumor types (bottom), where a strong correlation is not observed. Pearson correlation coefficients are shown for both absolute (left) and relative (right) exposures (normalized by mutation burden). Error bands (shaded areas) indicate 95% confidence intervals of the linear regressions.

# Reporting Summary

## Statistics

For all statistical analyses, confirm that the following items are present in the figure legend, table legend, main text, or Methods section.

| n/a | Confirmed | |
|---|---|---|
| ☐ | ☒ | The exact sample size (*n*) for each experimental group/condition, given as a discrete number and unit of measurement |
| ☐ | ☒ | A statement on whether measurements were taken from distinct samples or whether the same sample was measured repeatedly |
| ☐ | ☒ | The statistical test(s) used AND whether they are one- or two-sided *Only common tests should be described solely by name; describe more complex techniques in the Methods section.* |
| ☐ | ☒ | A description of all covariates tested |
| ☐ | ☒ | A description of any assumptions or corrections, such as tests of normality and adjustment for multiple comparisons |
| ☐ | ☒ | A full description of the statistical parameters including central tendency (e.g. means) or other basic estimates (e.g. regression coefficient) AND variation (e.g. standard deviation) or associated estimates of uncertainty (e.g. confidence intervals) |
| ☐ | ☒ | For null hypothesis testing, the test statistic (e.g. *F*, *t*, *r*) with confidence intervals, effect sizes, degrees of freedom and *P* value noted *Give P values as exact values whenever suitable.* |
| ☒ | ☐ | For Bayesian analysis, information on the choice of priors and Markov chain Monte Carlo settings |
| ☐ | ☒ | For hierarchical and complex designs, identification of the appropriate level for tests and full reporting of outcomes |
| ☐ | ☒ | Estimates of effect sizes (e.g. Cohen's *d*, Pearson's *r*), indicating how they were calculated |

*Our web collection on statistics for biologists contains articles on many of the points above.*

## Software and code

Policy information about availability of computer code

| Data collection | No software was used for data collection. |
|---|---|
| Data analysis | MuSiCal is implemented in Python and available at https://github.com/parklab/MuSiCal. SigProfilerExtractor (version 1.1.3) was downloaded from https://github.com/AlexandrovLab/SigProfilerExtractor. sigLASSO (version 1.1) was downloaded from https://github.com/gersteinlab/siglasso. signature.tools.lib (version 2.1.2) was downloaded from https://github.com/Nik-Zainal-Group/signature.tools.lib. SignatureAnalyzer (version 0.0.7) was downloaded from https://github.com/getzlab/SignatureAnalyzer. SigneR (version 1.22.0) was obtained from https://www.bioconductor.org/packages/release/bioc/html/signeR.html. Custom scripts and analysis notebooks for reproducing results in the paper are available on Zenodo: https://doi.org/10.5281/zenodo.10291569. |

For manuscripts utilizing custom algorithms or software that are central to the research but not yet described in published literature, software must be made available to editors and reviewers. We strongly encourage code deposition in a community repository (e.g. GitHub). See the Nature Portfolio guidelines for submitting code & software for further information.

## Data

Policy information about availability of data

All manuscripts must include a data availability statement. This statement should provide the following information, where applicable:
- Accession codes, unique identifiers, or web links for publicly available datasets
- A description of any restrictions on data availability
- For clinical datasets or third party data, please ensure that the statement adheres to our policy

> PCAWG data, including mutation count matrices of SBSs and IDs, as well as exposure matrices, were downloaded from https://www.synapse.org/#!
> Synapse:syn11726601/files/. Additional PCAWG data were downloaded from https://dcc.icgc.org/releases/PCAWG. All PCAWG data used in this paper are in the
> open tier. COSMIC signatures were downloaded from https://cancer.sanger.ac.uk/signatures/. Source data, including MuSiCal-derived signature catalog from the
> PCAWG reanalysis, are provided in Supplementary Tables 1-4.

## Research involving human participants, their data, or biological material

Policy information about studies with human participants or human data. See also policy information about sex, gender (identity/presentation), and sexual orientation and race, ethnicity and racism.

| | |
|---|---|
| Reporting on sex and gender | *Use the terms sex (biological attribute) and gender (shaped by social and cultural circumstances) carefully in order to avoid confusing both terms. Indicate if findings apply to only one sex or gender; describe whether sex and gender were considered in study design; whether sex and/or gender was determined based on self-reporting or assigned and methods used. Provide in the source data disaggregated sex and gender data, where this information has been collected, and if consent has been obtained for sharing of individual-level data; provide overall numbers in this Reporting Summary. Please state if this information has not been collected. Report sex- and gender-based analyses where performed, justify reasons for lack of sex- and gender-based analysis.* |
| Reporting on race, ethnicity, or other socially relevant groupings | *Please specify the socially constructed or socially relevant categorization variable(s) used in your manuscript and explain why they were used. Please note that such variables should not be used as proxies for other socially constructed/relevant variables (for example, race or ethnicity should not be used as a proxy for socioeconomic status). Provide clear definitions of the relevant terms used, how they were provided (by the participants/respondents, the researchers, or third parties), and the method(s) used to classify people into the different categories (e.g. self-report, census or administrative data, social media data, etc.) Please provide details about how you controlled for confounding variables in your analyses.* |
| Population characteristics | Participants were recruited in each individual ICGC study and in TCGA. All these studies complied with the required ethical guidelines. |
| Recruitment | See above. |
| Ethics oversight | See publications of the corresponding ICGC, TCGA, and PCAWG projects. |

Note that full information on the approval of the study protocol must also be provided in the manuscript.

# Field-specific reporting

Please select the one below that is the best fit for your research. If you are not sure, read the appropriate sections before making your selection.

☒ Life sciences   ☐ Behavioural & social sciences   ☐ Ecological, evolutionary & environmental sciences

For a reference copy of the document with all sections, see nature.com/documents/nr-reporting-summary-flat.pdf

# Life sciences study design

All studies must disclose on these points even when the disclosure is negative.

| | |
|---|---|
| Sample size | This study was designed to be a retrospective analysis of previously published data. No statistical method was used to predetermine sample size. |
| Data exclusions | We only considered sequencing data passing the Quality & Control criteria established by the Pan-Cancer Analysis of Whole Genomes (PCAWG) project. The quality control of this dataset is discussed in their corresponding publication. |
| Replication | No replication was performed since the study was a retrospective analysis of existing datasets. |
| Randomization | No randomization was performed since the study was observational. |
| Blinding | The Investigators were not blinded to allocation during experiments and outcome assessment since the study was observational. |

# Reporting for specific materials, systems and methods

We require information from authors about some types of materials, experimental systems and methods used in many studies. Here, indicate whether each material, system or method listed is relevant to your study. If you are not sure if a list item applies to your research, read the appropriate section before selecting a response.

| Materials & experimental systems | Methods |
|---|---|

**Materials & experimental systems**

| n/a | Involved in the study |
|---|---|
| ☒ | Antibodies |
| ☒ | Eukaryotic cell lines |
| ☒ | Palaeontology and archaeology |
| ☒ | Animals and other organisms |
| ☒ | Clinical data |
| ☒ | Dual use research of concern |
| ☒ | Plants |

**Methods**

| n/a | Involved in the study |
|---|---|
| ☒ | ChIP-seq |
| ☒ | Flow cytometry |
| ☒ | MRI-based neuroimaging |

