## [Peer Review File · Nature Genetics]

Peer Review Information

Manuscript Title: Accurate and sensitive mutational signature analysis with MuSiCal

Corresponding author name(s): Professor Peter Park

Reviewer Comments & Decisions:

Decision Letter, initial version:

1st Jun 2022

Dear Peter,

Your Technical Report entitled "Accurate and sensitive mutational signature analysis with MuSiCal" has now been seen by 3 referees, whose comments are attached. While they find your work of potential interest, they have raised serious concerns which in our view are sufficiently important that they preclude publication of the work in Nature Genetics, at least in its present form.

While the referees find your work of some interest, they raise concerns about the strength of the novel conclusions that can be drawn at this stage.

Briefly, while the reviewers have all provided detailed and thoughtful reports. All appreciate the technical/mathematical innovations presented in MuSiCal, they appear - to varying degrees - unconvinced of the advance provided and your study's suitability for Nature Genetics.

Reviewer #2 is the most positive, and supportive; they make a number of useful requests for further improvement.

Reviewer #1, on the other hand, thinks that the advance needed in the field would be integrating other clinico-genomic data into mutational signature analysis.

Reviewer #3 is downright skeptical, saying that it is extremely difficult to judge performance in the absence of a ground truth. They also provide some thoughtful suggestions for improvement.

In our reading of these reviews, we think that a revision would need to persuade Reviewers #2 and #3 of that advance for an eventual publication. Reviewer #3's fundamental criticism seems, to our eyes, hard to address with specific further analysis. Reviewer #2's request for the clinico-genomic data integration is a very thoughtful one, but it was unclear to us whether this is even possible in the current framework of MuSiCal. Thus, we have decided that the best decision is this rejection with the

option for an invited appeal if you think you can address these major concerns - it may well be that there is a more rapid path to publication elsewhere.

Should further experimental data allow you to fully address these criticisms we would be willing to consider an appeal of our decision (unless, of course, something similar has by then been accepted at Nature Genetics or appeared elsewhere). This includes submission or publication of a portion of this work someplace else.

The required new experiments and data include, but are not limited to those detailed here. We hope you understand that until we have read the revised manuscript in its entirety we cannot promise that it will be sent back for peer review.

If you are interested in attempting to revise this manuscript for submission to Nature Genetics in the future, please contact me to discuss a potential appeal. Otherwise, we hope that you find our referees' comments helpful when preparing your manuscript for resubmission elsewhere.

Sincerely,

Michael Fletcher, PhD
Senior Editor, Nature Genetics

ORCID: 0000-0003-1589-7087

Referee expertise: all three work in the field of computational cancer genomics and have mutational signature analysis experience.

Reviewers' Comments:

Reviewer #1:

Remarks to the Author:

Mutational signatures analysis is emerging as essential in different aspects of cancer genomics and translational research. In fact, certain mutational signatures have been linked to sensitivity to distinct treatments, and others have been associated with poor outcomes. Furthermore, mutational signatures are an essential piece for understanding the cancer pathogenesis and which mutational processes are involved in shaping the cancer genomic landscape over time. In this manuscript, the Authors develop a new computational tool aimed to improve the accuracy and sensitivity. Overall, the Authors show high expertise in mutational signatures analysis and clearly know the most important and still unsolved issues. I particularly appreciate the analysis focused on SBS40, a new flat signature often over-assigned, and without any clear differential etiology from the canonical clock-like SBS5. Despite the manuscript being well-written and the robust methods, I have some concerns about the overall accuracy of their tool.

Here are my major concerns and suggestions:

The assignment of flat signatures is definitely one of the major issues that every mutational tool has

to face. I agree that the introduction of new flat signatures in the latest COSMIC reference (e.g., SBS40) further complicates this mathematical assignment. Different groups have tried to solve this problem by developing new computational and mathematical tools. However, because several of these signatures are well defined and linked to distinct clinical and biological features, I believe that rather than improving the mathematical accuracy, we should try to define the presence/absence and the quantification of these signatures leveraging different and independent features (e.g. clinical history, presence of mutations, SV and indels signatures, etc). For example, as pointed out by the Authors, SBS3 is often extracted as a false positive due to its flat nature and similarities with SBS8, SBS40, and SBS5. Rather than working on better mathematical models, wouldn't be more appropriate and accurate to develop an orthogonal approach using SV and ID signatures to define if SBS3 is really present (i.e., HRD-detect)?

The Authors used the PCAWG catalog as the source for simulating genomes 96-profiles. While this approach is established, I wonder how similar the in-silico genomes are compared to the actual PCAWG tumors. Often the real tumor 96-profile is noisy, and some signatures might be active with slightly different shapes. Furthermore, a tumor type might have a certain number of signatures, but these might not be active altogether all the time. Serena Nik-Zainal lab has recently proposed the idea of tissue-specific signatures (Degasperi et al. Nat Cancer 2020), where the same signature might have slightly different shapes according to the tumor type and location. Have the Authors checked the cosine similarities of their simulated genomes with the real genome from the PCAWG? Or have the Authors included some noise (e.g. random mutations) to see how this affects the performance and the rate of false-positive and -negative?

As additional ground truth on which to test the tool's performances, the Authors could pick real tumors from PCAWG (and/or other datasets) and check their accuracy in defining signatures that are expected to be present. For example, they can test how many patients with BRCA-mutated breast cancer with high HRD-detect have a significant presence of SBS3. Alternatively, the Authors can check the contribution of temozolomide and platinum chemotherapy signatures in patients previously exposed to these chemotherapy agents.

Since 2020 SigProfiler has been the gold standard for mutational signatures analysis. Compared to the original NMF, it is built around three steps: 1) first it runs a de novo extraction, 2) then it matches the new extracted signatures with the latest COSMIC reference, and 3) it applies a fitting approach to correctly quantify their contribution. Across these different steps, it also provides alternative solutions. I agree that in the setting of in silico WGS SigProfiler tends to provide a slightly worse de novo extraction compared to MuSiCal. However, at the end of the final fitting step, it can still provide an accurate estimation of the right signatures, excluding the ones that are not relevant/active. Have the Authors compared the final PCAWG results with their MuSiCal re-analysis?

The MuSiCal F1 is significantly higher in 8 tumors out of 25 (Figure 3b). Most of these are known to be hypermutated, with one or two dominant signatures "covering" the others. This is a known problem in the de novo extraction performances, in particular using NMF. In this setting, MuSiCal seems to work better. Have the Authors explored if the better MuSiCal performance is related to the tumor with a high mutational burden?

I agree that manual adjustment for defining the right number of signatures is not optimal. However, a prior knowledge of which signatures we might expect in a certain tumor is definitely important and would reduce false positives. For example, in contrast to PCAWG, the Authors extracted SBS31 from Glioblastoma WGS. This signature is known to be caused by platinum exposure, so I wonder if the Authors verified if these cases had ever been exposed to platinum-containing regimens before the sample collection. The same platinum-induced signature seems to be extracted in MPN and AML. In both tumors, platinum is never used, and as far as I know, all samples were collected at baseline. Is this a sign that the high sensitivity also increases the number of false positives (reduce specificity)?

Another example of a possible false positive call is related to SBS84. This mutational signature is known to be caused by somatic hypermutation in normal and transformed post germinal center B-cells. Nevertheless, SBS84 activity is reported in bone benign tumors and in different myeloid tumors (MDS, MPN, and AML). This data is against the intrinsic biology of these tumors, known to be completely independent of somatic hypermutation. Similarly, SBS9 (poly-eta in post germinal center tumors) is extracted among MPN and AML.

I appreciate that the Authors introduced a fitting part in their workflow. However, it seems that their tool is not so different from SigProfiler fitting tool. Which are the key differences? Both seem to be based on cosine similarities and reconstruction after a leave one out of each signature.

Overall, I believe that the community needs a more standardized and reproducible mutational signature tool. However, my impression after reading this study is that mathematical unbiased modelings have reached the limit of their capability in mutational signature extraction. To make MuSiCal more innovative and accurate compared to the existing ones, the Authors might consider integrating their models with additional and established features (e.g. known signatures, clinical history, presence of distinct gene mutations). I believe that this will compensate for the rate of false-positive, further increasing the model accuracy.

Reviewer #2:

Remarks to the Author:

In the manuscript "Accurate and sensitive mutational signature analysis with MuSiCal", Jin and colleagues present a set of novel algorithms for the detection, assignment and validation of mutational signatures. The work contains three major novelties that aim to solve problems persisting in standard mutational signature calling: 1) The application of Minimum-volume NMF, a method from signal processing, that guarantees unique solutions. 2) A likelihood-based sparse NNLS method for assigning signatures to individual samples that aims to fix the over-assignment of low-exposure signatures. 3) A data-driven simulation approach to validate the detection and assignment of signatures. After explaining and validating the different parts of the model, the new workflow is applied to over 2,700 samples from the Pan Cancer Analysis of Whole Genomes to retrieve new signatures and alleviate some problems with the common approach including the over-representation of flat signatures.

The authors address a highly timely and important subject that can provide a substantial advance in mutational signature analysis over the current state-of-the art. The manuscript combines multiple interesting concepts that pose a substantial improvement to the field of mutational signatures. The concepts are well justified and thoroughly validated and the paper was well written. However, I'm having some issues with the article in its current form, in particular with respect to the comparison to the current state-of-the-art model, that I believe should be addressed before publication.

Major points

Pre processing

1. In the graphical abstract (Fig 1), the preprocessing module is presented as one of the four fundamental modules of MuSiCal. However, it is only briefly mentioned in the main text (Section 2.5). Since I assume that the preprocessing step affects the downstream analyses in a non-trivial way, this step and its effects should be explained in more detail, not only in the online methods. In particular, preprocessing seems to be an important part of the re-analysis of the PCAWG data. Also, in Ext Fig 10a: What does "reasonable k's" mean?

2. For the re-analysis of PCAWG in section "Implementation of preprocessing" the authors write "Next, we stratified samples into subsets with distinct signature compositions with hierarchical clustering." Here, the final number of subsets per cancer type should be reported (best to give the mean and range). Furthermore, the range of removed samples should be reported (not just the average).

Showing improvements from PCAWG reanalysis in Ext. Data Fig. 2

3. Extended Data Fig. 2 shows the problems with the current COSMIC signatures quite well. It would be good to have a before/after comparison with the de novo detected signatures from Section 2.5.

Simulation studies

4. The simulation studies detailed in Methods 4.3 are an integral part of this paper and in support of the authors' claim that their method outperforms both standard NMF and SigProfileExtractor. The source code for the simulation studies should be available along with the code for MuSiCal so that the outcome can be validated independently.

5. Furthermore, from my understanding, the current simulation studies does not include randomly generated noise but only a random combination of the true signatures. As real biological data includes a variety of noise it is important to check the effect of noise on the performance of MuSiCal and SigProfileExtractor, both uniformly random noise as well as spurious signatures that are spiked in, in addition to the "true" signatures. I would suspect that this would have a measurable effect, for example for the faint detection of some signatures seen e.g. in Fig. 3a. I therefore recommended that the comparison be repeated with varying strength of noise and the robustness of MuSiCal to random noise reported.

Uniqueness of the mvNMF solutions

6. Theoretically, mvNMF should be able to find a unique solution to the NMF problem "if the exposure matrix is sparse enough" (p.3). The authors argue that the sparsity assumption is true for most tumor data based on the sparseness for the PCAWG dataset. However, looking for example at Fig 2c, we see that the cosine error for mvNMF has a similar spread as the one for standard NMF. This holds true for almost all samples (in some cases even larger, e.g. SBS5). If the resulting signatures are supposed to

be unique, how is the wide spread in cosine error explained?

7. Due to the claim of uniqueness for mvNMF I would like to see a comparison between standard NMF and mvNMF w.r.t. the actual volume spanned by both solutions in the mutational space. This should be performed not on simulated data but on the PCAWG data. When grouping the results by tumor type, the results should correspond to the sparsity of the exposures as seen in Ext Fig 4c.

Reconstruction error

8. How does mvNMF perform w.r.t. reconstruction errors of the original matrix in comparison to NMF?

De-novo signature detection in comparison to SigProfileExtractor

9. The poor performance of SigProfilerExtractor is quite surprising. In Fig. 3, SigProfilerExtractor only detects three de-novo signatures which to me seems like some parameter was chosen incorrectly (or rather the default parameters were a poor choice). Due to the poor performance of SigProfilerExtractor, I would strongly advise that more competing frameworks are compared here. In the SigProfilerExtractor paper, they compared their framework to seven other ones: SigProfiler_PCAWG, SignatureAnalyzer, SigneR, MutationalPatterns, MutSpec, SomaticSignatures, and SignatureTools. At least some more tools should be compared here.

10. The authors adjust the detection threshold to get the optimal F1 value for both methods compared. However, this procedure is not representative of the actual use case. If model A performs better than model B when both are perfectly tuned does not mean that model A is also better than model B when the hyperparameters are tuned in absence of the ground truth. Why isn't MuSiCal's own parameter tuning used here?

11. The metric used in the comparison (precision, recall and the F-score) might be less than ideal for the problem at hand. Recall is a measure for "completeness" of the results but when COSMIC signatures are present with as low as 0.034 (SBS2 in Fig. 3a) one can hardly speak about "recovering SBS2". As the Precision does almost not change for any of the cancer types (see Fig 3b) all of the F1 gain likely comes from the Recall gains. In this context it seems that the main advantage of MuSiCal over SigProfileExtractor is that it recovers more signatures and therefore has a higher chance of recovering COSMIC signatures at a low percentage. Also this comparison should be repeated where SigProfileExtractor is forced to have the same number of signatures as MuSiCal to see that MuSiCal is better in general and not just better at choosing the number of signatures.

12. Some of the refitting results seem strange. For example the MuSiCal Signature Sig 7 looks very much like SBS17b but is actually assigned a higher similarity to SBS40. According to Fig 3b, Skin-Melanoma has the highest difference between MuSiCal and SigProfilerExtractor. It would be interesting to see Fig 3a recreated for all cancer types in which MuSiCal outperforms SigProfilerExtractor.

13. It seems strange to me that for some cancers, the F1 value has zero variance for both MuSiCal and SigProfilerExtractor. Here it would also be interesting to see the Fig 3a-like plot for these examples.

Computation time

14. In the discussion it is mentioned that "the scalability of MuSiCal could be improved." However, it is not stated in the manuscript what computational costs are expected. Therefore, the time and memory cost for all substeps should be included based on the application on the PCAWG data. I myself tried to run MuSiCal's de-novo detection of signatures on a datasets with ~2.500 samples and 40 features on

a high computing cluster with 32 cores (parameters taken from the example notebook of the GitHub repository), which took multiple days to complete.

Benchmarking the likelihood-based sparse NNLS

15. Same issue as above (point 10 in De-novo signature detection in comparison to SigProfileExtractor): Selecting the best threshold does not reflect the real-world usage.

16. Furthermore, the metric of "correct support discovery rate" seems not ideal as it only regards perfect fits. If a model gets 90% correct signatures it is counted the same as a model that gets only 10% correct. This seems wrong to me as for real-life examples we can never really expect 100% correct signature retrieval. As precision and recall was used in the previous section I am unsure why the authors decided not to use it here. Especially because it seems to make more sense than in Fig 3. Since this is a threshold-adjustment problem, a precision-recall curve and the respective area under the curve seems to be a good measure of the performance without prior knowledge about the threshold. Therefore I would like to see the same analysis but using the metric of precision recall. A F1 score for the perfect threshold is ok but then the area under the precision recall curve should at least be included in the supplement to show that MuSiCal is better even without prior knowledge of the perfect threshold. Also since MuSiCal has a built in parameter tuning algorithm, why isn't it used here?

in silico validation

17. How does the (simul - data) score compare to the mere reconstruction error? So for the PCAWG data, how is the reconstruction error and the (simul - data) score related? Also mention what are the problems with using just the reconstruction error.

18. The term "PCAWG" is a bit ambiguous when referring to the assignment (e.g. W^{PCAWG} or in Fig 5b). This got me confused initially as the dataset itself can be referred to as PCAWG and W_{data} as well as W_{simul} are of course also based on the PCAWG data. So maybe rename this to "original assignment".

19. Why is the glioblastoma dataset chosen here? Without an appropriate explanation this seems like cherry-picking. It is fine to use a single tumor type for demonstration purposes but this should be repeated for multiple tumor types.

Preprocessing in the re-analysis of PCAWG

20. In the Methods it is stated that "the suggested stratification needs to be investigated in a case-specific manner in order to maximize the benefit for signature discovery (see below for detailed approach taken for reanalysis of PCAWG data)" but in the PCAWG section it is only described as "we stratified samples into subsets with distinct signature compositions with hierarchical clustering". Does this mean the normal preprocessing step based on the exposure matrix H was used? What did the "case-specific investigation" look like?

21. Furthermore, since the stratification of the input greatly improves the sensitivity of the signature detection I would be interested as to how much of the newly discovered signatures can be attributed to this preprocessing. If this stratification is turned off, does MuSiCal still find the new ID and SBS signatures? If it doesn't, that needs to be made clear in the text.

New signature ID11b

22. The newly discovered signature ID11b is very close to the established ID11 (cosine of 0.96).

ID11b correlates both with SBS4 and ID3 linking it to tobacco smoke. Does this correlation also exist to the standard ID11 signature (i.e. when using the original set of signatures)? If this is the case, then ID11b and its relation to tobacco smoke is not necessarily a new discovery but seems like a false positive. If it is the case then this point should be made in the text, i.e. "through the new stratification of ID11 into ID11a and ID11b, we have discovered a previously unknown etiology". Same for Extended Data Fig. 15: In the original PCAWG assignment, does the inclusion of the original ID11 also explain the observed T insertions after T. One would expect so as both ID11a and ID11b have strong T following T insertions.

Minor points

* Section 2.1: "any cone that encloses the data points" -> I think it has to be "any convex cone" or "any linear cone"

* Fig 1

* missing explanation for purple and dashed/orange arrows

* after reading the paper this visual abstract became quite clear for me but on first glance I didn't understand the four boxes

* Fig 2a:

* state the meaning of the x, y and z axis. I understand this figure is inspired from Ref. 46 but the figure might not be clear for a general audience.

* I assume that the fact that all grey points are along the dashed lines is due to the sparseness of the exposure matrix. If this is the case that should be mentioned in the figure caption.

* The grey circles are not in the legend

* Fig 2b: Why are the triangles not filled like in Fig 1a?

* Fig 2c: Maybe mark the ones where vanilla NMF is better as it is hard to see (and especially hard to gauge how many are better/worse)

* Fig 3a:

* The meaning of the percentages is not explained in the figure caption

* Some entries are too big for their cells in the matrices

* Fig 4a/b: X-axis ticklabels are off-center (shifted to the left)

* Fig 5a

* What are H_a and W_a ? They are not mentioned in the text. I guess it should be W_s and H_s

* This figure is not 100% self-explanatory. For example "r" should be written out as "number of signatures", the orange arrow should say "simulate" and point to X_{simul} , the arrow under "Refit" should be in the same design as all the other arrows, "MusiCal assignment" should maybe be "final assignment"

* the dashed lines have varying alpha

* Fig 5b

* This should state "Signature assignments for glioblastoma (N=?)" to make clear that this is just for a single tumor type

* It is also very unclear for the rest of the Figure (c-f) if all of these results are for glioblastoma only.

* Ext Fig 10a: What does “reasonable k’s” mean?

Reviewer #3:

Remarks to the Author:

Jin et al present a new algorithm for mutational signature learning from cancer genome sequencing data.

Mutational signature analysis is an essential pillar of cancer genomics as it reveals the footprints of the mutational processes that contributed to cancer development. While the concept of deconvolving the patterns of mutations into individuals signatures is simple it is still a procedure that is not fully resolved and many commonly used algorithms suffer from over- and under-fitting. The authors’ contribution to the problem is a statistically principled algorithm that employs a minimal volume variant of the non-negative matrix factorisation (mvNMF) that has been the workhorse for mutational signature analysis in the last 9 years. mvNMF regularises the exposure matrix of the NMF product and can thus increase the stability of inference, as demonstrated in different context. The authors demonstrate the accuracy of their algorithm termed MuSiCal using simulations and apply it to 2,700 genomes from the PCAWG consortium.

I do commend the authors for developing a new, statistically sound inference framework, which is lacking in many other algorithms and which might be a critical step forward. The analysis is also clearly presented and some of the problems of NMF based signature analysis are well explained.

The biggest challenge with mutational signature analysis – and why some elements of it have become controversial – is that there is usually no ground truth. For that reason a great number of algorithms have been developed, each claiming superior performance based on circumstantial evidence.

A similar issue applies also to this study. The authors claim it ‘solves fundamental problems’, ‘resolve long-standing issues with the ambiguous ‘flat’ signatures’ and ‘give insights into signatures with unknown etiologies’. Further ‘We expect MuSiCal and the improved catalog to be a step towards establishing best practices for mutational signature analysis.’ These are all very bold, but ultimately and unfortunately somewhat empty statements in the absence of hard evidence.

As I don’t see that all claims are well supported I’m unfortunately not convinced that Musical presents more than an interesting new mathematical concept.

Claim: ‘solves fundamental problems’.

It is known that NMF has a tendency to overfit. For that reason most signature analyses employ various types of resampling approaches to establish solutions that do not overly depend on individual samples. More work is needed here to demonstrate that their approach genuinely outperforms others. The authors only compare their algorithm to one competing method, even though there is a plethora of them. It is an open secret in the mutational signature field that SigProfiler, when run out of the box, has a number of issues. This (<http://dx.doi.org/10.1038/s43018-020-0027-5>) for example, appears to be one of the better alternatives, but there is also MutationalPatterns, mmsig, MutSignatures, HDP, eMU, and many more. The authors would need to make a more comprehensive comparison to

demonstrate that their algorithm is truly superior.

Claim: 'Our simulation studies demonstrate that MuSiCal outperforms state-of-the-art algorithms for both signature discovery and assignment.'

This may in part be a self-fulfilling prophecy. The authors essentially simulate according to their algorithms' model. But it is fundamentally unclear what the generative process is and it's likely to be way more complex than a matrix product of exposures and signature with a bit of Multinomial sampling on top. Mutagenesis has been shown to be influenced by many factors ranging from local DNA accessibility, histone modifications, repair efficiency and deficiency, and perhaps also mutagen metabolism. All these and many more factors are likely to influence the observed mutation spectra and complicate the analysis of mutational signatures. Understanding how the algorithm copes with a range of signature distortions and spiked-in outliers would be helpful.

Claim: 'give insights into signatures with unknown etiologies'.

Generally, the statements about new biological insights are very weakly supported, lacking experimental evidence. The existing references appear fairly handpicked rather than systematic. A convincing demonstration would be to show that the algorithm reproduces the correct exposure and signature matrixes from experimental systems with known mutagenic exposures (e.g <http://dx.doi.org/10.1016/j.cell.2019.03.001> or <https://doi.org/10.1038/s41467-020-15912-7>). Alternatively, one could show evidence for better correlation of chemotherapeutic exposures (<https://doi.org/10.1038/s41588-019-0525-5>) or DNA repair deficiency conditions (MMRD, POLE/D exonuc variants, BERD (MUTYH, NTHL1), etc).

Further, the authors investigate the correlations between SBS and ID signatures and claim that their method produces better correlations. If a high degree of correlations were the proof, then why do the authors run analysis of different parts of the data?

Claim: 'resolve long-standing issues with the ambiguous 'flat' signatures'.

In the same vein this is only supported by handwaving arguments. A recent publication has attributed the subtle distinction of SBS5 and SBS40 to heterochromatin (<https://doi.org/10.1038/s41467-021-23551-9>).

Claim: 'By reanalyzing over 2,700 cancer genomes, we provide an improved reference catalog of signatures and their assignments'. For their catalogue to be truly an improved reference, the authors should analyse additional data sets, such as data from the Hartwig Medical Foundation, which encompasses another 4000 cancer genomes. It's also worth noting that a recent analysis of an extra 10,000 cancer genomes from Genomics England was published (<http://dx.doi.org/10.1126/science.abl9283>). So if the aim of the manuscript were to define a better reference catalogue, the bar would certainly be higher.

Decision Letter, Appeal – initial version:

2nd Feb 2023

Dear Peter,

Thank you for your message of 2nd Feb 2023, asking us to reconsider our decision on your manuscript "Accurate and sensitive mutational signature analysis with MuSiCal". I have now discussed the points of your letter with my colleagues, and we think that your revision addresses at least some of the major concerns we highlighted from the last round of peer review. We therefore invite you to resubmit your manuscript for a further round of review.

When preparing a revision, please ensure that it fully complies with our editorial requirements for format and style; details can be found in the Guide to Authors on our website (<http://www.nature.com/ng/>).

Please be sure that your manuscript is accompanied by a separate letter detailing the changes you have made and your response to the points raised. At this stage we will need you to upload:

1) a copy of the manuscript in MS Word .docx format.

2) The Editorial Policy Checklist:

<https://www.nature.com/documents/nr-editorial-policy-checklist.pdf>

3) The Reporting Summary:

(Here you can read about the role of the Reporting Summary in reproducible science:

<https://www.nature.com/news/announcement-towards-greater-reproducibility-for-life-sciences-research-in-nature-1.22062>)

Please use the link below to be taken directly to the site and view and revise your manuscript:

[redacted]

With kind wishes,

Michael Fletcher, PhD
Senior Editor, Nature Genetics

ORCID: 0000-0003-1589-7087

Author Rebuttal to Initial comments

Reviewer #1:

Remarks to the Author:

Mutational signatures analysis is emerging as essential in different aspects of cancer genomics and translational research. In fact, certain mutational signatures have been linked to sensitivity to distinct treatments, and others have been associated with poor outcomes. Furthermore, mutational signatures are an essential piece for understanding the cancer pathogenesis and which mutational processes are involved in shaping the cancer genomic landscape over time. In this manuscript, the Authors develop a new computational tool aimed to improve the accuracy and sensitivity. Overall, the Authors show high expertise in mutational signatures analysis and clearly know the most important and still unsolved issues. I particularly appreciate the analysis focused on SBS40, a new flat signature often over-assigned, and without any clear differential etiology from the canonical clock-like SBS5. Despite the manuscript being well-written and the robust methods, I have some concerns about the overall accuracy of their tool.

We are delighted that the Reviewer appreciates the importance of robust signature analysis and the need to make progress on unresolved issues such as dealing with flat signatures. In the detailed responses below, we address the concerns regarding the accuracy of our method.

Here are my major concerns and suggestions:

1. The assignment of flat signatures is definitely one of the major issues that every mutational tool has to face. I agree that the introduction of new flat signatures in the latest COSMIC reference (e.g., SBS40) further complicates this mathematical assignment. Different groups have tried to solve this problem by developing new computational and mathematical tools. However, because several of these signatures are well defined and linked to distinct clinical and biological features, I believe that rather than improving the mathematical accuracy, we should try to define the presence/absence and the quantification of these signatures leveraging different and independent features (e.g. clinical history, presence of mutations, SV and indels signatures, etc). For example, as pointed out by the Authors, SBS3 is often extracted as a false positive due to its flat nature and similarities with SBS8, SBS40, and SBS5. Rather than working on better mathematical models, wouldn't be more appropriate and accurate to develop an orthogonal approach using SV and ID signatures to define if SBS3 is really present (i.e., HRD- detect)?

We agree with the Reviewer that integration of multiple clinical and genomic features is critical for identifying of clinically actionable mutational processes. However, our manuscript is aimed at solving an upstream problem on which subsequent integration analysis is based. We detail the reasons below from two perspectives (**Figure R1**).

Figure R1. MuSiCal lays the foundation of multivariate classification algorithms for detecting clinically actionable mutational processes. Top: MuSiCal improves the accuracy of signature assignments, which in turn could improve the performance of downstream multivariate classifiers integrating multiple genomic and clinical features. Bottom: MuSiCal improves the accuracy and sensitivity of *de novo* signature discovery, thus facilitating the discovery of new clinically relevant mutational signatures.

First, the multivariate classification approaches to detect certain phenotypes (such as HRD and MMRD) rely on accurate signature assignment results. Here, it is critical to distinguish the concept of “signatures” (e.g., HRD-related SBS3) from that of “mutational processes” (e.g., HRD). While determining the presence/absence of a mutational process can benefit from integrating multiple features, this classification task requires the mathematical quantification of each feature in the first place (**Figure R1**). In HRDetect, the input features include estimated exposure levels of HRD-related SBS, indel, and SV signatures. The quality of these inputs thus influences the final prediction accuracy of the classifier. For example, if the SBS3 assignment is more accurate, i.e., HRD samples are assigned with high exposures of SBS3 while HRP samples are not, then this feature will aid the classification task and improve the HRD prediction accuracy. Indeed, MuSiCal-derived SBS3 assignment achieved better sensitivity in detecting HRD than the SBS3 feature that was fed into the HRDetect algorithm (**Figure R2**). This result suggests that the performance of multivariate classifiers such as HRDetect can be further improved if MuSiCal-derived signature assignments are used as input, especially when MuSiCal is applied to other signature types as well (e.g., indel and SV). In a separate study (unpublished), we are extending our previously developed multivariate classification framework SigMA [1] to incorporate MuSiCal-derived signature assignments as well as other genomic/clinical features, exactly as the Reviewer suggested.

Figure R2. MuSiCal-derived SBS3 assignment outperforms the SBS3 feature used in HRDetect for detecting HRD. We compared the performance of SBS3 assignments derived from NNLS,

signature.tools.lib, and MuSiCal for detecting HRD in breast, ovary, pancreas, and prostate tumors. The signature.tools.lib-derived SBS3 assignment was used as input to HRDetect in [2] (Degasperi et al. Nat Cancer 2020). The HRDetect final classification was considered as true labels. Sensitivity was shown at FPR values 5% and 10%.

Second, apart from producing more accurate signature assignments, MuSiCal also improves the accuracy and sensitivity of *de novo* signature discovery. With MuSiCal being applied to broader datasets in the future, we expect new and more accurate signatures that are clinically actionable to be discovered (**Figure R1**). As an example, in our manuscript we discovered a new MMRD-associated indel signature ID26 through PCAWG reanalysis with MuSiCal (Figure 6a, c). This new signature, when incorporated into a multivariate classification algorithm, may help improve MMRD detection accuracy.

2. The Authors used the PCAWG catalog as the source for simulating genomes 96-profiles. While this approach is established, I wonder how similar the in-silico genomes are compared to the actual PCAWG tumors. Often the real tumor 96-profile is noisy, and some signatures might be active with slightly different shapes. Furthermore, a tumor type might have a certain number of signatures, but these might not be active altogether all the time. Serena Nik-Zainal lab has recently proposed the idea of tissue-specific signatures (Degasperi et al. Nat Cancer 2020), where the same signature might have slightly different shapes according to the tumor type and location. Have the Authors checked the cosine similarities of their simulated genomes with the real genome from the PCAWG? Or have the Authors included some noise (e.g. random mutations) to see how this affects the performance and the rate of false-positive and -negative?

Following the Reviewer's suggestion, we have calculated the cosine distances between our simulated genomes and the real genomes from PCAWG (**Figure R3a**). The small cosine distances (mean 0.047, standard deviation 0.051) suggest that our simulations faithfully represented the real genomes in the

vast majority of cases. Although the cosine distances were relatively large for a small number of samples (cosine distance > 0.2 for 2% of samples), the similarities to the real genomes were limited by the accuracy of PCAWG results, but not our simulation procedure (**Figure R3b**). Specifically, we simulated our synthetic genomes from PCAWG decomposition results (i.e., signature and exposure matrices) in order to have a known ground truth, and thus our simulated genomes will be less similar to the real genomes when the PCAWG results do not reconstruct the real genomes accurately (**Figure R3b**).

Figure R3. Similarity between simulated genomes and real genomes from PCAWG. (a) Probability distribution of the cosine distances between simulated genomes and real genomes from PCAWG. **(b)** The cosine distance between our simulated genomes and real genomes closely followed the cosine distance between real genomes and reconstructed genomes using the signature and exposure matrices produced by the PCAWG consortium.

We thank the Reviewer for the suggestion of studying how noise affects the performance. To clarify, our original simulated genomes already included random Poisson sampling noises, since the simulation was performed with a multinomial sampler from true signatures and exposures provided by the PCAWG consortium. To include additional noise, we further added random Gaussian noise at different levels (1%, 2.5%, 5%, and 10%) to the simulated genomes. In more detail, we followed the SigProfilerExtractor paper [3] and resampled each element of the mutation count matrix from a Gaussian distribution where the mean was equal to the original count and the standard deviation was equal to the original count multiplied by the noise level. The resulting matrix was then rounded to the closest integers, and negative values were set to 0. As a result, 4 additional synthetic datasets (corresponding to the 4 noise levels) were simulated for each of the 250 existing ones (25 tumor types \times 10 replicates). In total, 1,000 additional synthetic datasets were generated, corresponding to $\sim 91k$ simulated genomes. We benchmarked the performance of MuSiCal against SigProfilerExtractor on these datasets and summarized the results in the new **Extended Data Figure 8** (shown below). While the performance of both tools slightly deteriorated as the noise level increased, MuSiCal consistently outperformed SigProfilerExtractor at all noise levels. Of note, following the suggestion from Reviewer 2, we now use the entire precision-recall curve (PRC) and the area under PRC (auPRC)

to compare the performances, instead of a single F1 score in our original submission (see response to Reviewer 2's comment 10 for more details).

Extended Data Figure 8. MuSiCal outperforms SigProfilerExtractor for *de novo* signature discovery at different noise levels. (a) Area under precision-recall curve (auPRC) for MuSiCal and SigProfilerExtractor at different noise levels. Each box in the box plot represents 250 synthetic datasets (25 tumor types \times 10 replicates). auPRC was calculated for each dataset separately, as in Fig. 3b. ***: $p < 0.0005$. p -values were calculated with paired t-tests. Box plots indicate median (center line), upper and lower quartiles (box limits), and 1.5x interquartile range (whiskers). **(b)** Precision-recall curve (PRC) for MuSiCal and SigProfilerExtractor at different noise levels. Each PRC represents the average result of 250 synthetic datasets (25 tumor types \times 10 replicates), as in Fig. 3c. **(c)** Precision of MuSiCal and SigProfilerExtractor averaged across all tumor types at different noise levels. Recall was fixed at 0.9. Error bars indicate standard deviation over 10 replicates. **(d)** Recall of MuSiCal and SigProfilerExtractor averaged across all tumor types at different noise levels. Precision was fixed at 0.98, corresponding to a false discovery rate (FDR) of 2%. The black triangle indicates the case where a precision of 0.98 was never achieved and the recall at the highest achieved precision was shown. Error bars indicate standard deviation over 10 replicates.

There are other complicating factors in real data, as pointed out by the Reviewer. For example, not all signatures present in a tumor type are active in every single sample of that tumor type. However, this scenario was already captured in our simulations: as in the PCAWG results (which our simulations were based on), only a subset of our simulated samples have nonzero exposures of a signature within a given tumor type. Furthermore, the same signature may have different shapes in different tumor types (we're familiar with Degasperi et al. Nat Cancer 2020 – we were one of the referees). However,

our simulations and benchmark analysis were already performed per tumor type and thus not affected by tumor type-specific differences. Capturing this complexity would simply amount to slightly modifying the starting signatures for each tumor type. As our simulations already covered a large range of variations of signatures present in different tumor types, these relatively smaller tumor type-specific differences of the same signatures will not affect the final benchmark result.

3. As additional ground truth on which to test the tool's performances, the Authors could pick real tumors from PCAWG (and/or other datasets) and check their accuracy in defining signatures that are expected to be present. For example, they can test how many patients with BRCA-mutated breast cancer with high HRD-detect have a significant presence of SBS3. Alternatively, the Authors can check the contribution of temozolomide and platinum chemotherapy signatures in patients previously exposed to these chemotherapy agents.

Following the Reviewer's suggestion, we checked MuSiCal-derived SBS3 assignment against the HRD status predicted by HRDetect [2]. Of note, testing how many HRD cases have SBS3 provides only a partial view (i.e., sensitivity) of the problem, as a high sensitivity could be accompanied by a large false positive rate if SBS3 is over-assigned to many HRP tumors. We therefore calculated the sensitivity of SBS3-based HRD classification while controlling for fixed false positive rates, such that different methods can be compared on the same ground. Specifically, we used MuSiCal-derived SBS3 assignment to classify HRD status in breast, ovary, pancreas, and prostate tumors, with the HRDetect final prediction considered as true labels [2]. We compared the performance with NNLS-derived SBS3 assignment and the SBS3 input used in HRDetect [2]. As shown in **Figure R2** (described in the response to Reviewer's comment 1, copied below for convenience), MuSiCal-derived SBS3 assignment achieved the best sensitivity at the same false positive rate (FPR). We expect that the performance of multivariate classifiers such as HRDetect can be further improved in the future if MuSiCal-derived signature assignments are used as input, especially when MuSiCal is applied to other signature types as well (e.g., indel and SV).

Figure R2. MuSiCal-derived SBS3 assignment outperforms the SBS3 feature used in HRDetect for

detecting HRD. We compared the performance of SBS3 assignments derived from NNLS, signature.tools.lib, and MuSiCal for detecting HRD in breast, ovary, pancreas, and prostate tumors. The signature.tools.lib-derived SBS3 assignment was used as input to HRDetect in [2] (Degasperi et al. Nat Cancer 2020). The HRDetect final classification was considered as true labels. Sensitivity was shown at FPR values 5% and 10%.

Clinical annotation of PCAWG samples is limited. In particular, treatment type is only coarsely separated into chemotherapy, surgery, combined chemo + radiation therapy, other therapy, no treatment, and not available (NA), with no information about the specific drug used. Therefore, a similar analysis as in **Figure R2** could not be performed for temozolomide- or platinum-related signatures. We thus investigated relevant samples individually and provided empirical evidence using the reconstruction error spectra, as detailed below.

SBS11 is associated with temozolomide treatment (**Figure R4a**). MuSiCal assigned SBS11 only to one sample in the PCAWG dataset. The mutational spectrum of this sample clearly resembles SBS11 (**Figure R4b**). PCAWG assigned SBS11 to two additional samples. In **Figure R4c**, we plotted the reconstruction error spectra of MuSiCal assignments (i.e., raw sample spectrum - MuSiCal reconstruction) for these two samples. The positive (+) component of the error spectra reflects signatures potentially missed by the MuSiCal assignment, but it did not match to SBS11 (matched to SBS32 and SBS58 instead, **Figure R4c**).

This observation suggests that MuSiCal did not under-assign SBS11 in these two samples, and the PCAWG assignment of SBS11 in these two samples are likely to be false positives.

Figure R4. Comparison of MuSiCal and PCAWG assignments for SBS11. (a) Spectrum of COSMIC SBS11. (b) Raw sample spectrum of the single sample with nonzero SBS11 exposure in MuSiCal assignment. PCAWG also assigned SBS11 to this sample. (c) Left: Raw sample spectra of the two additional samples with nonzero SBS11 exposure in PCAWG assignment. MuSiCal did not assign SBS11 to these two samples. Sample names were annotated along with the corresponding tumor types, treatment types, and relative

exposures of SBS11. Right: MuSiCal reconstruction error spectra of the two samples shown to the left. Reconstruction error spectra were calculated as raw sample spectra - MuSiCal reconstruction. Top signatures matched to the positive (+) components of the error spectra were annotated along with the corresponding cosine similarities.

Two COSMIC signatures, SBS31 and SBS35, are associated with platinum drug treatment (**Figure R5a**). Samples with nonzero exposures of SBS31 and/or SBS35 in both MuSiCal and PCAWG assignments showed clear visual evidence in their raw mutational spectra (**Figure R5b**). For samples with SBS31/SBS35 assignments by MuSiCal but not by PCAWG, their PCAWG reconstruction error spectra often matched to SBS31/SBS35, suggesting that in these samples SBS31/SBS35 were potentially under- assigned by PCAWG, and the SBS31/SBS35 assignment by MuSiCal were likely to be real (**Figure R5c**). By comparison, for samples with SBS31/SBS35 assignments by PCAWG but not by MuSiCal, their MuSiCal reconstruction error spectra did not match to SBS31/SBS35, suggesting that in these samples MuSiCal did not under-assign SBS31/SBS35, and the SBS31/SBS35 assignment by PCAWG are likely to be false positives (**Figure R5d**).

Figure R5. Comparison of MuSiCal and PCAWG assignments for SBS31 and SBS35. (a) Spectra of COSMIC SBS31 and SBS35. **(b)** Raw sample spectra of the samples with nonzero SBS31/SBS35 exposures in both MuSiCal and PCAWG assignments. Sample names were annotated along with the corresponding tumor types, treatment types, and relative exposures of SBS31/SBS35. Top 5 samples ranked by the relative exposure of SBS31/SBS35 were shown. **(c)** Left: Raw sample spectra of the samples with nonzero SBS31/SBS35 exposures in MuSiCal assignment but zero SBS31/SBS35 exposures in PCAWG assignment. Sample names were annotated along with the corresponding tumor types, treatment types, and relative exposures of SBS31/SBS35. Top 5 samples ranked by the relative exposure of SBS31/SBS35 were shown. Right: PCAWG reconstruction error spectra of the samples shown to the left. Reconstruction error spectra were calculated as raw sample spectra - PCAWG reconstruction. Top signatures matched to the positive (+) components of the error spectra were annotated along with the corresponding cosine similarities. **(d)** See below.

Figure R5 (cont'd) (d) Left: Raw sample spectra of the samples with nonzero SBS31/SBS35 exposures in PCAWG assignment but zero SBS31/SBS35 exposures in MuSiCal assignment. Sample names were annotated along with the corresponding tumor types, treatment types, and relative exposures of SBS31/SBS35. Right: MuSiCal reconstruction error spectra of the samples shown to the left. Reconstruction error spectra were calculated as raw sample spectra - MuSiCal reconstruction. Top signatures matched to the positive (+) components of the error spectra were annotated along with the corresponding cosine similarities.

4. Since 2020 SigProfiler has been the gold standard for mutational signatures analysis. Compared to the original NNMF, it is built around three steps: 1) first it runs a de novo extraction, 2) then it matches the new extracted signatures with the latest COSMIC reference, and 3) it applies a fitting approach to correctly quantify their contribution. Across these different steps, it also provides alternative solutions. I agree that in the setting of in silico WGS SigProfiler tends to provide a slightly worse de novo extraction compared to MuSiCal. However, at the end of the final fitting step, it can still provide an accurate estimation of the right signatures, excluding the ones that are not relevant/active. Have the Authors compared the final PCAWG results with their MuSiCal re-analysis?

Indeed! We had compared the final results produced by the PCAWG consortium with our MuSiCal reanalysis results extensively throughout our manuscript. First, direct comparison of the final signature assignments was presented in Extended Data Figure 18 (previously Extended Data Figure 14), copied here. As seen in the “Comparison” matrix, the difference in SBS40 exposure is perhaps most noticeable, but there are many other differences, such as SBS3, 8, 18, 34, etc. I think the Reviewer will agree with us that these differences are substantial, given the potential use of signatures in clinical applications (e.g., SBS3 for HRD).

Extended Data Figure 18. Comparison of signature assignments obtained by MuSiCal and the PCAWG consortium.

Second, we developed an *in silico* validation approach to assess the quality of the final signature assignment (Results Section 2.4). In Figure 5 (bottom half copied below), we illustrated this approach using the glioblastoma dataset as an example and demonstrated that the signature assignments from MuSiCal were more consistent with original data compared to those from the PCAWG consortium.

Figure 5 (bottom half). Signature assignments from MuSiCal achieved improved consistency in *in silico* validation for the glioblastoma dataset.

Further application of this approach to other tumor types showed overall improvement achieved by MuSiCal for both SBS and indel signatures (Figure 7a, b, Extended Data Figure 21 (previously Extended Data Figure 17), Results Section 2.7). Figure 7a (for SBS signatures) and Extended Data Figure 21 (for indel signatures) were copied below.

Figure 7a (left) and Extended Data Figure 21 (right). Comparison between MuSiCal- and PCAWG-derived SBS and ID signature assignments in terms of their consistency with data.

Third, we provided additional examples to show that the final signature assignments from MuSiCal were more biologically meaningful compared to those from the PCAWG consortium: MuSiCal resolved the over-assignment issue of SBS40 and achieved improved correlation between clock-like signatures SBS1 and SBS5 (Figure 7c, d, Extended Data Figure 22, 23 (previously Extended Data Figure 18, 19), Results Section 2.7); MuSiCal results showed expected associations between SBS and ID

signatures more clearly and revealed novel associations supported by orthogonal data (Figure 6c, Extended Data Figure 19, 20 (previously Extended Data Figure 15, 16), Results Section 2.6; we did not copy the figures mentioned in this paragraph here because there are too many)

SigProfiler is a solid algorithm and so many discoveries have been made using it — it has helped to launch the field and we've used it in previous papers ourselves. As the Reviewer noted, it does have many steps, and its developers are continuing to refine them. However, I hope our earlier result, e.g., Figure R2, has convinced the Reviewer that once the *de novo* extraction does not produce the correct signatures, it is difficult to recover in subsequent steps, no matter how much filtering and optimization are carried out. We have found that marked improvements can be made at each step — not just in *de novo* signature discovery with mvNMF but also in matching/refitting and in optimization steps. As the size of WGS data increases, more signatures will be “discovered” by existing methods, and the problem of false positive and false negative assignments will grow. We think a lot of work remains to be done by the community and we offer MuSiCal as a considerable advance.

5. The MuSiCal F1 is significantly higher in 8 tumors out of 25 (Figure 3b). Most of these are known to be hypermutated, with one or two dominant signatures “covering” the others. This is a known problem in the *de novo* extraction performances, in particular using NMF. In this setting, MuSiCal seems to work better. Have the Authors explored if the better MuSiCal performance is related to the tumor with a high mutational burden?

We plotted the performance gain by MuSiCal against the tumor mutational burden for each tumor type and did not observe a correlation (Pearson $r = -0.08$, $p = 0.70$, **Figure R6**). Therefore, the better MuSiCal performance was not related to higher mutational burden. Of note, following the suggestion from Reviewer 2, we now use the area under precision-recall curve (auPRC) to quantify the performances, instead of a single F1 score in our original submission (see the response to Reviewer 2's comment 10 for more details). Using the F1 score in our original submission, a significant correlation with tumor mutational burden was also not observed (Pearson $r = 0.11$, $p = 0.60$ without Skin-Melanoma, Pearson $r = 0.38$, $p = 0.06$ with Skin-Melanoma).

Figure R6. The better performance of MuSiCal was not related to higher tumor mutational burden. The auPRC gain of MuSiCal compared to SigProfilerExtractor was plotted against the mean SBS count for each tumor type. A correlation was not observed between the two variables. Pearson correlation coefficient (r) and the corresponding p -value were annotated.

6. I agree that manual adjustment for defining the right number of signatures is not optimal. However, a prior knowledge of which signatures we might expect in a certain tumor is definitely important and would reduce false positives. For example, in contrast to PCAWG, the Authors extracted SBS31 from

Glioblastoma WGS. This signature is known to be caused by platinum exposure, so I wonder if the Authors verified if these cases had ever been exposed to platinum-containing regimens before the sample collection. The same platinum-induced signature seems to be extracted in MPN and AML. In both tumors, platinum is never used, and as far as I know, all samples were collected at baseline. Is this a sign that the high sensitivity also increases the number of false positives (reduce specificity)?

We agree with the Reviewer that it is important to impose prior knowledge of which signatures we should expect in a given tumor type. In fact, we recommended in our tutorial that the signature catalog should be restricted to tumor type-specific signatures during the matching and refitting steps (<https://github.com/parklab/MuSiCal/tree/main/examples>). However, such prior knowledge needs to be established in the first place through an unbiased approach. In this manuscript, we aimed at revisiting the problem of which signatures are likely to occur in each tumor type and thus took the unbiased approach without using any prior information. This approach was important for, e.g., ruling out the presence of SBS40 in most tumor types where SBS40 were previously believed to exist (Results Section 2.7).

MuSiCal assigned SBS31 to 3 glioblastoma samples. By contrast, PCAWG did not assign SBS31 to any glioblastoma samples. As mentioned in the response to Reviewer's comment 3, clinical annotation of PCAWG samples is limited. Specifically for these 3 glioblastoma samples with SBS31 assignment from MuSiCal, treatment type is annotated as NA (not available). Therefore, we could not validate the result with treatment information. We thus again investigated these samples individually and provided empirical evidence using the reconstruction error spectra, as we did in the response to Reviewer's comment 3. As shown in **Figure R7**, for these 3 samples, the reconstruction error spectra according to the results produced by the PCAWG consortium matched to SBS31 or SBS35, both associated with platinum chemotherapy treatment. This observation suggests that in these 3 samples SBS31/SBS35 were potentially under-assigned by PCAWG, and the SBS31 assignment by MuSiCal were likely to be real.

Furthermore, in Figure 5b-f, we have compared the signature assignment in glioblastoma by MuSiCal and PCAWG with our *in silico* validation approach and showed that the MuSiCal assignment (with SBS31) achieved much improved consistency between simulation and data. Finally, we do not rule out the possibility that the small amount of SBS31 in glioblastoma represents an unknown mutational process (e.g., another drug treatment) that produces a similar spectrum to SBS31.

Figure R7. Comparison of MuSiCal and PCAWG assignments for SBS31 in glioblastoma. (a) Spectra of COSMIC SBS31 and SBS35, both associated with platinum chemotherapy treatment. (b) Left: Raw sample spectra of the CNS.GBM samples with nonzero SBS31 exposures in MuSiCal assignment. Sample names were annotated along with the corresponding tumor types, treatment types, and relative exposures of SBS31. Right: PCAWG reconstruction error spectra of the samples shown to the left. Reconstruction error spectra were calculated as raw sample spectra - PCAWG reconstruction. Top signatures matched to the positive (+) components of the error spectra were annotated along with the corresponding cosine similarities.

MuSiCal assigned SBS31 to Myeloid.AML and Myeloid.MDS, but not to Myeloid.MPN (the confusion was due to misreading the columns corresponding to MDS and MPN from the heatmap in Figure 6b, which is large and challenging to read). One Myeloid.AML sample (SP127706) was clearly dominated by SBS31, and the presence of SBS31 could be visually confirmed by the raw spectrum shown in **Figure R8**. PCAWG also assigned SBS31 to this sample. We think that this sample may be an example of secondary therapy- related AML (tAML) where the patient received platinum chemotherapy treatment for a prior solid malignancy. Indeed, SBS31 was observed in all platinum treated tAML samples in a previous study [4].

Figure R8. SBS31 in myeloid tumors. (a) Spectrum of COSMIC SBS31. (b) Spectrum of the Myeloid.AML sample SP127706, which clearly resembles SBS31 (cosine similarity = 0.98).

Figure R8. SBS31 in myeloid tumors. (a) Spectrum of COSMIC SBS31. **(b)** Spectrum of the Myeloid.AML sample SP127706, which clearly resembles SBS31 (cosine similarity = 0.98).

Seven other AML/MDS samples received small SBS31 exposures from MuSiCal (on average 8.6% compared to 91.9% in SP127706). Although it is possible that some of these samples are also secondary therapy-related malignancies, we agree with the Reviewer that they are more likely false positives, because they were analyzed together with SP127706, which was clearly an outlier in this case. It was our oversight not to perform a separate analysis with this outlier removed in our original submission. We thus repeated the analysis (*de novo* signature discovery followed by matching and refitting) after excluding SP127706 during the revision. In the updated calculation, SBS31 was not assigned anymore. As a result, no other myeloid tumor samples except SP127706 has nonzero SBS31 exposures in the revised MuSiCal assignment. We have modified relevant figures in the manuscript to reflect this update. We have also double checked other tumor types with the outlier removal module in MuSiCal to make sure that similar issues were not present in other tumor types. Of note, AML and MDS were combined with MPN in the analysis due to their small sample sizes (11 for AML and 4 for MDS, see Methods Section 4.5.1 for more details). We also point out that myeloid tumors were among the most challenging tumor types to analyze, due to their small sample sizes and mutational burdens. Therefore, in the updated calculation described above, we used the joint two-dimensional grid search for parameter optimization instead of the stepwise one-dimensional grid searches in order to achieve better resolution (see Extended Data Figure 13 for more details). In general, we do not think that MuSiCal sacrifices specificity for sensitivity (see the response to Reviewer's comment 3, Figure 3, and Extended Data Figure 12).

7. Another example of a possible false positive call is related to SBS84. This mutational signature is known to be caused by somatic hypermutation in normal and transformed post germinal center B-cells. Nevertheless, SBS84 activity is reported in bone benign tumors and in different myeloid tumors (MDS, MPN, and AML). This data is against the intrinsic biology of these tumors, known to be completely independent of somatic hypermutation. Similarly, SBS9 (poly-eta in post germinal center tumors) is extracted among MPN and AML.

As described in the response above to Reviewer's comment 6, we have repeated the analysis (*de novo* signature discovery followed by matching and refitting) of myeloid tumors after excluding the outlier SP127706 dominated by SBS31. In the updated results, SBS84 was not assigned anymore, but SBS9 persisted. Two Myeloid.MPN samples still received nonzero SBS9 exposures, while PCAWG did not assign SBS9 to any myeloid tumors. The raw spectra of these two samples demonstrated characteristic peaks of SBS9 at T>G (especially TTA>TGA) and T>A (**Figure R9**). The reconstruction error spectra according to the results produced by the PCAWG consortium matched to SBS9, suggesting that in these two samples SBS9 was potentially under-assigned by PCAWG (**Figure R9**). Together, these observations suggest that the SBS9 assignment in these two samples by MuSiCal were likely to be true positives. We reasoned that since SBS9 is associated with the activity of polymerase eta, which has roles beyond somatic hypermutation in lymphoid cells, observing SBS9 in tumor types other than lymphoid tumors is likely.

Figure R9. SBS9 in myeloid tumors. (a) Spectrum of COSMIC SBS9. **(b) Left:** Raw sample spectra of the myeloid tumor samples with nonzero SBS9 exposures in MuSiCal assignment. Sample names were annotated along with the corresponding tumor types, treatment types, and relative exposures of SBS9. **Right:** PCAWG reconstruction error spectra of the samples shown to the left. Reconstruction error spectra were calculated as raw sample spectra - PCAWG reconstruction. Top signatures matched to the positive (+) components of the error spectra were annotated along with the corresponding cosine similarities.

MuSiCal did not assign SBS84 to Bone.Benign tumors, but to Bladder.TCC (the confusion was again due to misreading the large heatmap in Figure 6b). Regardless, we agree with the Reviewer that it is unlikely to observe the AID-related SBS84 in tumor types other than lymphoid tumors. However, as already explained in the response to Reviewer's comment 6, we chose not to incorporate any prior information in the manuscript because we aimed at re-establishing this prior knowledge from scratch. In this case, again with the aid of analyzing the reconstruction error spectrum, we found that SBS84 was indeed potentially under-assigned by PCAWG and that the SBS84 assignment by MuSiCal in Bladder.TCC was likely to be real (**Figure R10**). We think that this assignment may result from a subtle unknown mutational process that produces a spectrum similar to SBS84. Additional work will be required in the future to further elucidate this possibility.

Figure R10. SBS84 in Bladder.TCC. (a) Spectrum of COSMIC SBS84. **(b)** Left: Raw sample spectrum of the top Bladder.TCC sample with nonzero SBS84 exposures in MuSiCal assignment. The sample name was annotated along with the corresponding tumor type, treatment type, and relative exposure of SBS84. The rest of the Bladder.TCC samples with SBS84 assignments ($n = 4$) had negligible exposures (2.3% on average). Right: PCAWG reconstruction error spectrum of the sample shown to the left. Reconstruction error spectrum was calculated as raw sample spectra - PCAWG reconstruction. The top signature matched to the positive (+) component of the error spectrum was annotated along with the corresponding cosine similarity.

Finally, we point out that although our signature assignments with MuSiCal represent a pronounced overall improvement compared to the results produced by the PCAWG consortium (see response to Reviewer's comment 4 for detailed explanations), we do not rule out the possibility that errors are still present in our results. After all, signature assignment is a challenging problem, especially given the limitations of the current signature definitions (see Discussion section) and of the data analyzed in this manuscript. We believe that application of MuSiCal to broader datasets with improved signature definitions in the future will help further revise our prior knowledge of tumor-type specific signatures, although such work is out of scope of this manuscript.

8. I appreciate that the Authors introduced a fitting part in their workflow. However, it seems that their tool is not so different from SigProfiler fitting tool. Which are the key differences? Both seem to be based on cosine similarities and reconstruction after a leave one out of each signature.

Our matching and refitting algorithm in MuSiCal is fundamentally different from that in the SigProfiler suite. Importantly, our algorithm is not based on cosine similarities, but on multinomial likelihoods. In our previous paper describing SigMA [1], we had realized that multinomial likelihood is more robust and statistically sound, and thus should be a preferred similarity measure compared to cosine similarity not just for our algorithm but for others as well (see, for example, Supplementary Figures 6-8 in [1]). In MuSiCal, we generalized this idea and developed a sparse NNLS algorithm based on multinomial likelihoods. In the old version, we explained our algorithm in detail in Results Section 2.3 and Methods Section 4.3.2. To further emphasize the difference between multinomial

likelihoods and cosine similarities, we also provided two intuitive examples demonstrating that multinomial likelihoods are more powerful in separating similar signatures (Extended Data Figure 11b) and better suited for alleviating the over-assignment problem in matching and refitting (comparison between Figure 4b and Extended Data Figure 11c).

The stepwise fitting procedures implemented in MuSiCal and the SigProfiler suite are more similar, being inspired by the commonly used forward and backward stepwise regression algorithm. However, our algorithm employs a single threshold (per-mutation log likelihood difference), whereas SigProfiler requires three. This simplification was enabled by the unified statistical interpretation of both the forward and backward steps in our algorithm, and allowed efficient parameter optimization with the *in silico* validation/optimization module.

Finally, systematic benchmark studies were performed in our manuscript to compare the performance of the two algorithms, and much improved accuracy was achieved by our likelihood-based approach (Figure 4c, d and the new Extended Data Figure 12, see the response to Reviewer 2's comment 15 for more details).

9. Overall, I believe that the community needs a more standardized and reproducible mutational signature tool. However, my impression after reading this study is that mathematical unbiased modelings have reached the limit of their capability in mutational signature extraction. To make MuSiCal more innovative and accurate compared to the existing ones, the Authors might consider integrating their models with additional and established features (e.g. known signatures, clinical history, presence of distinct gene mutations). I believe that this will compensate for the rate of false-positive, further increasing the model accuracy.

We agree with the Reviewer that the community needs a more standardized and reproducible mutational signature tool. Although the standard framework of signature analysis has generated extensive biological insights in the past decade, we do not think that a limit has been reached in terms of improving this framework. In our manuscript, we have identified methodological problems with the standard framework and developed new algorithms to solve these problems. We have shown that these developments were not only conceptual, but translated into practical improvements in both simulation studies and real data applications. As the number of whole genomes of cancer and other diseases keeps growing rapidly, the mathematically unbiased approach of signature analysis will continue to be a major driver of new discoveries in the future. We thus believe that MuSiCal will be of great value to the community going forward. Finally, as explained in detail in the response to Reviewer's comment 1, MuSiCal lays the foundation and improves the performance of the multivariate approach proposed by the Reviewer. We are actively working on integrating MuSiCal with other clinical and genomic features through the SigMA framework in a separate study.

Reviewer #2:

Remarks to the Author:

In the manuscript “Accurate and sensitive mutational signature analysis with MuSiCal”, Jin and colleagues present a set of novel algorithms for the detection, assignment and validation of mutational signatures. The work contains three major novelties that aim to solve problems persisting in standard mutational signature calling: 1) The application of Minimum-volume NMF, a method from signal processing, that guarantees unique solutions. 2) A likelihood-based sparse NNLS method for assigning signatures to individual samples that aims to fix the over-assignment of low-exposure signatures. 3) A data-driven simulation approach to validate the detection and assignment of signatures. After explaining and validating the different parts of the model, the new workflow is applied to over 2,700 samples from the Pan Cancer Analysis of Whole Genomes to retrieve new signatures and alleviate some problems with the common approach including the over-representation of flat signatures.

The authors address a highly timely and important subject that can provide a substantial advance in mutational signature analysis over the current state-of-the-art. The manuscript combines multiple interesting concepts that pose a substantial improvement to the field of mutational signatures. The concepts are well justified and thoroughly validated and the paper was well written. However, I’m having some issues with the article in its current form, in particular with respect to the comparison to the current state-of-the-art model, that I believe should be addressed before publication.

We thank the Reviewer for the positive comments and the thoroughness in reading our manuscript. In the detailed responses below, we have carefully addressed all major and minor points, which have helped greatly improve the clarity and comprehensiveness of our manuscript. In particular, we have substantially improved the comparison to the state-of-the-art methods by adding the following analyses:

- Additional benchmark analyses at different levels of random noise.
- Additional benchmark analyses with the spike-in of unknown spurious signatures.
- Additional comparisons to signature.tools.lib, SignatureAnalyzer, and SigneR, which, together with SigProfilerExtractor included in our original submission, covered both different underlying algorithms and the most popular tools in the field.
- Comparison of performances using the precision-recall curve (PRC) and area under PRC (auPRC).

MuSiCal outperformed all competing methods under multiple performance metrics, further strengthening the conclusion that MuSiCal outperforms the current state-of-the-art.

Major points

Pre processing

1. In the graphical abstract (Fig 1), the preprocessing module is presented as one of the four fundamental modules of MuSiCal. However, it is only briefly mentioned in the main text (Section 2.5). Since I assume that the preprocessing step affects the downstream analyses in a non-trivial way, this step and its effects should be explained in more detail, not only in the online methods. In particular, preprocessing seems to be an important part of the re-analysis of the PCAWG data. Also, in Ext Fig 10a: What does “reasonable k’s” mean?

We apologize that the preprocessing module was not presented in more detail in the main text. The main text was already long (> 6,000 words), and we wanted to focus on the three major developments

(mvNMF, likelihood-based sparse NNLS, and *in silico* validation/optimization) in the main text, with the rest in Methods and Extended Data Figures. For example, we also had to skip the details of the new method for selecting the number of signatures in *de novo* signature discovery (Methods Section 4.3.1, Extended Data Figure 6). However, we agree with the Reviewer that preprocessing is an important step that requires more explanations in the main text. The text is even longer now with the additional analyses we have done in response to the three Reviewers, so we have added the following text to Results Section 2.5.

... Tumor type-specific *de novo* signature discovery is performed after stratifying heterogeneous datasets and removing outliers with the preprocessing module (Extended Data Fig. 14).

Specifically, the preprocessing module employs two complementary methods to further improve the sensitivity of *de novo* signature discovery. In automatic cohort stratification, heterogeneous datasets are stratified into subsets with gap statistic-informed hierarchical clustering [54, 55] (Methods), such that subtle signatures within each subset can be discovered cleanly (Extended Data Fig. 14a-c). In outlier removal, the Gini coefficient is applied to identify outliers with strong exposures of signatures that are not present in other samples [63] (Methods). Removing these outliers increases the sensitivity and accuracy of detecting other more prevalent signatures (Extended Data Fig. 14d-f). The discovered signatures are subsequently matched to the COSMIC catalog, and potential novel signatures are identified when a reasonable match cannot be found. ...

It was our oversight not to explain what “reasonable k’s” means. We have now added the following text to the caption of Extended Data Figure 14 (previously Extended Data Figure 10).

In panel (a), reasonable k means any k satisfying $\text{Gap}(k) \geq \text{Gap}(k + 1) - s_{k+1}$, where $\text{Gap}(k)$ denotes the gap statistic (indicated by dots in the plot) and s_k the dispersion due to simulation (indicated by error bars in the plot). The smallest reasonable k is chosen as the optimal k. See [54, 55] for more details.

2. For the re-analysis of PCAWG in section “Implementation of preprocessing” the authors write “Next, we stratified samples into subsets with distinct signature compositions with hierarchical clustering.” Here, the final number of subsets per cancer type should be reported (best to give the mean and range). Furthermore, the range of removed samples should be reported (not just the average).

We have modified the section “Implementation of preprocessing” and provided additional information (shown below).

After an initial step of *de novo* signature discovery within each tumor type, the Gini-based outlier removal was applied, removing on average 2.3% of samples (ranging between 0% to 7.0% in different tumor types). For Kidney.RCC, Liver.HCC, and Skin.Melanoma, we further stratified samples into subsets with distinct signature compositions using hierarchical clustering on the exposure matrix *H* obtained from the initial *de novo* signature discovery. The distinct clusters within these tumor types were defined by several hypermutator processes including aristolochic acid, aflatoxin, and UV exposures. Note that we separated the MMRD and POLE-exo mutant samples before this stage, although preprocessing can be used to stratify these samples if they had not been isolated beforehand (Extended Data Fig. 14a-c).

To clarify, on top of separating MMRP and MMRD/POLE-exo samples at the very beginning, further cohort stratification was only done in Kidney.RCC (with and without aristolochic acid exposure), Liver.HCC (with and without aristolochic acid/aflatoxin exposures), and Skin.Melanoma (with and without UV exposure).

Showing improvements from PCAWG reanalysis in Ext. Data Fig. 2

3. Extended Data Fig. 2 shows the problems with the current COSMIC signatures quite well. It would be good to have a before/after comparison with the *de novo* detected signatures from Section 2.5.

The main purpose of Extended Data Figure 2 was to show that the high similarity among COSMIC SBS signatures poses a significant challenge to accurate signature assignment. Although it is possible that some COSMIC SBS signatures are subtle variations of other ones, we do not think that the overall signature similarity represents the quality of the catalog itself. It is likely that certain mutational processes do produce similar signatures. For example, signatures that only differ in extended nucleotide contexts (e.g., +/-2bp) may look similar in the 96-channel trinucleotide definition. Therefore, we do not think that a before/after comparison with *de novo* signatures extracted from MuSiCal is informative.

Note that we did not perform a global update for COSMIC SBS signatures in the manuscript, except for adding 6 new ones. The COSMIC SBS signature catalog also contains signatures extracted from multiple data sources other than PCAWG, as well as signatures manually curated from experimental studies.

Substantial additional work will be required to further refine the SBS signature catalog in the future. In **Figure R11** (shown below), we show a before/after comparison for ID signatures, as a global update was

performed for the ID signature catalog in our manuscript. MuSiCal-derived ID signatures were not particularly more similar or dissimilar than COSMIC ones.

Figure R11. Similarity among ID signatures in the COSMIC catalog and the MuSiCal-derived catalog. (a) Similarity among ID signatures in the COSMIC catalog. For each signature, the maximum cosine similarity to another signature in the catalog is plotted on the y-axis, and the cosine similarity to the signature reconstructed by fitting with all other signatures in the catalog through NNLS is plotted on the x-axis. **(b)** Same as (a), but for MuSiCal-derived ID signatures.

Simulation studies 4. The simulation studies detailed in Methods 4.3 are an integral part of this paper and in support of the authors' claim that their method outperforms both standard NMF and SigProfilerExtractor. The source code for the simulation studies should be available along with the code for MuSiCal so that the outcome can be validated independently.

The source code for the simulation studies was already included in the Github repository of MuSiCal. Specifically, the function `musical.simulation.simulate_LDA()` was used for simulations in Methods Sections 4.4.1 and 4.4.4 (previously Methods Sections 4.3.1 and 4.3.3), and the function `musical.utils.simulate_count_matrix()` was used to simulate synthetic genomes from PCAWG signature and exposure matrices in Methods Sections 4.4.2 (previously Methods Section 4.3.2) and 4.4.3. We apologize for omitting these references in our original submission; we have modified the Methods section to include this information.

5. Furthermore, from my understanding, the current simulation studies does not include randomly generated noise but only a random combination of the true signatures. As real biological data includes a variety of noise it is important to check the effect of noise on the performance of MuSiCal and SigProfileExtractor, both uniformly random noise as well as spurious signatures that are spiked in, in addition to the “true” signatures. I would suspect that this would have a measurable effect, for example for the faint detection of some signatures seen e.g. in Fig. 3a. I therefore recommended that the comparison be repeated with varying strength of noise and the robustness of MuSiCal to random noise reported.

To clarify, our original simulated data already included random Poisson sampling noises, since the simulation was performed with a multinomial sampler from true signatures and exposures provided by the PCAWG consortium. In other words, the synthetic mutation count matrices were not simply matrix products of the true signature and exposure matrices but resampled from these matrix products.

But we do agree that additional random noise would be helpful in benchmarking the robustness of MuSiCal better. To follow the Reviewer’s suggestion, we further added random Gaussian noise at different levels (1%, 2.5%, 5%, and 10%) to the simulated genomes. In more detail, we followed the SigProfilerExtractor paper [3] and resampled each element of the mutation count matrix from a Gaussian distribution where the mean was equal to the original count and the standard deviation was equal to the original count multiplied by the noise level. The resulting matrix was then rounded to the closest integers, and negative values were set to 0. As a result, 4 additional synthetic datasets (corresponding to the 4 noise levels) were simulated for each of the 250 existing ones (25 tumor types × 10 replicates). In total, 1,000 additional synthetic datasets were generated, corresponding to ~91k simulated genomes. We benchmarked the performance of MuSiCal against SigProfilerExtractor on these datasets and summarized the results in the new **Extended Data Figure 8** (shown below). While the performance of both tools slightly deteriorated as the noise level increased, MuSiCal consistently outperformed SigProfilerExtractor at all noise levels. Of note, as suggested by the Reviewer in comment 10, we now use the entire precision-recall curve (PRC) and the area under PRC (auPRC) to compare the performances, instead of a single F1 score in our original submission (see response to Reviewer’s comment 10 below for more details).

Extended Data Figure 8. MuSiCal outperforms SigProfilerExtractor for *de novo* signature discovery at different noise levels. (a) Area under precision-recall curve (auPRC) for MuSiCal and SigProfilerExtractor at different noise levels. Each box in the box plot represents 250 synthetic datasets (25 tumor types \times 10 replicates). auPRC was calculated for each dataset separately, as in Fig. 3b. ***: $p < 0.0005$. p -values were calculated with paired t-tests. Box plots indicate median (center line), upper and lower quartiles (box limits), and 1.5x interquartile range (whiskers). **(b)** Precision-recall curve (PRC) for MuSiCal and SigProfilerExtractor at different noise levels. Each PRC represents the average result of 250 synthetic datasets (25 tumor types \times 10 replicates), as in Fig. 3c. **(c)** Precision of MuSiCal and SigProfilerExtractor averaged across all tumor types at different noise levels. Recall was fixed at 0.9. Error bars indicate standard deviation over 10 replicates. **(d)** Recall of MuSiCal and SigProfilerExtractor averaged across all tumor types at different noise levels. Precision was fixed at 0.98, corresponding to a false discovery rate (FDR) of 2%. The black triangle indicates the case where a precision of 0.98 was never achieved and the recall at the highest achieved precision was shown. Error bars indicate standard deviation over 10 replicates.

In addition, we also performed a spike-in analysis to further investigate the robustness of *de novo* signature discovery by MuSiCal when an unknown spurious signature is present. In more detail, additional mutations from COSMIC SBS48 were spiked-in to our original synthetic datasets. The number of spike-in mutations were equal to 5% of the mutational burden for each sample. SBS48 was chosen to represent a spurious signature because it is not present in the original synthetic dataset and is annotated in COSMIC as a possible sequencing artifact found in cancer samples that were subsequently blacklisted for poor quality of sequencing data. To pretend that SBS48 was unknown, we removed SBS48 from the COSMIC catalog when matching the *de novo* signatures and calculating precision and recall.

The benchmark results using these datasets were summarized in the new **Extended Data Figure 9** (shown below). Even with the spike-in of the unknown spurious signature, MuSiCal still significantly outperformed SigProfilerExtractor.

Extended Data Figure 9. MuSiCal outperforms SigProfilerExtractor for *de novo* signature discovery when an unknown spurious signature is present. **(a)** Area under precision-recall curve (auPRC) for MuSiCal and SigProfilerExtractor with and without spike-in. Each box in the box plot represents 250 synthetic datasets (25 tumor types \times 10 replicates). auPRC was calculated for each dataset separately, as in Fig. 3b. ***: $p < 0.0005$. p -values were calculated with paired t-tests. Box plots indicate median (center line), upper and lower quartiles (box limits), and 1.5x interquartile range (whiskers). **(b)** Precision-recall curve (PRC) for MuSiCal and SigProfilerExtractor with and without spike-in. Each PRC represents the average result of 250 synthetic datasets (25 tumor types \times 10 replicates), as in Fig. 3c. **(c)** Precision of MuSiCal and SigProfilerExtractor averaged across all tumor types with and without spike-in. Recall was fixed at 0.9. Error bars indicate standard deviation over 10 replicates. **(d)** Recall of MuSiCal and SigProfilerExtractor averaged across all tumor types with and without spike-in. Precision was fixed at 0.98, corresponding to a false discovery rate (FDR) of 2%. The black triangle indicates the case where a precision of 0.98 was never achieved and the recall at the highest achieved precision was shown. Error bars indicate standard deviation over 10 replicates.

Uniqueness of the mvNMF solutions

6. Theoretically, mvNMF should be able to find a unique solution to the NMF problem "if the exposure matrix is sparse enough" (p.3). The authors argue that the sparsity assumption is true for most tumor data based on the sparseness for the PCAWG dataset. However, looking for example at Fig 2c, we see that the cosine error for mvNMF has a similar spread as the one for standard NMF. This holds true for almost all samples (in some cases even larger, e.g. SBS5). If the resulting signatures are supposed to be unique, how is the wide spread in cosine error explained?

The apparent similar or larger spread of cosine errors for mvNMF was simply because y-axis was plotted in log scale in Figure 2c. Since mvNMF usually produced smaller cosine errors (e.g., from 0.0001 to 0.01) than NMF did (e.g., from 0.01 to 0.1), the apparent similar or larger spread in log scale (two orders of magnitude vs. one order of magnitude) actually corresponds to a smaller spread

in linear scale (0.0099 vs. 0.09). This point becomes clear when we changed the y-axis of Figure 2c to linear scale (**Figure R12a**) or compared the standard deviation of the cosine errors directly (**Figure R12b**). Therefore, mvNMF indeed produced more stable solutions. The small remaining variations in mvNMF solutions were due to difficulties in the optimization problem. We still chose to keep the log scale in Figure 2c in our manuscript, because in linear scale many signatures with small cosine errors became difficult to see (**Figure R12a**).

Figure R12. Comparison of the spread of cosine errors between NMF and mvNMF. (a) Same as Figure 2c, except that y-axis is plotted in linear scale. It now becomes apparent that mvNMF actually produced smaller spreads in cosine errors compared to NMF. **(b)** Standard deviation (SD) of the cosine errors produced by NMF and mvNMF for different signatures. For the majority of signatures (34/45), mvNMF had smaller SD.

7. Due to the claim of uniqueness for mvNMF I would like to see a comparison between standard NMF and mvNMF w.r.t. the actual volume spanned by both solutions in the mutational space. This should be performed not on simulated data but on the PCAWG data. When grouping the results by tumor type, the results should correspond to the sparsity of the exposures as seen in Ext Fig 4c.

We compared the volumes spanned NMF and mvNMF solutions on real PCAWG data (**Figure R13**). As expected, mvNMF produced smaller volumes for all tumor types (60 fold reduction on average, range 1.29 ~ 776). The connection between these volumes and the exposure sparsity in Extended Data Figure 4c is more subtle, and a direct comparison between the two is not straightforward in our opinion. In more detail, the volume in Figure R13 is a property of the signature matrix in each dataset. Different tumor types have different signatures, resulting in different volumes. For example, the volume could be small simply because of the presence of similar signatures. By comparison, the sparsity in Extended Data Figure 4c is a property of the exposure matrix in each dataset. The exposure sparsity characterizes how tightly the samples distribute within the simplex formed by the

signatures, but does not reflect the volume of the simplex itself. Therefore, we do not expect a correspondence between the volumes in **Figure R13** and the exposure sparsity in Extended Data Figure 4c.

Figure R13. Comparison of the volumes spanned by NMF and mvNMF solutions on PCAWG data. Each dot represents a tumor type. Volumes of *de novo* signatures extracted by NMF and mvNMF were plotted on the y-axis and x-axis, respectively. Since volume is only comparable between simplices of the same dimension, the NMF solution was forced to have the same number of signatures as the mvNMF solution for the corresponding tumor type. Dashed line represents the diagonal.

Reconstruction error

8. How does mvNMF perform w.r.t. reconstruction errors of the original matrix in comparison to NMF?

We compared the reconstruction errors of NMF and mvNMF solutions on real PCAWG data (**Figure R14**). In **Figure R14a**, we plotted the per-sample reconstruction errors of NMF against those of mvNMF. Most of the data points were close to the diagonal, suggesting that NMF and mvNMF resulted in similar reconstruction errors (Pearson $r = 0.88$, $p < 1e-300$). Slightly more than half (57%) of the samples had larger reconstruction errors from mvNMF. This is as expected, since mvNMF sacrificed some of the reconstruction accuracy for a smaller volume (note that a small reconstruction error is not the sole indicator of a good solution, as we illustrated in Figure 2a – as long as the convex cone encloses the data points, the reconstruction error is zero). However, the difference in reconstruction errors was not statistically significant between NMF and mvNMF (**Figure R14b**). This behavior was guaranteed by our method for selecting the hyperparameter that controls the volume regularization strength in mvNMF. Specifically, we selected the strongest volume regularization as long as the reconstruction error was not significantly worse when compared to the NMF solution (see Methods Section 4.3.1 “Selecting the regularization parameter”).

Figure R14. Comparison of reconstruction errors of NMF and mvNMF solutions on PCAWG data. (a) Per-sample reconstruction errors of NMF vs. mvNMF solutions. Dashed line represents the diagonal. To ensure a fair comparison, the NMF solution was forced to have the same number of signatures as the mvNMF solution for the corresponding tumor type. **(b)** Distribution of the per-sample reconstruction errors for NMF and mvNMF solutions separately. *P*-values from four different statistical tests were annotated for testing the difference between the two distributions.

De-novo signature detection in comparison to SigProfileExtractor

9. The poor performance of SigProfilerExtractor is quite surprising. In Fig. 3, SigProfilerExtractor only detects three de-novo signatures which to me seems like some parameter was chosen incorrectly (or rather the default parameters were a poor choice). Due to the poor performance of SigProfilerExtractor, I would strongly advise that more competing frameworks are compared here. In the SigProfilerExtractor paper, they compared their framework to seven other ones: SigProfiler_PCAWG, SignatureAnalyzer, SigneR, MutationalPatterns, MutSpec, SomaticSignatures, and SignatureTools. At least some more tools should be compared here.

In the Skin-Melanoma example presented in Figure 3a, we used default parameters for SigProfilerExtractor, and only three *de novo* signatures were identified. In **Figure R15**, we show the plot produced by SigProfilerExtractor for selecting the number of signatures in this example. In our opinion, even with manual selection based on visual inspection of **Figure R15**, it is unlikely that one would select the number of signatures to be greater than 3, since there was a clear drop in average stability starting from 4. Therefore, we do not think that the poor performance of SigProfilerExtractor in this example could be rescued by a better choice of parameters (e.g., stability thresholds including “stability”, “min_stability”, and “combined_stability” used in SigProfilerExtractor). Even if a larger number of signatures can be achieved by some combination of these parameters, it will be hardly generalizable when the underlying truth is unknown. Rather, we

think that this poor performance is rooted in the intrinsic non-uniqueness and thus instability of NMF solutions. See Figure R17 in the response to Reviewer's comment 11 for more details.

Figure R15. Plot produced by SigProfilerExtractor for selecting the number of signatures in the example presented in Figure 3a. Mean sample cosine distance (similar to per-sample reconstruction error) and average stability (i.e., mean silhouette score of solutions from multiple replicate runs) were plotted against the number of signatures.

Nevertheless, we agree with the Reviewer that our manuscript will greatly benefit from comparing MuSiCal with more methods. We originally focused only on SigProfilerExtractor because it has been the most popular tool in the field and was shown to outperform a number of other existing tools in the SigProfilerExtractor paper [3]. In this revision, we have further compared to three other tools, signature.tools.lib [2, 5], SignatureAnalyzer [6-8], and Signer [9]. We selected these three additional tools based on the following considerations. Although numerous signature analysis tools have been developed, many of them differ only in implementation details and specific functionalities, such as plotting, support for different signature types, etc. Their core algorithms are based on two methods – NMF and Bayesian NMF (see the nice summary in Table 1 of [3]). We argue that it is redundant to compare tools that share the same underlying methodology but differ only in implementation details. Therefore, we selected these three additional tools which, together with SigProfilerExtractor, represented both NMF- and Bayesian NMF-based frameworks and appeared to be among the most popular in the field, especially in large landmark studies. Specifically, SigProfilerExtractor is based on a Python implementation of NMF, has been the single most popular tool, and was used in the landmark PCAWG signature paper; signature.tools.lib is based on an R implementation of NMF, included several methodological improvements [2], and was used in the recent landmark study of the Genomics England Cohort [5]; SignatureAnalyzer is based on a Python implementation of Bayesian NMF and was used in the landmark PCAWG signature paper; Signer is based on an R implementation of Bayesian NMF. In total, we have now benchmarked the performance of MuSiCal for *de novo* signature discovery against these four tools. Most other tools not included in the benchmark can be represented by at least one of these four included ones. For example, among the tools mentioned by the Reviewer, SigProfiler_PCAWG is an older version of SigProfilerExtractor; MutationalPatterns, MutSpec, and SomaticSignatures are all based on R implementations of NMF and thus redundant

with signature.tools.lib.

We performed the benchmark at two noise levels – 0% (only Poisson sampling noise, no additional Gaussian random noise) and 5% (Poisson sampling noise + 5% Gaussian random noise; see the response to Reviewer’s comment 5 for more details). At each noise level, every tool was applied to 250 synthetic datasets (25 tumor types × 10 replicates) corresponding to ~23k simulated genomes. The results were summarized in the new **Extended Data Figure 10** (shown below). MuSiCal significantly outperformed all four existing tools at both noise levels. Among the four existing tools, SigProfilerExtractor was the best,

consistent with the results in [3]. Of note, as suggested by the Reviewer in comment 10, we now use the entire precision-recall curve (PRC) and the area under PRC (auPRC) to compare the performances, instead of a single F1 score in our original submission (see response to Reviewer’s comment 10 below for more details).

Extended Data Figure 10. MuSiCal outperforms SigProfilerExtractor and three additional existing tools – signature.tools.lib, SignatureAnalyzer, and SigneR – for *de novo* signature discovery at different noise levels. (a) Area under precision-recall curve (auPRC) for all five tools at different noise levels. Random noise were added as in Extended Data Fig. 8. Each box in the box plot represents 250 synthetic datasets (25 tumor types × 10 replicates). auPRC was calculated for each dataset separately, as in Fig. 3b. ***: $p < 0.0005$. p -values were calculated with paired t-tests. Box plots indicate median (center line), upper and lower quartiles (box limits), and 1.5x interquartile range (whiskers). **(b)** Precision-recall curve (PRC) for all five tools at different noise levels. Each PRC represents the average result of 250 synthetic datasets (25 tumor types × 10 replicates), as in Fig. 3c. **(c)** Precision of all five tools averaged across all tumor types at

different noise levels. Recall was fixed at 0.9. Error bars indicate standard deviation over 10 replicates. **(d)** Recall of all five tools averaged across all tumor types at different noise levels. Precision was fixed at 0.9, corresponding to a false discovery rate (FDR) of 10%. Here, precision was fixed at a smaller value compared to Fig. 3e, Extended Data Fig. 7d, and Extended Data Fig. 8d, because a precision of 0.98 was never achieved by SignatureAnalyzer or SigneR in many cases. Error bars indicate standard deviation over 10 replicates.

10. The authors adjust the detection threshold to get the optimal F1 value for both methods compared. However, this procedure is not representative of the actual use case. If model A performs better than model B when both are perfectly tuned does not mean that model A is also better than model B when the hyperparameters are tuned in absence of the ground truth. Why isn't MuSiCal's own parameter tuning used here?

To clarify, we did not tune any parameters for the benchmark analysis shown in Figure 3 and the old Extended Data Figure 7cd (now replaced, see details below). We simply ran MuSiCal and SigProfilerExtractor for *de novo* signature discovery on the synthetic datasets with default parameters, and then used the Decomposition module from SigProfilerExtractor (again with default parameters) to match the *de novo* signatures identified by both MuSiCal and SigProfilerExtractor to the COSMIC catalog to obtain F1 scores. This procedure was described in the old Methods Section 4.3.2.

The Reviewer's comment refers to the old Extended Data Figure 7ab (now replaced, see details below). However, we would like to clarify that even here, we did not perform parameter tuning to select the best performing model. Here, the same *de novo* signature discovery results (from MuSiCal and SigProfilerExtractor with default parameters) were used as in Figure 3 and the old Extended Data Figure 7cd. The only difference was that in the old Extended Data Figure 7ab, the MuSiCal matching algorithm was used instead of the Decomposition module from SigProfilerExtractor. We performed this complementary analysis to show that the conclusion was not affected by the exact algorithm used for matching the *de novo* signatures to the COSMIC catalog. Therefore, even in the old Extended Data Figure 7ab, there was no parameter tuning in *de novo* signature discovery, which was the part we were trying to benchmark. In order to apply the MuSiCal matching algorithm, i.e., the likelihood-based sparse NNLS, a likelihood threshold needs to be specified. If the threshold was set too small, recall would be artificially large, and precision would be artificially small. Likewise, if the threshold was set too large, recall would be artificially small, and precision would be artificially large. We thus reasoned that it would be fair to report the largest F1 score by scanning the likelihood threshold in order to minimize the impact of the matching step on evaluating *de novo* signatures. Effectively, we were reporting the largest F1 score achieved by a precision-recall curve (PRC) (see explanations below and the example in the new Figure 3 shown below).

Therefore, in our opinion, the benchmark results on *de novo* signature discovery in comparison to SigProfilerExtractor were well grounded. Nevertheless, we do think that the presentation in our original submission was unnecessarily complicated. The Reviewer's comment also inspired us to

report the entire PRC and the area under PRC (auPRC) instead of a single F1 score. After all, the balance between precision and recall should be judged according to specific application scenarios, and the full information provided by the entire PRC should be retained. As a result, in this revision, we have now switched to reporting PRC and auPRC for benchmarks on *de novo* signature discovery (Figure 3, Extended Data Figures 7-10). The MuSiCal matching algorithm is now consistently used for matching the *de novo* signatures to the COSMIC catalog. The Decomposition module from SigProfilerExtractor is not used anymore, as it involves multiple (at least three) parameters, and thus a PRC could not be generated.

Below, we show the new **Figure 3**. In Figure 3a, we added the demonstration of how PRC and auPRC were obtained for the Skin-Melanoma example. It is clear from this example that what we reported in the old Extended Data Figure 7ab was the largest F1 score achieved by the PRC. The conclusion that MuSiCal outperforms SigProfilerExtractor still holds, with MuSiCal achieving larger auPRC than SigProfilerExtractor in 18/25 tumor types (Figure 3b). Averaged across all tumor types, MuSiCal achieved auPRC of 0.932, outperforming the auPRC of 0.895 by SigProfilerExtractor (Figure 3c). Finally, the use of the entire PRC allowed us to conclude that compared to SigProfilerExtractor, MuSiCal achieved better precision at the same recall, and better recall at the same precision (Figure 3c-e).

Figure 3. MuSiCal outperforms the state-of-the-art algorithm SigProfilerExtractor for *de novo* signature discovery. (a) An example based on Skin-Melanoma demonstrating the metric for evaluating the quality of *de novo* signature discovery. A synthetic dataset is simulated from 13 SBS signatures specific to Skin-Melanoma (percentages below signature names denote exposure strengths) and the exposure matrix produced by the PCAWG consortium. MuSiCal and SigProfilerExtractor are applied to derive *de novo* signatures, which are subsequently decomposed as non-negative mixtures of COSMIC signatures with likelihood-based sparse NNLS at different likelihood thresholds (see next section for more details). Precision and recall are then calculated at each threshold by comparing matched COSMIC signatures with the 13 true signatures, and auPRC is obtained. The matching result corresponding to the largest achieved F1 score is shown in the heatmap on the right. (b) Comparison of MuSiCal and SigProfilerExtractor for *de novo* signature discovery from synthetic datasets based on 25 PCAWG tumor types with at least 20 samples. 10 independent simulations are performed for each tumor type. Box plot shows auPRC for each individual dataset. Tumor types are sorted according to the mean auPRC gain by MuSiCal. (c) PRC of MuSiCal and SigProfilerExtractor averaged across all tumor types. (d) Precision of MuSiCal and

SigProfilerExtractor averaged across all tumor types. Recall is fixed at 0.9. Error bars indicate standard deviation from 10 independent simulations. **(e)** Recall of MuSiCal and SigProfilerExtractor averaged across all tumor types. Precision is fixed at 0.98, corresponding to an FDR of 2%. Error bars indicate standard deviation from 10 independent simulations. **(f)** Cosine reconstruction errors for MuSiCal- and SigProfilerExtractor-derived *de novo* signatures. Each *de novo* signature is decomposed into a non-negative mixture of the true underlying signatures with NNLS. Cosine distance is then calculated between the reconstructed and the original *de novo* signature. *: $p < 0.05$. **: $p < 0.005$. ***: $p < 0.0005$. p -values are calculated with paired t-tests in (b) and t-tests in (f). Box plots indicate median (center line), upper and lower quartiles (box limits), and 1.5x interquartile range (whiskers).

11. The metric used in the comparison (precision, recall and the F-score) might be less than ideal for the problem at hand. Recall is a measure for "completeness" of the results but when COSMIC signatures are present with as low as 0.034 (SBS2 in Fig. 3a) one can hardly speak about "recovering SBS2". As the Precision does almost not change for any of the cancer types (see Fig 3b) all of the F1 gain likely comes from the Recall gains. In this context it seems that the main advantage of MuSiCal over SigProfilerExtractor is that it recovers more signatures and therefore has a higher chance of recovering COSMIC signatures at a low percentage. Also this comparison should be repeated where SigProfilerExtractor is forced to have the same number of signatures as MuSiCal to see that MuSiCal is better in general and not just better at choosing the number of signatures.

Several issues are involved in this comment. We elaborate on each of them below.

First, it is important to point out that a small NNLS weight (plotted in the heatmaps in Figure 3a) does not necessarily mean that the corresponding signature is there in proportionally small quantity. This disconnect is slightly unintuitive but crucial. We illustrate it using the example in **Figure R16** shown below. We consider SBS2 and SBS40 and mix the two signatures with a linear coefficient β . Specifically, the mixed signature $w = \beta \times \text{SBS2} + (1 - \beta) \times \text{SBS40}$. Note that if NNLS is used to decompose w into nonnegative linear combinations of SBS2 and SBS40, β will be exactly the output coefficient (i.e., the NNLS weight) for SBS2. In **Figure R16b**, we plotted the mixed signature at three relatively small β values as examples. Visually, SBS2 was clearly discernible even at $\beta = 0.05$, and the mixed signature was almost dominated by SBS2 already at $\beta = 0.15$. To be more quantitative, in **Figure R16c**, we scanned β from 0 to 1 and plotted the cosine similarity of the mixed signature w to SBS2 and SBS40 separately.

Importantly, the mixed signature started to be more similar to SBS2 at a β value much smaller than 0.5. The reason behind this seemingly unintuitive observation is that SBS2 has a very defined shape with peaks only at T[C>T]N mutations, whereas SBS40 is flat. In **Figure R16d**, we plotted the fraction of T[C>T]N mutations contributed by SBS2 in the mixed signature at β values from 0 to 1. Even at β values much smaller than 0.5, the majority of T[C>T]N mutations were already coming from SBS2. Therefore, we can appreciate from this example that a small NNLS weight does not necessarily mean that the corresponding signature is too weak to be detected, because the NNLS weight quantifies the total contribution from all mutation types defined in the signature (e.g., 4 types for SBS2 vs. ~96 types for SBS40), whereas the detection limit is really a per-mutation-type concept and depends on

the shape of the signature. This point was explained in Results Section 2.3 in our manuscript:

... different mutational signatures require different numbers of mutations to provide confidence in their presence. Some signatures may be statistically significant even with a small number of mutations, whereas others may not be significant even with many mutations. For example, 20 C>T mutations at CpGs may be sufficient to claim the presence of SBS1, which has prominent weights only at 4 of the 96 SBS categories, whereas the same number of mutations can provide only limited evidence for the flatter SBS3. ...

In conclusion, we believe that precision and recall are still suitable metrics for evaluating the quality of *de novo* signature discovery.

Figure R16. An example illustrating that a small NNLS weight does not necessarily mean that the corresponding signature is too weak to be detected. (a) Spectra of COSMIC SBS2 and SBS40. **(b)** Mixed signatures were created by combining SBS2 and SBS40 with a linear coefficient β , which corresponds to the NNLS weight of SBS2. Mixed signatures at β values of 0.05, 0.15, and 0.25 were shown. **(c)** Cosine similarity of the mixed signature to SBS2 and SBS40 separately when β was scanned from 0 to 1. β values shown in (b) were highlighted. **(d)** Fraction of T[C>T]N mutations contributed by SBS2 in the mixed signature when β was scanned from 0 to 1. β values shown in (b) were highlighted.

Second, in our original submission, the conclusion was indeed that MuSiCal mainly improves recall (i.e., sensitivity) of *de novo* signature discovery while maintaining similar precision in comparison to SigProfilerExtractor, as pointed out by the Reviewer. However, upon further reflection, we now believe that this may have been an incomplete view due to reporting a single F1 score (see, for example, the Skin-Melanoma example in the new Figure 3a shown earlier). As suggested by the Reviewer's comment 10 above and described in the response thereof, we now use the entire PRC and report auPRC for comparison. Now, the full information retained in the PRC allowed us to

conclude that MuSiCal outperforms SigProfilerExtractor overall in both precision and recall (new Figure 3c). Specifically, MuSiCal achieved better precision at the same recall (new Figure 3d) and better recall at the same precision (new Figure 3e).

Third, the ability or inability to choose a proper number of signatures (denoted as r) is closely related to the underlying algorithm used for *de novo* signature discovery. We have shown in Figure R15 (see the response to Reviewer's comment 9) that even with manual selection, it is unlikely to select r to be greater than 3 for the Skin-Melanoma example according to SigProfilerExtractor results, since there was a clear drop in average stability starting from $r = 4$. We believe that this poor performance of SigProfilerExtractor is because of the intrinsic non-uniqueness and thus instability of NMF solutions which SigProfilerExtractor relies on for *de novo* signature discovery. We further elaborate on this point in **Figure R17**, where we show a side-by-side comparison of mvNMF and NMF solutions in terms of their average stability (i.e., mean silhouette score) and reconstruction error for the same Skin-Melanoma example. Compared to NMF solutions which started to show a significant drop in average stability from $r = 4$, mvNMF solutions maintained a high stability up to $r = 9$ or $r = 10$. It is this stability of mvNMF solutions that allowed a larger r to be selected by MuSiCal in this example. Therefore, even if MuSiCal is only better at choosing the number of signatures (it is in fact not the case, as we will show below), this ability is still deeply rooted in the mvNMF algorithm and could not be achieved with NMF-based *de novo* signature discovery.

Figure R17. Comparison of mvNMF (left) and NMF (right) solutions at different numbers of signatures for the example presented in Figure 3a. In each plot, average stability (i.e., mean silhouette score) was plotted on the y-axis on the left, and reconstruction error was plotted on the y-axis on the right. While mvNMF solutions maintained high stability up to 9 or 10 signatures, NMF solutions already had a significant drop in stability at 4 signatures. To ensure a fair comparison, our implementations within MuSiCal were used to derive both NMF and mvNMF solutions.

Finally, we agree with the Reviewer that it could still be informative to compare to SigProfilerExtractor when it is forced to have the same number of signatures as MuSiCal. We have now added this comparison to **Extended Data Figure 7** (shown below; this replaces the old Extended Data Figure 7).

Although the performance of SigProfilerExtractor was improved with this forced selection of number of signatures, it still significantly underperformed MuSiCal. We thus concluded that MuSiCal was indeed better in general and not just better at selecting the number of signatures.

Extended Data Figure 7. MuSiCal outperforms the state-of-the-art algorithm SigProfilerExtractor for *de novo* signature discovery. Here we further benchmark the performance of MuSiCal for *de novo* signature discovery against two variations of SigProfilerExtractor. In one variation, SigProfilerExtractor's built-in input normalization is turned on (-nx gmm), and the corresponding result is denoted as SigProfilerExtractor-norm. In the other variation, SigProfilerExtractor is forced to select the same number of signatures as MuSiCal, and the corresponding result is denoted as SigProfilerExtractor-forced. **(a)** Area under precision-recall curve (auPRC) for MuSiCal, SigProfilerExtractor, SigProfilerExtractor-norm, and SigProfilerExtractor-forced. Each box in the box plot represents 250 synthetic datasets (25 tumor types × 10 replicates). auPRC was calculated for each dataset separately, as in Fig. 3b. ***: $p < 0.0005$. p -values were calculated with paired t-tests. Box plots indicate median (center line), upper and lower quartiles (box limits), and 1.5x interquartile range (whiskers). **(b)** Precision-recall curve (PRC) for the four algorithms. Each PRC represents the average result of 250 synthetic datasets (25 tumor types × 10 replicates), as in Fig. 3c. **(c)** Precision of the four algorithms averaged across all tumor types. Recall was fixed at 0.9. Error bars indicate standard deviation over 10 replicates. **(d)** Recall of the four algorithms averaged across all tumor types. Precision was fixed at 0.98, corresponding to a false discovery rate (FDR) of 2%. Error bars indicate standard deviation over 10 replicates.

12. Some of the refitting results seem strange. For example the MuSiCal Signature Sig 7 looks very much like SBS17b but is actually assigned a higher similarity to SBS40. According to Fig 3b, Skin-Melanoma has the highest difference between MuSiCal and SigProfilerExtractor. It would be interesting to see Fig 3a recreated for all cancer types in which MuSiCal outperforms SigProfilerExtractor.

The colors and numbers plotted in the heatmaps of Figure 3a represented NNLS weights, which could not be interpreted as similarity. This counter-intuitive disconnection between similarity and NNLS weight has been explained in detail in the response to Reviewer's comment 11 above (see

Figure R16).

As suggested by the Reviewer in comment 10, we now use the entire PRC and auPRC to compare the performances for *de novo* signature discovery, instead of a single F1 score in our original submission (see response to Reviewer's comment 10 above for more details). As a result, Skin-Melanoma is not the tumor type with the largest improvement achieved by MuSiCal anymore (see the new Figure 3b shown in the response to Reviewer's comment 10 above). Nevertheless, we have now added 24 Supplementary Figures (not copied here to save space) to show examples as in Figure 3a but for all the other 24 tumor types.

13. It seems strange to me that for some cancers, the F1 value has zero variance for both MuSiCal and SigProfilerExtractor. Here it would also be interesting to see the Fig 3a-like plot for these examples.

In the new Figure 3b, the auPRC has zero variance for both MuSiCal and SigProfilerExtractor in two tumor types. Both algorithms performed perfectly in these two tumor types for all replicates, resulting in perfect auPRC values. Figure 3a-like plots for all tumor types are now provided in the Supplementary Figures (not copied here to save space).

Computation time

14. In the discussion it is mentioned that "the scalability of MuSiCal could be improved." However, it is not stated in the manuscript what computational costs are expected. Therefore, the time and memory cost for all substeps should be included based on the application on the PCAWG data. I myself tried to run MuSiCal's *de-novo* detection of signatures on a datasets with ~2.500 samples and 40 features on a high computing cluster with 32 cores (parameters taken from the example notebook of the GitHub repository), which took multiple days to complete.

We now provide the computational cost of MuSiCal in comparison to SigProfilerExtractor in the new **Extended Data Figure 24** (shown below). MuSiCal with mvNMF was slightly slower than SigProfilerExtractor, but required considerably less memory. MuSiCal with NMF was significantly faster than SigProfilerExtractor. As we have pointed out in the Discussion, we are aware that MuSiCal with mvNMF could potentially take significant amount of time to run for large datasets. In fact, efficient

mutational signature discovery from large datasets is still an unsolved problem in the field. For example, even with 64 CPUs and GPU boosting, SigProfilerExtractor also requires multiple days (2 days and 7 hours) to extract SBS signatures from the entire PCAWG dataset (numbers taken from the SigProfilerExtractor paper [3]). We are actively working on more efficient algorithms for solving the optimization problem in mvNMF and hope to make MuSiCal more scalable in the future. Of note, Extended Data Figure 24 shows the computational cost for *de novo* signature discovery. The computational costs of matching and refitting steps were negligible compared to *de novo* signature discovery. The *in silico* optimization step requires rerunning *de novo* signature discovery for a grid of thresholds. But during this grid search, there is no need to select the regularization parameter in

mvNMF or the number of signatures (as they are both fixed), which are the most time-consuming calculations. Thus, the computational cost for *in silico* optimization is also small compared to *de novo* signature discovery when parallelized properly.

Extended Data Figure 24. Computational cost of MuSiCal in comparison to SigProfilerExtractor. MuSiCal (in both mvNMF and NMF modes) and SigProfilerExtractor (based on NMF) were run on each PCAWG tumor type separately for *de novo* signature discovery, and the corresponding computation time and memory usage were shown. MuSiCal was run with 10 CPUs on a high-performance cluster, and SigProfilerExtractor was run with 12 CPUs. **(a)** Computation time (in seconds) for *de novo* signature discovery was plotted against the number of samples. Each dot represents a PCAWG tumor type. Solid lines represent linear fits in the log space, i.e., $\log(t) \sim \log(n)$, where t denotes computation time, and n number of samples. MuSiCal with mvNMF and SigProfilerExtractor were comparable in computation time for PCAWG tumor types, although MuSiCal with mvNMF scaled slightly worse ($t \propto n^{0.65}$ for MuSiCal with mvNMF and $t \propto n^{0.57}$ for SigProfilerExtractor). MuSiCal with NMF was much faster and scaled the best ($t \propto n^{0.39}$). **(b)** Same as panel (a) but for memory usage (in GB). MuSiCal with either mvNMF or NMF required considerably less memory than SigProfilerExtractor.

```
## Benchmarking the likelihood-based sparse NNLS
15. Same issue as above (point 10 in De-novo signature detection in comparison to SigProfileExtractor):
Selecting the best threshold does not reflect the real-world usage.
```

Similar to our response to Reviewer’s comment 10, we now provide PRC and auPRC without threshold selection for the benchmark results of refitting in the new **Extended Data Figure 12** (shown below). MuSiCal (with likelihood-based sparse NNLS) outperformed thresholded NNLS overall, achieving better auPRC (Extended Data Figure 12a). Compared to SigProfilerExtractor and sigLASSO, MuSiCal

achieved better precision at the same recall and better recall at the same precision (Extended Data Figure 12a, c, d).

Extended Data Figure 12. MuSiCal (with likelihood-based sparse NNLS) outperforms thresholded NNLS, SigProfilerExtractor Decomposition, and sigLASSO for refitting. (a) Precision and recall of all four algorithms based on the same benchmark results as in Fig. 4c, d. For MuSiCal and thresholded NNLS, the entire precision-recall curve (PRC) was shown by scanning the corresponding threshold. For SigProfilerExtractor Decomposition and sigLASSO, a PRC could not be generated since multiple parameters were involved in the algorithm, and thus the precision and recall achieved with default parameters were shown. **(b)** The largest achieved F1 score of MuSiCal and thresholded NNLS in comparison to the F1 score of SigProfilerExtractor Decomposition and sigLASSO achieved with default parameters. **(c)** Precision of MuSiCal and thresholded NNLS at the same recall as SigProfilerExtractor Decomposition (left) and recall of MuSiCal and thresholded NNLS at the same precision as SigProfilerExtractor Decomposition (right). **(d)** Precision of MuSiCal and thresholded NNLS at the same recall as sigLASSO (left) and recall of MuSiCal and thresholded NNLS at the same precision as sigLASSO (right). Error bars indicate standard deviation from 10 independent simulations.

16. Furthermore, the metric of "correct support discovery rate" seems not ideal as it only regards perfect fits. If a model gets 90% correct signatures it is counted the same as a model that gets only 10% correct. This seems wrong to me as for real-life examples we can never really expect 100% correct signature retrieval. As precision and recall was used in the previous section I am unsure why the authors decided not to use it here. Especially because it seems to make more sense than in Fig 3. Since this is a threshold-adjustment problem, a precision-recall curve and the respective area under the curve seems to be a good measure of the performance without prior knowledge about the threshold. Therefore I would like to see the same analysis but using the metric of precision recall. A F1 score for the perfect threshold is ok but then the area under the precision recall curve should at least be included in the supplement to show that MuSiCal is better even without prior knowledge of the perfect threshold. Also since MuSiCal has a built in parameter tuning algorithm why isn't it used here?

We considered refitting and matching as a support discovery problem or a sparse recovery problem as in the statistics literature (see, e.g., [10]), where it is common to report the "success rate", i.e., the "correct support discovery rate". But we agree with the Reviewer that this metric does not capture how wrong the solution is, and that precision and recall could provide additional information. We have now added Extended Data Figure 12 (shown in the response to Reviewer's comment 16 above) to compare different methods using precision and recall. Specifically, we included both the largest achieved F1 score (Extended Data Figure 12b) and the entire PRC (Extended Data Figure 12a, with auPRC annotated).

Likelihood-based sparse NNLS performed the best with all different measures. We did not use *in silico* optimization in this analysis because we wanted to ensure a fair comparison across different algorithms (*in silico* optimization makes use of information obtained from *de novo* signature discovery), and we hoped to show that likelihood-based sparse NNLS by itself was already a better algorithm for refitting and matching.

in silico validation

17. How does the (simul - data) score compare to the mere reconstruction error? So for the PCAWG data, how is the reconstruction error and the (simul - data) score related? Also mention what are the problems with using just the reconstruction error.

Following the notation used in Figure 5a, reconstruction error refers to the distance between the raw data matrix $X_{i \times \#}$ and the reconstructed matrix $X_{\$} = W_{\$}H_{\$}$ from the sparse signature assignment results $W_{\$}$ and $H_{\$}$. The key to our *in silico* validation approach is an additional step of *de novo* signature discovery on the synthetic data matrix $X_{\% \& \cdot \cdot}$ simulated from $W_{\$}$ and $H_{\$}$, resulting in $W_{\% \& \cdot \cdot}$ and $H_{\% \& \cdot \cdot}$. Compared to the reconstruction error which could be interpreted as the sum of independent per-sample errors, the difference between $(W_{i \times \#}, H_{i \times \#})$ and $(W_{\% \& \cdot \cdot}, H_{\% \& \cdot \cdot})$ captures more intricate structures of the dataset (such as correlations between different signatures) and more subtle problems in the matching/refitting steps (such as which signature combinations are more suitable). To be more specific, consider the over-assignment problem of

SBS40. As we have shown in the manuscript, SBS40 was frequently over-assigned in many tumor types by the PCAWG consortium because of its flat spectrum and overlap with other signatures. In fact, the addition of SBS40 to the signature assignment (W_s, H_s) would not do any harm to reconstruction error, since more available signatures could only result in a better fit for the raw data matrix. Thus reconstruction error by itself could not inform over-assignment problems. By contrast, the over-assignment of SBS40 will be revealed by *in silico* validation when a strong SBS40 component is present in $W_{\&cdot}$ but not in $W_{!\"\#}$ (see Figure 5 and Figure 7a).

18. The term “PCAWG” is a bit ambiguous when referring to the assignment (e.g. W^{PCAWG} or in Fig 5b). This got me confused initially as the dataset itself can be referred to as PCAWG and W_{data} as well as W_{simul} are of course also based on the PCAWG data. So maybe rename this to “original assignment”.

We fully understand that the term “PCAWG” can be a little ambiguous as it refers to the “PCAWG dataset” or the “results obtained by the PCAWG consortium”. We have thus been careful in explicitly spelling out these terms in the text to avoid the confusion as much as possible. For the notations in Figure 5, we would still prefer to use W^{PCAWG} to contrast with W^{MuSiCal} such that it is more clear that we are comparing signature assignments obtained by the PCAWG consortium with signature assignments obtained by MuSiCal. This notation would also be consistent with similar comparisons presented in other figures. We felt that “original” may be more ambiguous here as it may be understood as “before simulation”.

19. Why is the glioblastoma dataset chosen here? Without an appropriate explanation this seems like cherry-picking. It is fine to use a single tumor type for demonstration purposes but this should be repeated for multiple tumor types.

The glioblastoma dataset was presented in Figure 5 only to illustrate the *in silico* validation method with a concrete example. It was chosen because it was relatively small and simple. The *in silico* validation results for all tumor types were already presented in Figure 7a for SBS signatures and Extended Data Figure 21 (previously Extended Data Figure 17) for ID signatures.

Preprocessing in the re-analysis of PCAWG

20. In the Methods it is stated that “the suggested stratification needs to be investigated in a case-specific manner in order to maximize the benefit for signature discovery (see below for detailed approach taken for reanalysis of PCAWG data)” but in the PCAWG section it is only described as “we stratified samples into subsets with distinct signature compositions with hierarchical clustering”. Does this mean the normal preprocessing step based on the exposure matrix H was used? What did the “case-specific investigation” look like?

We apologize for being unclear here. We have modified the section “Implementation of preprocessing” and provided additional information (shown below).

After an initial step of *de novo* signature discovery within each tumor type, the Gini-based outlier

removal was applied, removing on average 2.3% of samples (ranging between 0% to 7.0% in different tumor types). For Kidney.RCC, Liver.HCC, and Skin.Melanoma, we further stratified samples into subsets with distinct signature compositions using hierarchical clustering on the exposure matrix *H* obtained from the initial *de novo* signature discovery. The distinct clusters within these tumor types were defined by several hypermutator processes including aristolochic acid, aflatoxin, and UV exposures. Note that we separated the MMRD and POLE-exo mutant samples before this stage, although preprocessing can be used to stratify these samples if they had not been isolated beforehand (Extended Data Fig. 10a-c).

By “case-specific investigation”, we meant that one has to compare the *de novo* signature discovery results before and after cohort stratification for each dataset separately, since splitting the dataset into clusters is not always helpful. In more detail, the motivation of cohort stratification is that it reduces data complexity and thus allows more subtle signatures within subsets to be discovered. For example, by removing 5 MMRD samples from a dataset of 50 samples, the size of the dataset is only minimally reduced (from 50 to 45), while the complexity of the dataset is substantially reduced with only MMRP samples remaining. In this example, it is more likely that stratification will be helpful, as more signatures in the MMRP samples will be discovered, and the cross-contamination of MMRD signatures to MMRP signatures will be reduced. On the contrary, cohort stratification can significantly reduce the power of

de novo signature discovery when much smaller subsets are produced (e.g., a dataset of 50 samples split into two subsets with 25 samples each). In these cases, fewer signatures may actually be discovered due to loss of power, and it would be more helpful not to split the cohort. Specifically for the PCAWG reanalysis of SBS signatures, on top of separating MMRP and MMRD/POLE-exo samples at the very beginning, further cohort stratification was only done in Kidney.RCC, Liver.HCC, and Skin.Melanoma, where the distinct subsets were defined by hypermutator processes such as aristolochic acid, aflatoxin, and UV exposures.

21. Furthermore, since the stratification of the input greatly improves the sensitivity of the signature detection I would be interested as to how much of the newly discovered signatures can be attributed to this preprocessing. If this stratification is turned off, does MuSiCal still find the new ID and SBS signatures? If it doesn't, that needs to be made clear in the text.

Most of the new SBS signatures (SBS95-SBS98) were discovered in MMRD tumors, which were separated from the other samples at the very beginning using a dedicated method described in Methods Section 4.5.1 (previously Methods Section 4.4.1) and Extended Data Figure 25a, b (previously Extended Data Figure 20a, b). For these signatures, the joint analysis of MMRD tumors was likely to be crucial for their discovery. We have already described the MMRD origin of these signatures in both the main text and Extended Data Figure 16a (previously Extended Data Figure 12a). The other new SBS signatures -- SBS99 and SBS100 -- were discovered in Prostate.AdenoCA and Breast.AdenoCA/Ovary.AdenoCA respectively, where cohort stratification was not performed (see the response to Reviewer's comment 20 above for more details). Thus, the use of mvNMF was more

likely to have facilitated the discovery of these two signatures. For ID signatures, we did not rely on a tumor type-specific analysis for *de novo* signature discovery because of the lower number of indels per sample. Instead, a joint analysis (after removing MMRD samples) was performed to increase power, as described in Methods Section 4.5.2 (previously Methods Section 4.4.2). Therefore, the discovery of new ID signatures was also more likely to be attributed to the application of mvNMF.

New signature ID11b

22. The newly discovered signature ID11b is very close to the established ID11 (cosine of 0.96). ID11b correlates both with SBS4 and ID3 linking it to tobacco smoke. Does this correlation also exist to the standard ID11 signature (i.e. when using the original set of signatures)? If this is the case, then ID11b and its relation to tobacco smoke is not necessarily a new discovery but seems like a false positive. If it is the case then this point should be made in the text, i.e. “through the new stratification of ID11 into ID11a and ID11b, we have discovered a previously unknown etiology”. Same for Extended Data Fig. 15: In the original PCAWG assignment, does the inclusion of the original ID11 also explain the observed T insertions after T. One would expect so as both ID11a and ID11b have strong T following T insertions.

MuSiCal discovered two ID signatures (ID11a and ID11b) that were similar to the COSMIC ID11. ID11a was more similar to COSMIC ID11 (cosine similarity = 0.97), with prominent weights in both 1-bp C and T insertions. ID11b was less similar to COSMIC ID11 (cosine similarity = 0.94) and was dominated by 1-bp T insertions (**Figure R18a**). We have shown in Extended Data Figure 19b (previously Extended Data Figure 15b) that ID11b correlated with both SBS4 and ID3, suggesting a potential association with tobacco smoking. This correlation was not observed for ID11a according to signature assignments obtained by MuSiCal (**Figure R18b**), nor was it observed for COSMIC ID11 according to signature assignments obtained by the PCAWG consortium (**Figure R18c**). Further, we have shown in Extended Data Figure 19d (previously Extended Data Figure 15d) that the observed enrichment of 1-bp T insertions following a T in

smoking-related samples could only be explained with the addition of ID11b. Specifically, the scatter plot on the right-hand side of Extended Data Figure 19d was based on signature assignments obtained by the PCAWG consortium, which already included COSMIC ID11. Thus COSMIC ID11 alone could not explain the observed enrichment of 1-bp T insertions following a T. Although COSMIC ID11 also has a strong peak at this particular mutation type, the simultaneous presence of the strong C insertion part in COSMIC ID11 prevented it from being assigned to these samples. In **Figure R18d**, we plotted the average ID spectrum of smoking-related samples. The average spectrum showed a clear component of 1-bp T insertions similar to ID11b, but did not have any considerable weight at 1-bp C insertions, unlike COSMIC ID11. Finally, previous experimental data also showed that tobacco smoking produced only ID11b-like T insertions, but not any COSMIC ID11-like C insertions (Extended Data Figure 19c). In conclusion, all these results strongly suggest that ID11b is a true smoking-related signature distinct from COSMIC ID11, but not a false positive (e.g., an algorithmic split of COSMIC ID11). The apparent similarity between ID11b and COSMIC ID11 might be coincidental, since the 83-channel ID signature definition is rather degenerate. We believe that another unknown mutational process potentially underlies COSMIC ID11 (i.e., MuSiCal ID11a).

Figure R18. Additional analyses related to ID11a, ID11b, and COSMIC ID11. (a) Spectra of MuSiCal-derived ID11a and ID11b, and COSMIC ID11. (b) The per-sample exposure of ID11a was plotted against the per-sample exposure of SBS4/SBS92 (left) and ID3 (right). Signature assignments obtained by MuSiCal were used. Pearson correlation coefficient (r) and the corresponding p -value were annotated. “nan” indicates that ID11a was not assigned to that tumor type, and thus the correlation could not be calculated. Of note, we now use both SBS4 and SBS92 to characterize smoking-related SBS signatures. SBS92 was recently discovered as a new smoking-related signature in the SigProfilerExtractor paper [3]. (c) Same as (b) but for COSMIC ID11. Signature assignments obtained by the PCAWG consortium were used. (d) Average ID spectrum of smoking-related samples. The y-axis was plotted in the full scale in the left panel and zoomed in in the right panel to better show 1-bp C and T insertions.

Minor points

We thank the reviewer for reading our manuscript thoroughly and the attention to detail. We have addressed all these minor points and improved the presentation of relevant figures.

* Section 2.1: "any cone that encloses the data points" -> I think it has to be "any convex cone" or "any linear cone"

“Convex cone” is indeed more accurate. We have modified relevant sentences such that “convex cone” is now consistently used.

* Fig 1
 * missing explanation for purple and dashed/orange arrows
 * after reading the paper this visual abstract became quite clear for me but on first glance I didn't understand the four boxes

We have added the following explanation in the caption of Figure 1:

Solid purple arrows represent a typical workflow of mutational signature analysis. Dashed orange arrows represent that some steps in the workflow potentially require repeating an earlier step (see Results).

We agree with the Reviewer that the method highlights within the four boxes become more clear after reading the paper. We hope that the visual abstract in Figure 1 is still helpful at the beginning of the paper both to clarify the relation among different steps in mutational signature analysis and to serve as a teaser for new algorithms to be introduced later in Results sections.

* Fig 2a:
 * state the meaning of the x, y and z axis. I understand this figure is inspired from Ref. 46 but the figure might not be clear for a general audience.
 * I assume that the fact that all grey points are along the dashed lines is due to the sparseness of the exposure matrix. If this is the case that should be mentioned in the figure caption.
 * The grey circles are not in the legend

The meaning of the x-, y-, and z-axis is now explicitly stated in the caption:

Synthetic samples are simulated from signatures with three mutation channels (represented by the three axes x, y, and z).

How random exposures were generated is now described in the caption:

In all panels, random exposures are generated from symmetric Dirichlet distributions with a concentration parameter of $\alpha = 0.1$, which is a representative value according to real exposure matrices obtained by the PCAWG consortium [14] (Extended Data Fig. 4).

The grey circles are now included in the legend.

* Fig 2b: Why are the triangles not filled like in Fig 2a?

We chose to use empty triangles in Figure 2b because here more replicate solutions were shown and thus there were more overlap. Using empty triangles helps to visualize each solution better.

* Fig 2c: Maybe mark the ones where vanilla NMF is better as it is hard to see (and especially hard to gauge how many are better/worse)

We have now explicitly marked the signatures where NMF outperformed mvNMF. In these cases, the cosine errors for both algorithms were much smaller than 0.001 and thus negligible.

- * Fig 3a:
- * The meaning of the percentages is not explained in the figure caption
- * Some entries are too big for their cells in the matrices

The percentages denote exposure strengths of corresponding signatures. This is now explained in the caption. The latter issue is also fixed now.

- * Fig 4a/b: X-axis ticklabels are off-center (shifted to the left)

The tick labels were not off-center, but right-aligned to the ticks. We have now moved the tick labels closer to the ticks for better visual matching.

- * Fig 5a
- * What are H_a and W_a? They are not mentioned in the text. I guess it should be W_s and H_s
- * This figure is not 100% self-explanatory. For example “r” should be written out as “number of signatures”, the orange arrow should say “simulate” and point to X_{simul}, the arrow under “Refit” should be in the same design as all the other arrows, “MuSiCal assignment” should maybe be “final assignment”
- * the dashed lines have varying alpha

We have improved this figure according to the Reviewer’s comments. Detailed modifications were listed below.

- There were no H_a or W_a in the figure.
- We preferred not to explicitly write out the meaning of “r” so that the figure is not overcrowded. We did write “See main text for details” in the caption and explained the meaning of “r” in the main text.
- We have now added “simulate” to the orange arrow and made it point to X_{simul}.
- We have changed the design of the arrow under “Refit” such that it is in the same design as other arrows.
- We have changed “MuSiCal assignment” to “Signature assignment”.
- The varying alpha in the dashed lines was due to overlap with the boxes above. We think that this varying alpha helps visualize different layers and thus retained it.

- * Fig 5b
- * This should state “Signature assignments for glioblastoma (N=?)” to make clear that this is just for a single tumor type
- * It is also very unclear for the rest of the Figure (c-f) if all of these results are for glioblastoma only.

We have now added the annotation “**b-f**: Example using CNS.GBM (n = 39)” in Figure 5 to make it more clear that the analyses presented here were based on glioblastoma samples.

* Ext Fig 10a: What does “reasonable k’s” mean?

It was our oversight not to explain what “reasonable k’s” means. We have now added the following text to the caption of Extended Data Figure 13a (previously Extended Data Figure 10a).

In panel (a), reasonable k means any k satisfying $\text{Gap}(k) \geq \text{Gap}(k + 1) - s_{k+1}$, where $\text{Gap}(k)$ denotes the gap statistic (indicated by dots in the plot) and s_k the dispersion due to simulation (indicated by error bars in the plot). The smallest reasonable k is chosen as the optimal k. See [54, 55] for more details.

Reviewer #3:

Remarks to the Author:

Jin et al present a new algorithm for mutational signature learning from cancer genome sequencing data.

Mutational signature analysis is an essential pillar of cancer genomics as it reveals the footprints of the mutational processes that contributed to cancer development. While the concept of deconvolving the patterns of mutations into individuals signatures is simple it is still a procedure that is not fully resolved and many commonly used algorithms suffer from over- and under-fitting. The authors’ contribution to the problem is a statistically principled algorithm that employs a minimal volume variant of the non-negative matrix factorisation (mvNMF) that has been the workhorse for mutational signature analysis in the last 9 years. mvNMF regularises the exposure matrix of the NMF product and can thus increase the stability of inference, as demonstrated in different context. The authors demonstrate the accuracy of their algorithm termed MuSiCal using simulations and apply it to 2,700 genomes from the PCAWG consortium.

I do commend the authors for developing a new, statistically sound inference framework, which is lacking in many other algorithms and which might be a critical step forward. The analysis is also clearly presented and some of the problems of NMF based signature analysis are well explained.

We thank the reviewer for the positive comments.

The biggest challenge with mutational signature analysis – and why some elements of it have become controversial – is that there is usually no ground truth. For that reason a great number of algorithms have been developed, each claiming superior performance based on circumstantial evidence.

A similar issue applies also to this study. The authors claim it ‘solves fundamental problems’, ‘resolve long-standing issues with the ambiguous ‘flat’ signatures’ and ‘give insights into signatures with unknown etiologies’. Further ‘We expect MuSiCal and the improved catalog to be a step towards establishing best practices for mutational signature analysis.’ These are all very bold, but ultimately and unfortunately somewhat empty statements in the absence of hard evidence.

As I don’t see that all claims are well supported I’m unfortunately not convinced that Musical presents more than an interesting new mathematical concept.

We understand the Reviewer’s concern about the lack of ground truth. However, similar problems exist in multiple fields of computational biology, and carefully designed simulation studies have been the workhorse in benchmarking different tools in these cases. In our manuscript, we generated synthetic datasets based on real PCAWG samples in order to capture a wide variety of complexities present in real-life data, such as diverse signature combinations, different signature strengths, correlated exposures, etc. As you can see from Reviewer 2’s comments and our responses, there are indeed numerous technical issues that must be handled carefully to make comparisons fair and realistic (especially regards to parameter tuning and metrics to measure errors), so that our claims are not confined to specific situations or based on synthetic data generated using the same statistical model assumed by the algorithm. We also don’t think that anyone would dispute, e.g., that the non-uniqueness of solution for NMF is problematic or that signature analysis results are confounded when flat signatures are present. Also please note that our concerns that prompted us to work on these issues are not simply academic — we’re engaged in analysis of multiple clinical trials data (especially for the flat HRD signature in PARP inhibitor and combination therapy trials) on which accurate determination of whether a specific signature is present informs treatment selection.

In this revision, we have further strengthened the benchmark studies by adding the following analyses (see detailed responses below):

- Additional benchmark analyses at different levels of random noise.
- Additional benchmark analyses with the spike-in of unknown spurious signatures.
- Additional comparisons to signature.tools.lib, SignatureAnalyzer, and SigneR, which, together with SigProfilerExtractor included in our original submission, covered both different underlying algorithms and the most popular tools in the field.
- Comparison of performances using more comprehensive metrics based on the precision-recall curve (PRC) and the area under PRC (auPRC) (suggested by Reviewer 2).

MuSiCal outperformed all competing methods under multiple performance metrics. We hope that these analyses together show convincingly that MuSiCal outperforms the current state-of-the-art.

With respect to analysis of real data, we also don't disagree with the Reviewer's concern that, in analyzing a large amount of data, it is possible to find *some* instance in which any method gives a result that appears to be more biologically meaningful. However, we do not think one should conclude that all such analyses therefore are invalid. In our case, one obvious example of improvement that would be appreciated by many in the community (including the other two reviewers) is the issue of flat signature

40. We also found new SBS/ID signatures and revised the spectra of existing ones by analyzing the same PCAWG data and for two of these signatures, we provided strong orthogonal evidence from existing experimental data (Extended Data Figure 15 for the TOP1-associated ID signature and Extended Data Figure 19 for the new tobacco smoking-associated ID signature). Further, we have revised the pan- cancer signature assignments in PCAWG data and provided evidence of improvements using our *in silico* validation method and the analysis of clock-like signatures (Figures 5-7). In this revision, we provide another example to demonstrate that MuSiCal-derived signature assignments could improve the accuracy of signature-based detection of clinically actionable mutational processes (see detailed responses below).

Nonetheless, given the disagreement on whether a problem is "fundamental", we have replaced the term "fundamental" with "major" in the revised version.

1. Claim: 'solves fundamental problems'

It is known that NMF has a tendency to overfit. For that reason most signature analyses employ various types of resampling approaches to establish solutions that do not overly depend on individual samples. More work is needed here to demonstrate that their approach genuinely outperforms others. The authors only compare their algorithm to one competing method, even though there is a plethora of them. It is an open secret in the mutational signature field that SigProfiler, when run out of the box, has a number of issues. This (<http://dx.doi.org/10.1038/s43018-020-0027-5>) for example, appears to be one of the better alternatives, but there is also MutationalPatterns, mmsig, MutSignatures, HDP, eMU, and many more. The authors would need to make a more comprehensive comparison to demonstrate that their algorithm is truly superior.

We agree with the Reviewer that more methods need to be included for a comprehensive comparison. We originally focused only on SigProfilerExtractor because it has been the most popular tool in the field and was shown to outperform a number of other existing tools in the SigProfilerExtractor paper [3]. It also underlies the COSMIC signature catalog and PCAWG results, which have had profound impact in all applications of signature analysis. In this revision, we have further compared MuSiCal with three other tools, signature.tools.lib [2, 5], SignatureAnalyzer [6-8], and Signer [9]. We selected these three additional tools based on the following considerations. Although numerous signature analysis tools have been developed, many of them differ only in implementation details and specific functionalities, such as plotting, support for different signature

types, etc. Their core algorithms are based on two methods – NMF and Bayesian NMF (see the nice summary in Table 1 of [3]). We argue that it is redundant to compare tools that share the same underlying methodology but differ only in implementation details.

Therefore, we selected these three additional tools which, together with SigProfilerExtractor, represented both NMF- and Bayesian NMF-based frameworks and appeared to be among the most popular in the field, especially in large landmark studies. Specifically, SigProfilerExtractor is based on a Python implementation of NMF, has been the single most popular tool, and was used in the landmark PCAWG signature paper; signature.tools.lib (also suggested by the Reviewer with the link to the paper) is based on an R implementation of NMF, included several methodological improvements [2], and was used in the recent landmark study of the Genomics England Cohort [5]; SignatureAnalyzer is based on a Python implementation of Bayesian NMF and was used in the landmark PCAWG signature paper; SigneR is based on an R implementation of Bayesian NMF. In total, we have now benchmarked the performance of MuSiCal for *de novo* signature discovery against these four tools. Most other tools not included in the benchmark can be represented by at least one of these four included ones. Among those mentioned by the Reviewer, MutationalPatterns and MutSignatures are both based on R implementations of NMF and thus redundant with signature.tools.lib; EMu is a relatively outdated algorithm closely related to NMF; mmsig does not support *de novo* signature discovery. HDP refers to an interesting recent application of hierarchical Dirichlet processes to *de novo* signature discovery. We tested the most recent tool mSigHdp [11], but found that for a medium-sized dataset of ~100 samples, it did not finish within 7 days using 10 CPUs on a high-performance cluster. It was thus impractical to include mSigHdp in our large-scale benchmark analysis involving ~46k samples (described below).

We performed the benchmark at two noise levels – 0% (only Poisson sampling noise, no additional Gaussian random noise) and 5% (Poisson sampling noise + 5% Gaussian random noise; see the response to Reviewer’s comment 2 below for more details). At each noise level, every tool was applied to 250 synthetic datasets (25 tumor types × 10 replicates) corresponding to ~23k simulated genomes. The results were summarized in the new **Extended Data Figure 10** (shown below). MuSiCal significantly outperformed all four existing tools at both noise levels. Among the four existing tools, SigProfilerExtractor was the best, consistent with the results in [3]. Of note, following the suggestion from Reviewer 2, we now use the entire precision-recall curve (PRC) and the area under PRC (auPRC) to compare the performances, instead of a single F1 score in our original submission (see response to Reviewer 2’s comment 10 for more details).

Extended Data Figure 10. MuSiCal outperforms SigProfilerExtractor and three additional existing tools – signature.tools.lib, SignatureAnalyzer, and SigneR – for *de novo* signature discovery at different noise levels. (a) Area under precision-recall curve (auPRC) for all five tools at different noise levels. Random noise were added as in Extended Data Fig. 8. Each box in the box plot represents 250 synthetic datasets (25 tumor types \times 10 replicates). auPRC was calculated for each dataset separately, as in Fig. 3b. *****: $p < 0.0005$.** p -values were calculated with paired t-tests. Box plots indicate median (center line), upper and lower quartiles (box limits), and 1.5x interquartile range (whiskers). **(b)** Precision-recall curve (PRC) for all five tools at different noise levels. Each PRC represents the average result of 250 synthetic datasets (25 tumor types \times 10 replicates), as in Fig. 3c. **(c)** Precision of all five tools averaged across all tumor types at different noise levels. Recall was fixed at 0.9. Error bars indicate standard deviation over 10 replicates. **(d)** Recall of all five tools averaged across all tumor types at different noise levels. Precision was fixed at 0.9, corresponding to a false discovery rate (FDR) of 10%. Here, precision was fixed at a smaller value compared to Fig. 3e, Extended Data Fig. 7d, and Extended Data Fig. 8d, because a precision of 0.98 was never achieved by SignatureAnalyzer or SigneR in many cases. Error bars indicate standard deviation over 10 replicates.

2. Claim: 'Our simulation studies demonstrate that MuSiCal outperforms state-of-the-art algorithms for both signature discovery and assignment.'

This may in part be a self-fulfilling prophecy. The authors essentially simulate according to their algorithms' model. But it is fundamentally unclear what the generative process is and it's likely to be way more complex than a matrix product of exposures and signature with a bit of Multinomial sampling on top. Mutagenesis has been shown to be influenced by many factors ranging from local DNA accessibility, histone modifications, repair efficiency and deficiency, and perhaps also mutagen metabolism. All these and many more factors are likely to influence the observed mutation spectra and complicate the analysis of mutational signatures. Understanding how the algorithm copes with a range of signature distortions and spiked-in outliers would be helpful.

To clarify, the simulation procedure was not biased towards our own algorithm *a priori*; if anything, it should be biased towards SigProfilerExtractor with which the PCAWG results were obtained in the first place, since we simulated data from signature and exposure matrices provided by the PCAWG consortium. All tools included in the benchmark analyses are based on some variations of NMF (plain NMF for SigProfilerExtractor and signature.tools.lib, Bayesian NMF for SignatureAnalyzer and Signer, mvNMF for MuSiCal), and thus they all assume the same underlying Poisson model for count data, which is equivalent to a multinomial sampling process. Our simulation procedure does not involve any form of volume regularization. The generated synthetic datasets should thus be considered as following the plain NMF or Bayesian NMF models more closely and give an disadvantage to mvNMF in MuSiCal. Therefore, the fact that MuSiCal still outperformed the other tools in these simulation studies confirms the intrinsic deficiency of NMF (i.e., non-uniqueness) and that reliance on the standard NMF by other algorithms are not likely to generate the best solutions. We do agree with the Reviewer that the true generative process of mutations is likely to be way more complex and should depend on many factors. We think a simulation procedure that captures these complexities would be essential for algorithms that model these factors specifically. But for algorithms that directly model the mutation count matrix (as all methods benchmarked here do), such complexities should be less relevant.

Nevertheless, to further demonstrate the robustness of MuSiCal, we have performed two additional benchmark analyses in this revision. In the first analysis, we added extra random Gaussian noise at different levels (1%, 2.5%, 5%, and 10%) to the simulated genomes to represent a range of random signature distortions. In more detail, we followed the SigProfilerExtractor paper [3] and resampled each element of the mutation count matrix (in our original synthetic datasets obtained from multinomial sampling) from a Gaussian distribution where the mean was equal to the original count and the standard deviation was equal to the original count multiplied by the noise level. The resulting matrix was then rounded to the closest integers, and negative values were set to 0. As a result, 4 additional synthetic datasets (corresponding to the 4 noise levels) were simulated for each of the 250 existing ones (25 tumor types \times 10 replicates). In total, 1,000 additional synthetic datasets were generated, corresponding to \sim 91k simulated genomes. We benchmarked the performance of MuSiCal against SigProfilerExtractor on these datasets and summarized the results in the new **Extended Data Figure 8** (shown below). While the performance of both tools slightly deteriorated as the noise level

increased, MuSiCal consistently outperformed SigProfilerExtractor at all noise levels.

Extended Data Figure 8. MuSiCal outperforms SigProfilerExtractor for *de novo* signature discovery at different noise levels. **(a)** Area under precision-recall curve (auPRC) for MuSiCal and SigProfilerExtractor at different noise levels. Each box in the box plot represents 250 synthetic datasets (25 tumor types \times 10 replicates). auPRC was calculated for each dataset separately, as in Fig. 3b. ***: $p < 0.0005$. p -values were calculated with paired t-tests. Box plots indicate median (center line), upper and lower quartiles (box limits), and 1.5x interquartile range (whiskers). **(b)** Precision-recall curve (PRC) for MuSiCal and SigProfilerExtractor at different noise levels. Each PRC represents the average result of 250 synthetic datasets (25 tumor types \times 10 replicates), as in Fig. 3c. **(c)** Precision of MuSiCal and SigProfilerExtractor averaged across all tumor types at different noise levels. Recall was fixed at 0.9. Error bars indicate standard deviation over 10 replicates. **(d)** Recall of MuSiCal and SigProfilerExtractor averaged across all tumor types at different noise levels. Precision was fixed at 0.98, corresponding to a false discovery rate (FDR) of 2%. The black triangle indicates the case where a precision of 0.98 was never achieved and the recall at the highest achieved precision was shown. Error bars indicate standard deviation over 10 replicates.

In addition, we performed a spike-in analysis to further investigate the robustness of MuSiCal when an unknown spurious signature is present. In more detail, additional mutations from COSMIC SBS48 were spiked-in to our original synthetic datasets. The number of spike-in mutations were equal to 5% of the mutational burden for each sample. SBS48 was chosen to represent a spurious signature because it is not present in the original synthetic dataset and is annotated in COSMIC as a possible sequencing artifact found in cancer samples that were subsequently blacklisted for poor quality of sequencing data. To pretend that SBS48 was unknown, we removed SBS48 from the COSMIC catalog when matching the *de novo* signatures and calculating precision and recall. The benchmark results

using these datasets were summarized in the new **Extended Data Figure 9** (shown below). Even with the spike-in of the unknown spurious signature, MuSiCal still significantly outperformed SigProfilerExtractor.

Extended Data Figure 9. MuSiCal outperforms SigProfilerExtractor for *de novo* signature discovery when an unknown spurious signature is present. (a) Area under precision-recall curve (auPRC) for MuSiCal and SigProfilerExtractor with and without spike-in. Each box in the box plot represents 250 synthetic datasets (25 tumor types \times 10 replicates). auPRC was calculated for each dataset separately, as in Fig. 3b. ***: $p < 0.0005$. p -values were calculated with paired t-tests. Box plots indicate median (center line), upper and lower quartiles (box limits), and 1.5x interquartile range (whiskers). (b) Precision-recall curve (PRC) for MuSiCal and SigProfilerExtractor with and without spike-in. Each PRC represents the average result of 250 synthetic datasets (25 tumor types \times 10 replicates), as in Fig. 3c. (c) Precision of MuSiCal and SigProfilerExtractor averaged across all tumor types with and without spike-in. Recall was fixed at 0.9. Error bars indicate standard deviation over 10 replicates. (d) Recall of MuSiCal and SigProfilerExtractor averaged across all tumor types with and without spike-in. Precision was fixed at 0.98, corresponding to a false discovery rate (FDR) of 2%. The black triangle indicates the case where a precision of 0.98 was never achieved and the recall at the highest achieved precision was shown. Error bars indicate standard deviation over 10 replicates.

3. Claim: 'give insights into signatures with unknown etiologies'.

Generally, the statements about new biological insights are very weakly supported, lacking experimental evidence. The existing references appear fairly handpicked rather than systematic. A convincing demonstration would be to show that the algorithm reproduces the correct exposure and signature matrixes from experimental systems with known mutagenic exposures

(e.g <http://dx.doi.org/10.1016/j.cell.2019.03.001> or <https://doi.org/10.1038/s41467-020-15912-7>).

Alternatively, one could show evidence for better correlation of chemotherapeutic exposures (<https://doi.org/10.1038/s41588-019-0525-5>) or DNA repair deficiency conditions (MMRD, POLE/D exonuc variants, BERD (MUTYH, NTHL1), etc).

In **Figure R2** (described in the response to Reviewer 1's comment 3, copied below for convenience), we provide an example to show that MuSiCal-derived signature assignment for HRD-associated

SBS3 was more consistent with the true HRD status. Specifically, we used MuSiCal-derived SBS3 assignment for HRD detection in breast, ovary, pancreas, and prostate tumors, where the HRDetect final classification was considered as true labels [2]. We compared the performance with NNLS-derived SBS3 assignment and the SBS3 input used in HRDetect [2]. As shown in **Figure R2**, MuSiCal-derived SBS3 assignment achieved the best sensitivity at the same false positive rate (FPR), demonstrating improved accuracy of MuSiCal-derived signature assignments. This result further suggests that the performance of multivariate classifiers such as HRDetect can be improved if MuSiCal-derived signature assignments are used as input, especially when MuSiCal is applied to other signature types as well (e.g., indel and SV).

We are actively working on incorporating MuSiCal-derived signature assignments into our previously developed multivariate classification framework SigMA [1] and extending it to detect other clinically actionable mutational processes. As to data from experimental systems with known mutagenic exposures suggested by the Reviewer, they represent a different type of data where each sample is a clean system defined by a single mutational process. The mutational spectra from these samples are usually taken as signatures directly after removing a background spectrum from control samples. Thus these data do not represent a suitable application scenario of MuSiCal or any other tools for *de novo* signature discovery or refitting. In our opinion, these well-controlled experimental data serve as valuable orthogonal validations for signatures discovered from tumor genomes or other diseased samples (as we have used in Extended Data Figures 15 and 19), but cannot be used as ground truth benchmark datasets in our analysis.

Figure R2. MuSiCal-derived SBS3 assignment outperforms the SBS3 feature used in HRDetect for detecting HRD. We compared the performance of SBS3 assignments derived from NNLS, signature.tools.lib, and MuSiCal for detecting HRD in breast, ovary, pancreas, and prostate tumors. The signature.tools.lib-derived SBS3 assignment was used as input to HRDetect in [2] (Degasperi et al. Nat Cancer 2020). The HRDetect final classification was considered as true labels. Sensitivity was shown at FPR values 5% and 10%.

Further, the authors investigate the correlations between SBS and ID signatures and claim that their method produces better correlations. If a high degree of correlations were the proof, then why do the authors run analysis of different parts of the data?

We do not think the two matters are contradictory to each other. We performed analysis on different parts (e.g., tumor types) of the data to improve accuracy and sensitivity, such that signatures only present in one part of the data do not contaminate results in the other parts. But signatures from related mutational processes will still be correlated both within each part and across different parts. The high degree of expected correlations from signatures known to be associated with each other (e.g., ID and SBS signatures from the same underlying mutational process, clock-like signatures SBS1 and SBS5, etc.) indicates improved signature assignment results.

4. Claim: 'resolve long-standing issues with the ambiguous 'flat' signatures'.

In the same vein this is only supported by handwaving arguments. A recent publication has attributed the subtle distinction of SBS5 and SBS40 to heterochromatin (<https://doi.org/10.1038/s41467-021-23551-9>).

In our manuscript, we provided three different analyses to support our conclusions related to the signature assignment of SBS40. First, we developed an *in silico* validation approach to assess the quality of final signature assignments and inform problems present in specific signatures (Results Section 2.4, Figure 5). Our analysis showed that the assignment of SBS40 by the PCAWG consortium in many tumor types was not consistent with data and this over-assignment problem was largely mitigated by MuSiCal (Figure 7a). Second, to provide evidence from an independent perspective, we looked at clock-like signatures SBS1 and SBS5. The expected correlation between these two signatures was clearly confounded by the SBS40 over-assignment from the PCAWG consortium (SBS40 stealing weights from SBS5), while MuSiCal-derived signature assignments led to much improved consistency with expectation (Figure 7c, d, Extended Data Figure 23). Third, in tumor types where MuSiCal believed SBS40 should be present, we found a strong correlation between SBS40 and ID5 in results obtained by both MuSiCal and PCAWG, whereas in the other tumor types, this correlation was not present in the PCAWG results (Extended Data Figure 22). This observation suggests that SBS40 and ID5 are potentially produced from the same underlying mutational process that is tumor type-specific, and that the assignment of SBS40 in the other tumor types by PCAWG is unreliable. Together, we believe our conclusions related to SBS40 were well supported.

In the TensorSignatures paper pointed out by the Reviewer, we did not find any specific discussion on SBS40 except "... two signatures with relatively uniform base substitution spectra, TS03 (unknown/quiet chromatin), and TS04 (unknown/active chromatin), which loosely correspond to SBS40 and SBS5". Apart from the visual flatness of TS03 and TS04 and their prevalence in many tumor types, we found no further analyses on their direct relation to SBS40. Also, signature discovery in the TensorSignatures paper was performed with samples from all tumor types pooled together, potentially convoluting tumor type-specific signals. Nevertheless, we agree with the Reviewer that epigenomic information including chromatin states is valuable data that has been overlooked in signature analysis, and the TensorSignatures paper is an excellent addition to the field in this regard. We hope to further develop MuSiCal in the future to incorporate epigenomics information and provide more insights into signatures with unknown etiologies.

5. Claim: ‘By reanalyzing over 2,700 cancer genomes, we provide an improved reference catalog of signatures and their assignments’.

For their catalogue to be truly an improved reference, the authors should analyse additional data sets, such as data from the Hartwig Medical Foundation, which encompasses another 4000 cancer genomes. It’s also worth noting that a recent analysis of an extra 10,000 cancer genomes from Genomics England was published (<http://dx.doi.org/10.1126/science.abl9283>). So if the aim of the manuscript were to define a better reference catalogue, the bar would certainly be higher.

We agree with the Reviewer that a comprehensive reference catalog will eventually need to be derived from an integrative analysis of data from multiple landmark studies, including the Hartwig and the Genomic England cohorts. However, we think that such an analysis is out of scope of the current manuscript. For instance, we have been trying to gain access to the Hartwig data, but our applications have been denied—the Hartwig Foundation requires an institution to sign a data access agreement which includes a fine of \$50,000 *per day* for any data breach, a term that my institution is unwilling to sign. Incorporation of additional data is certainly something that we are interested in doing in the future, but if we do it for this paper, it will make it difficult to compare the performance of our method with SigProfiler because of the underlying difference in the data used. We do believe that the MuSiCal- derived catalog presented in this manuscript (especially for ID signatures) represents an improved version compared to the COSMIC catalog, which is also largely based on PCAWG data. In keeping with

the Reviewer’s comment, we have toned down the language throughout our manuscript. Specifically, we have removed the word “reference” and focused the descriptions more on direct comparisons with the COSMIC catalog and PCAWG results. The following text was also included in the Discussion section:

The number of whole genomes of cancer and other diseases continues to grow rapidly, especially by consortium projects such as Genomics England’s 100,000 Genomes Project [56, 70] and Hartwig Medical Foundation’s metastatic tumor project [71, 72]. Applications of MuSiCal to these datasets will further refine the set of mutational signatures and facilitate comparison of signatures from different contexts, such as tumor types, metastatic status, and treatments received.

References

1. Gulhan, D. C., Lee, J. J.-K., Melloni, G. E. M., Cortes-Ciriano, I. & Park, P. J. Detecting the mutational signature of homologous recombination deficiency in clinical samples. *Nature Genetics* 51 (5), 912 (2019).
2. Degasperi, A. et al. A practical framework and online tool for mutational signature analyses show inter-tissue variation and driver dependencies. *Nat Cancer* 1 (2), 249–263 (2020).
3. Islam, S. A. et al. Uncovering novel mutational signatures by de novo extraction with SigProfilerExtractor. *Cell Genomics* 100179 (2022).

4. Pich, Oriol, et al. The evolution of hematopoietic cells under cancer therapy. *Nature communications* 12.1 (2021): 1-11.
5. Degasperi, A. et al. Substitution mutational signatures in whole-genome-sequenced cancers in the UK population. *Science* 376 (6591) (2022).
6. Kim, J. et al. Somatic ERCC2 mutations are associated with a distinct genomic signature in urothelial tumors. *Nat Genet* 48 (6), 600–606 (2016).
7. Kasar, S. et al. Whole-genome sequencing reveals activation-induced cytidine deaminase signatures during indolent chronic lymphocytic leukaemia evolution. *Nature Communications* 6, 8866 (2015).
8. Taylor-Weiner, A. et al. Scaling computational genomics to millions of individuals with GPUs. *Genome Biol* 20 (1), 228 (2019).
9. Rosales, R. A., Drummond, R. D., Valieris, R., Dias-Neto, E. & da Silva, I. T. signeR: an empirical Bayesian approach to mutational signature discovery. *Bioinformatics (Oxford, England)* 33 (1), 8–16 (2017).
10. Slawski, Martin, and Matthias Hein. Sparse recovery by thresholded non-negative least squares. *Advances in neural information processing systems* 24 (2011).
11. Liu, Mo, Yang Wu, Nanhai Jiang, Arnoud Boot, and Steven G. Rozen. mSigHdp: hierarchical Dirichlet process mixture modeling for mutational signature discovery. *bioRxiv* (2022).

Decision Letter, first revision:

Dear Peter,

Your Technical Report entitled "Accurate and sensitive mutational signature analysis with MuSiCal" has now been seen by the original 3 referees, whose comments are attached. In the light of their advice we have decided that we cannot offer to publish your manuscript in *Nature Genetics*.

While the referees continue to find your work of some interest and acknowledge the improvement in revision, they also remain unconvinced that MuSiCal offers the overall advance for mutational signature analysis that would justify publication at the journal.

In very brief: Referee #2, previously positive, remains so. Reviewer #1, who requested integration of other (e.g. clinical) data in the analysis workflow, still thinks that this is needed, and goes on to make detailed comments regarding some of the signatures found. They make the point that a lack of a solid ground truth affects interpretation of your results and assessment of the advance presented by MuSiCal; and, indeed, Referee #3 - whose major criticism was exactly this point in the initial review - still thinks is the case.

We feel that these reservations are sufficiently important as to preclude publication of this study in *Nature Genetics*.

However, I have spoken with my colleagues at other journals, and they have expressed an interest in

your manuscript.

Firstly, you may be interested in transferring a further revised manuscript to Genome Biology (<https://genomebiology.biomedcentral.com/>). I have consulted with the editors at Genome Biology and they would be delighted to consider your manuscript, provided that you are able to compare MuSiCal with other tools in a more neutral and unbiased manner. The editors also recommend that you address the performance concerns raised such as false positives and threshold for reconstruction, ideally you should also include other ground truth(s) as suggested. With your submission please attach an updated point-by-point response addressing the remaining concerns from the reviewers. The further revised manuscript would either be sent back to the original reviewers or be assessed by an Editorial Board Member.

The instructions to authors are available at <http://genomebiology.biomedcentral.com/submission-guidelines/>, and you can submit your article at <http://www.editorialmanager.com/gbio>. If you would like to discuss your manuscript with an editor or encounter any issue in submitting your manuscript, please contact the Chief Editor, Veronique van den Berghe (veronique.vandenberghen@nature.com), who will be able to ensure that the processing of your manuscript is fast-tracked.

I have also discussed your manuscript and the reviewers' comments with our colleagues at Nature Communications. They would send the appropriately revised version out for further review if you transfer the revised manuscript to Nature Communications. Should you wish to have your revised paper considered by Nature Communications, please use the link to the Springer Nature manuscript transfer service in the footnote once the revision is ready, and include a point-by-point response to the reviewers' concerns. Nature Communications would expect your revision to fully address all the technical concerns that the reviewers continue to raise.

Your handling editor at Nature Communications would be Dr Ilse Valtierra (ilseariadna.valtierragutierrez@nature.com). If there is anything you would like to discuss before transferring the paper and its reviews, please don't hesitate to contact her by e-mail.

Please note that Nature Communications is a fully open access journal. For information about article processing charges, open access funding, and advice and support from Springer Nature, please consult the Nature Communications Open Access page (www.nature.com/ncomms/open_access/index.html).

I am sorry that we cannot be more positive on this occasion but hope that you will find our referees' comments helpful when preparing your paper for submission elsewhere.

With all best wishes,

Michael Fletcher, PhD
Senior Editor, Nature Genetics

ORCID: 0000-0003-1589-7087

Reviewers' Comments:

Reviewer #1:

Remarks to the Author:

Jin et al rebuttal and revised manuscript have definitely some merits. After carefully reading the updates, supplementary data, and rebuttal letter I think the Authors did a good job of showing the statistical advantages that MuSiCal compared to SigProfiler. However, there are still some aspects previously raised by me and Reviewer #3 that I feel have not been addressed properly.

One of the key weaknesses in the first version was the lack of solid ground truth on which to test MuSiCal performances. In the rebuttal, as "ground truth" analysis, the Authors combine different cancers and tested how MuSiCal SBS3 predicts the presence of HRD. It is important to highlight that SBS3 alone was the weakest feature in the original paper from Davies et al Nat Med 2017, and therefore, it should not be used alone to estimate HRD/BRCAness. According to the seminal paper from NIK Zainal lab, the combination of SV, ID, CNV, and SNV (i.e. HRDetect) strongly predicts BRCA-deficient ovarian and breast tumors with an AUC of 0.98. Taking into account these two elements, in my opinion, the right analysis should have shown how many more "real" HRD cases we can define using MuSiCal SBS3 calls for HRDetect, compared to the HRDetect SBS3 estimates.

As suggested by Reviewer #3, there are multiple ways to test the ground truth. Authors could have checked for POLY and MSI signatures, SBS9 in post-germinal center tumors (e.g. CLL), platinum, and temozolomide in exposed samples. As also suggested in my previous comments, chemotherapy-induced mutational signatures are quite useful as ground truth, and more and more small and large data sets are published almost every month. I understand the difficulties to retrieve and reanalyze large data sets with >10.000 WGS, but chemotherapy-based performances might require a much smaller sample size. Working around these robust and well-recognized mutational signatures and established biological associations would have helped to provide the biological ground truth that this tool needs.

I thank the Authors for providing 96 profiles of PCAWG WGS in Figure R4-6. This is very useful to understand MuSiCal data and interpretation. Looking carefully at these plots, I agree that SigProfiler made several mistakes. For example, in Figure R4, I agree that SBS11 is likely, not present in these two samples in line with their clinical history. As stated by the Authors, limitations, and mistakes in SigProfiler output are expected. However, I think that the Authors should have done more effort to correct MuSiCal false positive calls. Below I listed some of the likely false positive cases where I think the lack of biological ground truth or prior knowledge affected MuSiCal's performances.

Figures R5 and R7: both are somehow problematic. The Authors basically claim that SBS31/SBS35 is correctly extracted by MuSiCal, despite these samples were not prior exposed to platinum. It is hard to believe that SP135478 has SBS31 looking at its profile. In addition to the samples' clinical history, SBS31 known transcriptional strand bias could have been used to support the Authors' claims. Without a documented exposure and strand bias, it is hard from the clinical and biological point of view to support SBS31 in some of these samples. In the rebuttal letter (but not in the manuscript) they said: "[...] we do not rule out the possibility that the small amount of SBS31 in glioblastoma represents an unknown mutational process (e.g., another drug treatment) that produces a similar spectrum to SBS31 [...]". I think this is not a good explanation. SBS31 has been validated on several papers and is one of the most established chemotherapy-related signatures. I would be ok to consider the possibility that another process can mimic SBS31 if more evidence were provided.

In Figure R7, I don't think SBS31 is there as well. Considering that the GBC PCAWG/TCGA tumors should be treatment naïve, and platinum is not commonly used in GBC, claiming SBS31 in a non-platinum exposed sample is quite confusing and might affect how the community might react to this tool and its output. Again, strand bias should help.

The correction that the Authors made for the myeloid tumors with false SBS31 signature goes in the direction I previously suggested. Prior knowledge of exposure, clinical history, sample set composition, and biology are essential components for an accurate mutational signatures analysis and for reducing false positives.

In Figure R9 the Authors claim SBS9 in MPN. Again, I am not convinced and I believe this is a false positive. SBS9 (Poly-eta) is only found in post-germinal center normal and tumor B-cell. CLL and mantle cell lymphomas (MCL) are perfect examples and Elias Campo's Lab clearly showed that the presence of SBS9 is even more accurate than IGHV status in differentiating mutated and unmutated IGHV CLL and MCL (i.e. post- and pre-germinal-center tumors, respectively). Looking at the 96 mutational profiles, it seems that SBS9 extraction was likely due to an isolated high peak in T[T>G]A. Visually and biologically I cannot see any robust evidence of SBS9 in these two samples. Also, SBS9 is virtually always present with SBS84. This again highlights the need for integrating biology and clinic in any mutational signature extraction. For example, myeloid tumors are known to have only aging signatures across multiple studies, in line with hemopoietic stem cell biology and single-cell expansion WGS data (e.g., Mitchell et al Nature 2022; Williams et al Nature 2022). The only additional mutations we can expect in this clinical and biological context are rare genetic syndrome (e.g. Fanconi anemia, MBD4 deficiency) or exposure to some mutagenic element. To me, it does not make a lot of sense to fit/extract post-germinal center SBS9 in a tumor that biologically cannot have that signature without providing a biological rationale/explanation.

Figure R10. Again, I think here the Authors are overfitting a signature that cannot be present in the bladder. Furthermore, looking at the profiles, it is hard to support SBS84... SBS84 has been shown to be caused by AID in normal and tumor B-cells by dozens of papers. Its activity is usually clustered and not genome-wide. This is also supported by MuSiCal de novo extraction on lymphomas in Supplementary Fig. 6, where SBS84 is not extracted as expected by genome-wide analysis. It would be very problematic to show SBS84/AID genome-wide activity in a solid tumor that biologically cannot express AID.

Looking at the data and what is included in the rebuttal letter I am not really sure that the Authors can say: "[...] Third, we provided additional examples to show that the final signature assignments from MuSiCal were more biologically meaningful compared to those from the PCAWG consortium [...]". The main weakness that persists in this study is the lack of robust biological correlatives and ground truth elements that validate some of the claims.

"[...] once the de novo extraction does not produce the correct signatures, it is difficult to recover in subsequent steps, no matter how much filtering and optimization are carried out [...]". I think that this statement is quite against most of my previous suggestions. I do believe that there is a large margin for improvements in the post-extraction phase. Several labs currently use % of the contribution, the confidence of interval (CI) from fitting methods, samples' clinical history, presence of distinct drivers, integration with other genomic features, clonal/subclonal deconvolution, strand bias, and clustered/unclustered approaches. For example, SBS7 is often called by SigProfiler in non-UV

light-exposed cancer with high APOBEC SBS2. However, the contribution is usually low, CIs are usually spanning zero, and SBS7 always disappears after a proper fitting. This is a clear false positive that is easily corrected in the post-extraction phase.

"[...]In fact, we recommended in our tutorial that the signature catalog should be restricted to tumor type-specific signatures during the matching and refitting steps (<https://github.com/parklab/MuSiCal/tree/main/examples>). However, such prior knowledge needs to be established in the first place through an unbiased approach [...]" . If the scope of this paper is to create a more accurate and reliable catalog of signatures per cancer type, then a larger number of samples should be included for each tumor type (as suggested by Reviewer #3). Some of the tumor types in PCAWG are not well represented, and therefore, to create a robust catalog for each tumor type more samples should be added. I also explained above that some of the assignments are likely incorrect, and this could affect the creation definitive catalog per cancer type. Because we have now published mutational signatures studies with >10.000 WGS, the competition is hard. Maybe it would be more useful for the community to have a tool with prior knowledge and weight, including the knowledge derived from these recent large seminal papers.

Overall, I do believe that MuSiCal has improvements in its multi-step process compared to SigProfiler, but it is hard to quantify them without a proper and robust "ground truth" analysis. While the math and stats are superior in the simulation setting, the biology and clinical history must be taken into account, otherwise, the impact of this tool might be limited in a field where the bar is very high.

Reviewer #2:

Remarks to the Author:

I would like to express our appreciation for the extra steps and changes made to the manuscript, especially regarding the benchmarking routine. The inclusion of a better metric (PRC and auPRC) and more extensive simulations, including noise and a wider range of tools to compare to, has significantly strengthened the paper and claims about superiority of the method. I commend your thoroughness and attention to detail in this regard.

I am pleased to see that all minor and most of the major points have been addressed in the manuscript.

However, I do have some additional comments and questions following up on some of the major points I raised previously. I have outlined these comments and questions below (the numbers correspond to those from my initial review) and I look forward to your response.

Major Point 5

I appreciate the efforts you have made in conducting extra noise simulations in your study. It is important to assess the robustness of your results to noise, and I commend your attention to this issue. Two additional points:

Random noise: While I appreciate that the effect of random noise on the performance on MuSiCal appears to be limited, the introduced noise levels seem to be on the low side. At what noise level does the accuracy / precision-recall substantially fall off? Surely there must be a noise level at which MuSiCal is not able to faithfully reconstruct the underlying signatures anymore. Is there any estimate of the amount of noise that is present in real data and how it would relate to the simulated noise levels?

Spike-in noise: Furthermore, I noticed that the selection of SBS48 signatures in the simulations could

potentially be perceived as cherry-picking, although I understand the reasoning behind it. It appears that this selection has a stronger effect than random noise. To provide additional context for the analysis, I would recommend using a few spiked-in signatures, such as a flat one and one with a unique profile. Additionally, it would have been helpful to include multiple levels of such spike-in noise rather than just 5% in the simulations, similar to the case of random noise. Same as before, I would like to see at which noise level MuSiCal is not able to faithfully reconstruct the underlying signatures.

Major Point 6

Up to the authors: I think it's worth mentioning that the SD is lower for the MuSiCal results especially since the SD is shown in Fig 2c. This might not be obvious to all readers.

Major Point 8

Up to the authors: I think Fig R14 could be included in the main manuscript as an Extended Figure as other readers might also wonder about the effect of the mvNMF on the reconstruction error. Could be included with a single sentence like "While mvNMF increases the uniqueness of solutions, it did not significantly reduce the reconstruction error (Ext Data Fig XX)."

Major Point 10

Why have the MuSiCal and SigProfileExtractor de novo signatures changed from the first submission? I thought they are still chosen best on the ideal F1 value?

Major Point 11

I appreciate the extra analysis performed by the authors and acknowledge that MuSiCal still outperforms SigProfileExtractor even when forced to the same number of signatures. However, it also becomes clear that the selection of the right number of signatures has a larger effect on the accuracy of the results ($0.893 \rightarrow 0.914 = 0.024$) than the added benefit of using minimum volume NMF and the improved COSMIC matching/refitting routines which increase accuracy from ($0.914 \rightarrow 0.929 = 0.015$), at least in this example. I understand that choosing the right number of signatures is in itself a difficult task and one at which MuSiCal shines. However, I think it would be good to explicitly state the strength of this effect in the main text as otherwise the reader doesn't know which aspect of MuSiCal contributes how much to the improved accuracy.

On Figure R17

There is a very clear increase in reconstruction error for $r=10, 12, 14$ that really stumped me. Reconstruction error for these values of r is higher than for all other r 's (excluding $r=1$), but this does not seem to affect the "neighboring" odd values 11, 13 and 15 to the same degree. This very much makes me suspicious about potential numerical issues or a bug in the code. Do the authors have an explanation for this effect? I understand that this behavior does not affect the points raised in Major Point 11 but I do believe that this behavior might be reminiscent of a larger error in the MuSiCal code.

Major Point 14

I acknowledge the extra work performed by the authors and found the new Ext Data Fig 24 quite informative. Additionally, I think the runtime should be stated in the main text or at least in the Online Methods. Ideally, the authors could include a small section termed "Computational cost of MuSiCal" under section 4.3 as this is where readers are likely going to search for this information.

For Ext Data Fig 24a: Definitely show the y-axis in hours and maybe show the y-axis as a log-scale.

Major Point 17

Maybe I misunderstand the analysis done here but both simulations (PCAWG and MuSiCal) are

compared to W_{data} and H_{data} gathered from mvNMF, right?. But isn't W_{data}/H_{data} created with mvNMF and therefore directly linked to the simulated MuSiCal data? So MuSiCal uses W_{data}/H_{data} to match and refit W_s/H_s and in turn uses these matrices to simulate W_{simul}/H_{simul} . The difference (simul - data) therefore points towards errors introduced in the matching and refitting stages. On the other hand, PCAWG used a different set of H_{data}/W_{data} that is not available to the authors from which they assigned their W_s/H_s matrices. So for the PCAWG assignments, the difference (simul - data) is a sum of the errors introduced in the PCAWG matching and refitting stages as well as the difference in $data_{mvNMF}$ and $data_{PCAWG}$!

Therefore the fact that the simulated MuSiCal data outperforms the simulated PCAWG data is very much expected from the way this comparison is set up. A more fair comparison would be to use the original W_{data}/H_{data} matrices from the PCAWG analysis (though they are likely not publicly available).

If my assumptions are correct, then this analysis just produces what you put in and is therefore not very informative.

Major Point 21

After rereading the relevant sections I do believe that the MMRD/MMRP/POLE split is not sufficiently described in the main text given its strong effect on the discovery of new SBS signatures. Currently section 2.5 states: "In automatic cohort stratification, heterogeneous datasets are stratified into subsets". While this is true for individual stratification in Kidney.RCC, Liver.HCC, and Skin.Melanoma it is incorrect and feels disingenuous for the MMRD/MMRP/POLE stratification which in section 4.5.1 is very clearly described as a manual process.

Since this manual split has a strong effect on the outcome and is not performed using the MuSiCal preprocessing module it needs to be included in section 2.5. Something along the lines of "After an initial manual separation of MMRD / MMRP samples, tumor type-specific de novo signature discovery is performed ...".

Additionally, I would mention this again when the results are presented: "... in MMRD tumors and therefore benefitting from our manual stratification process, likely representing nonlinear interactions between DNA damage ..."

—

Small side point: Fig 6b is missing a "*" next to the black bar besides SBS95-100 (like it is present in 6c for ID19-26)

Major Point 22

I thank the authors for the extensive extra analysis in the case of the newly discovered ID11b signature.

Up to the authors: Include half a sentence before the description of the ID11b smoking correlations: "Unlike the original ID11 signature, ID11b correlates with both ..."

Reviewer #3:

Remarks to the Author:

In the revised manuscript the authors present additional bioinformatics benchmarking of their algorithm MuSiCal for mutational signature extraction. In brief, the algorithm uses a regularised form of NMF termed minimal volume NMF based on trinucleotide representation of single base substitutions or alternatively indels. The additional analyses entail further simulations of base substitution spectra with added Gaussian noise, a spike-in of a confounding signature as well as a comparison to a broader

series of tools. These analyses reveal a consistent improvement of MuSiCal over 4 other tools which are used widely in the field.

The implemented changes addressed my first two major concerns (performance of MuSiCal in relation to other tools). Based on the available data it can be concluded that MuSiCal outperforms other algorithms run out of the box and on simulated data. Yet I do maintain that such performance gains need to be taken with a pinch of salt because (i) the data is simulated according to the author's modelling assumptions and also because (ii) there is often an unintended bias because as an author one knows one's algorithm's ideal tuning parameters, whereas other software is much harder to run. Given that all methods are essentially working on the same NMF core algorithm I'm not convinced that the same performance cannot be achieved by other tools.

I have tested the available code from github though and can assert that one can run the provided examples and also my own analyses. Yet it is worth mentioning that the run time easily exceeds 48h.

Unfortunately, the authors chose to mostly dismiss my other three comments. These were related to the extent to which MuSiCal gives insights into signatures with unknown etiologies, resolves issues with 'flat' signatures and whether the authors provide an improved reference catalogue.

My overall verdict is thus that MuSiCal may provide an improvement over other widely-used tools for calculating mutational signatures based on 96-class single base substitutions and when evaluated on simulated data. This could be a useful, albeit conceptually incremental advance. Yet in the absence of proper experimental validation data, which link the data to true mutational processes, it remains questionable whether the reported differences truly provide new insights or whether they just confirm subjective assumptions about how mutational signatures should look like.

So it remains questionable whether there's an actual improvement and the methodological advance appears incremental. MuSiCal doesn't tackle any fundamental questions in the field.

Decision Letter, Appeal – first revision:

7th Jul 2023

Dear Peter,

Thank you for your message of 7th Jul 2023, asking us to reconsider our decision on your manuscript "Accurate and sensitive mutational signature analysis with MuSiCal".

I have now discussed the points of your letter with my colleagues, and we think that you have some valid points. We therefore invite you to submit your revised manuscript for further peer review.

When preparing a revision, please ensure that it fully complies with our editorial requirements for format and style; details can be found in the Guide to Authors on our website (<http://www.nature.com/ng/>).

Please be sure that your manuscript is accompanied by a separate letter detailing the changes you have made and your response to the points raised. At this stage we will need you to upload:

- 1) a copy of the manuscript in MS Word .docx format.
- 2) The Editorial Policy Checklist:
<https://www.nature.com/documents/nr-editorial-policy-checklist.pdf>
- 3) The Reporting Summary:
<https://www.nature.com/documents/nr-reporting-summary.pdf>
(Here you can read about the role of the Reporting Summary in reproducible science:
<https://www.nature.com/news/announcement-towards-greater-reproducibility-for-life-sciences-research-in-nature-1.22062>)

Please use the link below to be taken directly to the site and view and revise your manuscript:

[redacted]

With kind wishes,

Michael Fletcher, PhD
Senior Editor, Nature Genetics

ORCID: 0000-0003-1589-7087

Author Rebuttal, first revision:

Detailed responses to reviewers

Reviewer #1:

Remarks to the Author:

Jin et al rebuttal and revised manuscript have definitely some merits. After carefully reading the updates, supplementary data, and rebuttal letter I think the Authors did a good job of showing the statistical advantages that MuSiCal compared to SigProfiler. However, there are still some aspects previously raised by me and Reviewer #3 that I feel have not been addressed properly.

In this revision, we have added two additional ground truth analyses based on platinum-associated SBS31/35 (**Figure R20**) and HRD-associated SBS3 (included in the first revision but presented in a more straightforward manner in this revision; **Figure R19**). MuSiCal outperformed existing methods in both analyses, producing signature assignments that were more consistent with biological ground truth.

In response to Reviewer's concerns on signature assignments of specific signatures and samples, we have found additional orthogonal evidence that MuSiCal's results suggest new biology instead of being errors. Specifically, we found that a signature (SBS137) highly similar to the platinum-associated SBS31 was discovered in Genomes England (GEL) CNS tumors by a separate study (Degasperi et al. Science 2022) (see response to Comment 4 and **Figure R21**). Furthermore, a subset of myeloid tumors

in the same study clearly showed the presence of SBS9, previously believed to be specific to post-germinal center lymphoid tumors (see response to Comment 6 and **Figure R22**). We do not fundamentally disagree with the Reviewer that biological knowledge should have a role in interpretation of mutational signatures, but for both of the cases in which the Reviewer believed MuSiCal was incorrect, we believe that the prior expectation is likely to be incorrect. It is of course possible that the GEL paper results are also wrong, but we think the actual spectra of their samples are even more convincing than the spectra of our samples.

1. One of the key weaknesses in the first version was the lack of solid ground truth on which to test MuSiCal performances. In the rebuttal, as "ground truth" analysis, the Authors combine different cancers and tested how MuSiCal SBS3 predicts the presence of HRD. It is important to highlight that SBS3 alone was the weakest feature in the original paper from Davies et al Nat Med 2017, and therefore, it should not be used alone to estimate HRD/BRCAness. According to the seminal paper from Nik-Zainal lab, the combination of SV, ID, CNV, and SNV (i.e. HRDetect) strongly predicts BRCA-deficient ovarian and breast tumors with an AUC of 0.98. Taking into account these two elements, in my opinion, the right analysis should have shown how many more "real" HRD cases we can define using MuSiCal SBS3 calls for HRDetect, compared to the HRDetect SBS3 estimates.

The Reviewer is absolutely right in that a combination of multiple features should be used for predicting homologous recombination deficiency (HRD) or other clinically actionable mutational processes. As we have elaborated in the response to Reviewer's Comment 1 from the first revision, the aim of our manuscript is not to integrate multiple features for phenotype prediction, but to solve an upstream problem that enables the subsequent integration analysis. The Reviewer's comment above concerns part of this upstream problem that involves assigning the right mutational signatures to the right samples. To test MuSiCal's performance on this task using HRD and SBS3, we believe the right analysis would be to show improved consistency between MuSiCal-derived SBS3 assignments and ground truth HRD status. In Figure R2 from the first revision, we showed exactly that, i.e., using MuSiCal-derived SBS3

assignment alone, higher sensitivity was achieved at the same specificity for predicting the ground truth HRD labels, compared to using SBS3 assignments from other methods. Of note, we considered HRDetect classifications as ground truth because HRDetect has demonstrated high accuracy for HRD prediction (as Reviewer mentioned) and BRCA deficiency itself misses a significant fraction of real HRD samples, e.g., those with RAD51C or PALB2 losses and other unknown underlying causes (Nguyen et al. "Pan-cancer landscape of homologous recombination deficiency" Nature Communications 2020 and the HRDetect paper, Davies et al. Nature Medicine 2017).

To further demonstrate MuSiCal's improved performance, below we present the same results in a different and perhaps more straightforward manner. In **Figure R19**, we took SBS3 assignments from different methods and binarized them into SBS3 positive (nonzero) vs. negative (zero) labels. We then compared these SBS3 labels to the ground truth HRD labels. The difference here is that in Figure R2, we treated the SBS3 exposures as a continuous variable and compared to the ground truth HRD

labels directly. Using binarized SBS3 labels could be more intuitive, as we expect a perfect algorithm to assign SBS3 to all HRD samples, but not to any HRP samples. To quantify the performance of different methods, we used two popular measures for the accuracy of a binary classification problem – balanced accuracy (BAcc), defined as the arithmetic mean of sensitivity and specificity, and F1 score, defined as the harmonic mean of precision and recall. MuSiCal achieved the best performance among all included methods with both measures (**Figure R19**). Specifically, MuSiCal achieved BAcc of 0.92 (sensitivity = 0.96, specificity = 0.89) and F1 score of 0.80 (precision = 0.68, recall = 0.96). By comparison, SigProfilerExtractor and NNLS performed worst overall, having lower sensitivity/recall (0.94 and 0.92, respectively), specificity (0.80 and 0.72, respectively), and precision (0.54 and 0.46, respectively), and thus worse BAcc (0.87 and 0.82, respectively) and F1 score (0.69 and 0.61, respectively); signature.tools.lib achieved slightly higher sensitivity/recall (0.99), but much lower specificity (<0.57) and precision (<0.37), and thus worse BAcc (<0.78) and F1 score (<0.53). In conclusion, MuSiCal produced SBS3 assignments more consistent with the ground truth HRD status, outperforming other methods.

Finally, we agree with the Reviewer that a single example based on HRD and SBS3 may not be sufficient for testing MuSiCal's performance with solid ground truths. We therefore added another example based on platinum treatment and SBS31/35. See response to Reviewer's next comment below.

Figure R19. MuSiCal-derived SBS3 assignment is more consistent with ground truth HRD status. We compared SBS3 assignments from different methods to the ground truth HRD status in PCAWG breast, ovary, pancreas, and prostate tumors. The SBS3 assignments were binarized into SBS3 positive (nonzero) vs. negative (zero) labels before comparing to HRD labels. The HRDetect final classification from [1] (Degasperi et al. Nature Cancer 2020) was considered as ground truth. For SigProfilerExtractor, the final PCAWG signature assignment results from [2] (PCAWG signature paper, Alexandrov et al. Nature 2020) were used. For signature.tools.lib, two versions of the results were used, one from [1] (Degasperi et al. Nature Cancer 2020), and the other from [3] (GEL signature paper, Degasperi et al. Science 2022). For NNLS, the spectrum of each sample was decomposed into the entire set of COSMIC signatures using NNLS. We also included results from [4] (Hartwig metastatic vs. primary paper, Martinez-Jimenez et al. Nature 2023) to be consistent with Figure R20. **(a)** Sensitivity vs. 1 - specificity. Dashed lines indicate iso- BACC (balanced accuracy) lines. **(b)** Precision vs. recall. Dashed curves indicate iso-F1 curves.

2. As suggested by Reviewer #3, there are multiple ways to test the ground truth. Authors could have checked for POLY and MSI signatures, SBS9 in post-germinal center tumors (e.g. CLL), platinum, and temozolomide in exposed

samples. As also suggested in my previous comments, chemotherapy-induced mutational signatures are quite useful as ground truth, and more and more small and large data sets are published almost every month. I understand the difficulties to retrieve and reanalyze large data sets with >10,000 WGS, but chemotherapy-based performances might require a much smaller sample size. Working around these robust and well-recognized mutational signatures and established biological associations would have helped to provide the biological ground truth that this tool needs.

Following the Reviewer's suggestion, we have added a chemotherapy-based example to further demonstrate the improved performance of MuSiCal (**Figure R20**). We obtained the mutation count matrices and clinical annotations of Hartwig samples from [4] (Hartwig metastatic vs. primary paper, Martinez-Jimenez et al. Nature 2023) published in May this year. We focused on ovarian tumors and platinum treatment, as we could only find accurate before-biopsy treatment information in the ovarian cancer cohort from [4]. There were $n = 132$ ovarian tumors from Hartwig, among which $n = 111$ had platinum treatment before biopsy, and $n = 21$ had not. Of note, the following treatments were considered to be platinum: Cisplatin, Oxaliplatin, and Carboplatin. To make the dataset more balanced, we further included $n = 109$ ovarian tumors from PCAWG, which were all assumed to be treatment naïve. In total, we had 111 and 130 samples with and without platinum treatment, respectively. Since platinum treatment is associated with SBS31 and SBS35, we compared SBS31/35 assignments from different methods to the ground truth platinum treatment status. Similar to Figure R19, we binarized the SBS31/35 assignments into SBS31/35 positive (SBS31 nonzero or SBS35 nonzero) vs. negative (zero for both SBS31 and SBS35) before the comparison. MuSiCal achieved the best performance in both BAcc and F1 score among all included methods (**Figure R20**). Specifically, MuSiCal achieved BAcc of 0.85 (sensitivity = 0.87, specificity = 0.82) and F1 score of 0.84 (precision = 0.81, recall = 0.87). By comparison, signature.tools.lib performed worst overall, having lower sensitivity/recall (0.60), specificity (0.38), and precision (0.46), and thus worse BAcc (0.49) and F1 score (0.52); SigProfilerExtractor had much lower sensitivity/recall (0.48) albeit having higher specificity (0.97) and precision (0.93), resulting in worse BAcc (0.72) and F1 score (0.63); NNLS achieved slightly higher sensitivity/recall (0.92), but much lower specificity (0.55) and precision (0.64), and thus worse BAcc (0.74) and F1 score (0.75); Signature assignments in the Hartwig paper itself [4] had really low specificity (0.015). In conclusion, MuSiCal produced SBS31/35 assignments more consistent with the ground truth platinum treatment status, outperforming other methods. In particular, MuSiCal strikes the balance between sensitivity and specificity (or precision and recall), and was the only method that achieved > 0.8 in all measures of sensitivity/recall, specificity and precision.

Taken together, we believe that the two examples in **Figure R19** (based on HRD and SBS3) and **Figure R20** (based on platinum treatment and SBS31/35) strongly suggest that MuSiCal produces signature assignments that are more consistent with ground truth information when compared to other existing methods. As to other tests the Reviewer suggested, POLE and MSI samples were usually identified beforehand for a separate signature analysis (as in our manuscript and other papers as well), resulting in a perfect performance in terms of the binary classification problem as in Figures R19 and R20. Such an analysis is thus not so informative. For SBS9, we are seeing orthogonal evidence that it might not be specific to post-germinal center lymphoid tumors (see response to Reviewer's Comment 6 below), and thus not sure if it is a solid ground truth for testing the performance.

Figure R20. MuSiCal-derived SBS31/35 assignment is more consistent with ground truth platinum treatment status. We compared SBS31/35 assignments from different methods to the ground truth platinum treatment status in ovary tumors combined from Hartwig and PCAWG. $n = 111$ samples from Hartwig with before-biopsy treatments of Cisplatin, Oxaliplatin, or Carboplatin were considered platinum positive. $n = 130$ samples were considered platinum negative, which included $n = 21$ from Hartwig without before-biopsy treatments of Cisplatin, Oxaliplatin, or Carboplatin, and $n = 109$ from PCAWG (all assumed to be treatment naive). The SBS31/35 assignments were binarized into SBS31/35 positive (nonzero) vs. negative (zero) labels before comparing to platinum labels. MuSiCal and SigProfilerExtractor results were obtained by running the respective tool on this dataset. For signature.tools.lib, the signature assignment results from [3] (GEL signature paper, Degasperi et al. Science 2022) were used. For NNLS, the spectrum of each sample was decomposed into the entire set of COSMIC signatures using NNLS. We also included results from the Hartwig paper itself [4] (Hartwig metastatic vs. primary paper, Martinez-Jimenez et al. Nature 2023) to be complete. **(a)** Sensitivity vs. 1 - specificity. Dashed lines indicate iso-BAcc (balanced accuracy)

lines. **(b)** Precision vs. recall. Dashed curves indicate iso-F1 curves.

3. I thank the Authors for providing 96 profiles of PCAWG WGS in Figure R4-6. This is very useful to understand MuSiCal data and interpretation. Looking carefully at these plots, I agree that SigProfiler made several mistakes. For example, in Figure R4, I agree that SBS11 is likely, not present in these two samples in line with their clinical history. As stated by the Authors, limitations, and mistakes in SigProfiler output are expected. However, I think that the Authors should have done more effort to correct MuSiCal false positive calls. Below I listed some of the likely false positive cases where I think the lack of biological ground truth or prior knowledge affected MuSiCal's performances.

outputs are prevailing, yet SigProfilerExtractor is still the most popular tool in the field. This is exactly why we have developed MuSiCal. With the ground truth tests provided in Figure R19 and R20, we have shown that MuSiCal produces results that are more consistent with ground truth information, outperforming SigProfilerExtractor as well as other existing methods. However, no algorithm is perfect – although MuSiCal already represents a significant improvement over existing methods, it still did not achieve perfect sensitivity and specificity simultaneously in Figure R19 and R20, i.e., false positives and false negatives are still possible. We agree that for a few signatures where solid ground truth has been established, we could have incorporated those prior knowledge to refine MuSiCal's signature assignments. But for the majority of signatures with no known etiology, let alone a biological ground truth, such refinements are impossible. Our goal in this manuscript is not to make the results of a few signatures perfectly aligned with prior knowledge in a small subset of samples, but to provide an unbiased method that could produce improved results overall. In our opinion, such an unbiased approach is critical for establishing the biological ground truth for signatures with unknown etiologies in the first place.

We would also like to kindly bring up another possibility when algorithm and prior ground truth do not align – it may suggest new biology or revision of prior knowledge, rather than a faulty algorithm. In the response to Reviewer's comments below, we provide orthogonal evidence that at least two examples brought up by the Reviewer likely fall into this scenario (see response to Reviewer's Comment 4 and 6).

4. Figures R5 and R7: both are somehow problematic. The Authors basically claim that SBS31/SBS35 is correctly extracted by MuSiCal, despite these samples were not prior exposed to platinum. It is hard to believe that SP135478 has SBS31 looking at its profile. In addition to the samples' clinical history, SBS31 known transcriptional strand bias could have been used to support the Authors' claims. Without a documented exposure and strand bias, it is hard from the clinical and biological point of view to support SBS31 in some of these samples. In the rebuttal letter (but not in the manuscript) they said: "[...] we do not rule out the possibility that the small amount of SBS31 in glioblastoma represents an unknown mutational process (e.g., another drug treatment) that produces a similar spectrum to SBS31 [...]". I think this is not a good explanation. SBS31 has been validated on several papers and is one of the most established chemotherapy-related signatures. I would be ok to consider the possibility that another process can mimic SBS31 if more evidence were provided.

In Figure R7, I don't think SBS31 is there as well. Considering that the GBM PCAWG/TCGA tumors should be treatment naïve, and platinum is not commonly used in GBM, claiming SBS31 in a non-platinum exposed sample is quite confusing and might affect how the community might react to this tool and its output. Again, strand bias should help.

We agree with the Reviewer that SBS31 is a well-established and experimentally-validated signature associated with platinum treatment. However, it does not exclude the possibility that another unknown mutational process could produce a signature similar to SBS31, resulting in SBS31 being assigned to some samples without platinum treatment. This potential redundancy of mutational signatures is a well-known phenomenon as exemplified in the seminal paper [5] (Kucab et al. Cell 2019) from the Nik-Zainal

lab, where signatures of various environmental agents were directly obtained from exposed cell cultures. In [5], many different agents produced similar signatures. Although some similarities were expected due to related agents, surprises were also observed. For example, signatures of tobacco smoking related agents DBP and DBPDE were highly similar to the aristolochic acid signature.

In the first revision, we provided visual evidence of MuSiCal's SBS31 assignment in a small number of samples that lack clinical evidence of platinum treatment, and pointed out that an unknown mutational process with a similar signature might underlie this result. To obtain additional evidence, we looked at mutational signatures identified in the much larger Genomics England (GEL) cohort from [3] (Degasperi et al. Science 2022) and found SBS137, which is highly similar to SBS31 (cosine similarity 0.83, **Figure R21**). In fact, among all COSMIC signatures, SBS31 has the highest cosine similarity to SBS137.

Interestingly, SBS137 was identified in CNS tumors from GEL, in line with MuSiCal's SBS31 assignment in GBM. SBS137 is also visually evident in some samples from Figure R5c, e.g., in SP135478 mentioned by the Reviewer. Again, we are aware that MuSiCal's results can still have errors despite being a significant improvement over existing methods (Figure R19, R20). But in this case, we believe there is at least some merit in MuSiCal's SBS31 assignment in these seemingly unreasonable samples.

In our opinion, MuSiCal's results themselves are not problematic, as they are data-driven (Figure R5 and R7) and well-validated (Figure R20). The problem here lies in the inconsistency between data-driven results and prior biological knowledge, which could instead point to new biology – in this case, an unknown mutational process with a signature similar to SBS31. Validating and investigating this signature is definitely future work and out of scope of the current manuscript. We believe that this example also demonstrates the value of retaining unbiased data-driven results as in our manuscript, rather than incorporating prior knowledge that could evolve over time.

Figure R21. Comparison of SBS31 from COSMIC and SBS137 from [3] (GEL signature paper, Degasper et al. Science 2022). SBS137 was identified in CNS tumors from GEL.

Figure R21. Comparison of SBS31 from COSMIC and SBS137 from [3] (GEL signature paper, Degasper et al. Science 2022). SBS137 was identified in CNS tumors from GEL.

5. The correction that the Authors made for the myeloid tumors with false SBS31 signature goes in the direction I previously suggested. Prior knowledge of exposure, clinical history, sample set composition, and biology are essential components for an accurate mutational signatures analysis and for reducing false positives.

The correction we made for myeloid tumors was entirely based on outlier removal. Specifically, one myeloid tumor sample dominated by SBS31 (potentially therapy-related AML) was removed when analyzing other myeloid tumor samples to prevent bleeding in signature assignment. We did not rely on prior biological knowledge or clinical data to refine the results. As we have elaborated in the response to Reviewer's Comment 3, 4, and 6 (see below), we believe that unbiased data-driven approaches – as almost all popular signature analysis tools are (e.g., SigProfilerExtractor, signature.tools.lib, SignatureAnalyzer, etc.) – is essential for mutational signature analysis.

6. In Figure R9 the Authors claim SBS9 in MPN. Again, I am not convinced and I believe this is a false positive. SBS9 (Poly-eta) is only found in post-germinal center normal and tumor B-cell. CLL and mantle cell lymphomas (MCL) are perfect examples and Elias Campo's Lab clearly showed that the presence of SBS9 is even more accurate than IGHV status in differentiating mutated and unmutated IGHV CLL and MCL (i.e. post- and pre-germinal-center tumors, respectively). Looking at the 96 mutational profiles, it seems that SBS9 extraction was likely due to an isolated high peak in T[T>G]A. Visually and biologically I cannot see any robust evidence of SBS9 in these two samples. Also, SBS9 is virtually always present with SBS84. This again highlights the need for integrating biology and clinic in any mutational signature extraction. For example, myeloid tumors are known to have only aging signatures across multiple studies, in line with hemopoietic stem cell biology and single-cell expansion WGS data (e.g., Mitchell et al Nature 2022; Williams et al Nature 2022). The only additional mutations we can expect in this clinical and biological context are rare genetic syndrome (e.g. Fanconi anemia, MBD4 deficiency) or exposure to some mutagenic element. To me, it does not make a lot of sense to fit/extract post-germinal center SBS9 in a tumor that biologically cannot have that signature without providing a biological rationale/explanation.

To obtain additional evidence of SBS9 in myeloid tumors, we again looked at Genomics England (GEL) data from [3] (Degasperi et al. Science 2022) (**Figure R22**). Among the 91 myeloid tumors included in GEL, 29 received nonzero SBS9 exposures from [3], while the other 62 were SBS9-negative. In **Figure R22c**, we plotted the mean spectrum of the 29 SBS9-positive myeloid tumors. Visually, this mean spectrum clearly contained a component of SBS9. Quantitatively, this mean spectrum had a cosine similarity of 0.72 to COSMIC SBS9 and a normalized NNLS coefficient of 0.21 for SBS9 (i.e., about 21% of the mutations in these samples were contributed by SBS9). These observations suggest that SBS9 is likely to be present in myeloid tumors, consistent with MuSiCal's SBS9 assignment in myeloid tumors. As we have pointed out in the first revision, it is likely that SBS9-associated activity of polymerase eta has roles beyond somatic hypermutation in lymphoid cells. Of note, these observations are not necessarily contradictory to the previous studies mentioned by the Reviewer. SBS9 could be a perfect marker of IGHV status in lymphoid tumors, while still be present in some myeloid tumors due to a related but distinct mutational process. Mitchell et al Nature 2022 studied healthy hematopoiesis and thus may not be representative for blood tumors. Williams et al Nature 2022 studied a small cohort of 12 patients with myeloproliferative neoplasms and thus may not cover a full spectrum of myeloid tumors.

In our opinion, similar to SBS31 in the response to Reviewer's Comment 4 above, SBS9 could potentially be another example of a signature representing multiple (possibly related) mutational processes. It is also another example that demonstrates the value of unbiased data-driven approaches for signature analysis. Although prior biological knowledge could help refine signature assignments in certain scenarios where the prior is known and solid, the inconsistency between data-driven results and prior knowledge could point to broader new biology.

Figure R22. Evidence of SBS9 in GEL myeloid tumors. (a) SBS9 from COSMIC. **(b)** SBS9 from [3] (GEL signature paper, Degasperi et al. Science 2022). **(c)** Mean spectrum of GEL myeloid tumors with nonzero SBS9 exposures from [3]. Raw sample spectra and signature assignment results were directly taken from [3]. Out of the 91 myeloid tumors, 29 had nonzero exposures of SBS9. Detailed per-sample tumor type information was not provided in [3], except that tumors falling into the myeloid category included AML_ACUTE_MYELOID_LEUKAEMIA, CHRONIC_MYELOID_LEUKAEMIA, MULTIPLE_MYELOMA, MYELOYDYSPLASTIC_SYNDROME_HIGH_RISK. **(d)** Mean spectrum of GEL myeloid tumors with zero SBS9 exposures from [3].

7. Figure R10. Again, I think here the Authors are overfitting a signature that cannot be present in the bladder. Furthermore, looking at the profiles, it is hard to support SBS84... SBS84 has been shown to be caused by AID in normal and tumor B-cells by dozens of papers. Its activity is usually clustered and not genome-wide. This is also supported by MuSiCal de novo extraction on lymphomas in Supplementary Fig. 6, where SBS84 is not extracted as expected by genome-wide analysis. It would be very problematic to show SBS84/AID genome-wide activity in a solid tumor that biologically cannot express AID.

This comment falls along the same vein as Reviewer's Comments 3-6. We again emphasize that we

aimed at an unbiased data-driven approach, where any inconsistency between the results and prior biological knowledge could represent an error on the one hand – which does not undermine MuSiCal’s improvement over existing methods (Figure R19 and R20) – or suggest new biology on the other hand (Figure R21 and R22). For SBS84, the GEL signature paper [3] (Degasperi et al. Science 2022) also

assigned it to solid tumors such as colorectal cancer, which could potentially be orthogonal evidence that warrants future studies to look for a similar signature in solid tumors.

8. Looking at the data and what is included in the rebuttal letter I am not really sure that the Authors can say: “[...] Third, we provided additional examples to show that the final signature assignments from MuSiCal were more biologically meaningful compared to those from the PCAWG consortium [...]”. The main weakness that persists in this study is the lack of robust biological correlatives and ground truth elements that validate some of the claims.

In the original manuscript and the first revision, we have conducted comprehensive simulation studies to demonstrate the improved performance of MuSiCal. We have also included several examples (listed after the sentence referenced by the Reviewer) where orthogonal data supported our newly discovered signatures or revised signature assignments. In this revision, we further provided two ground truth analyses based on HRD-associated SBS3 (included in the first revision but presented in a more straightforward manner in this revision; see Figure R19) and platinum-associated SBS31/35 (Figure R20). MuSiCal outperformed existing methods in both analyses. In addition, we provided orthogonal evidence suggesting new biology in two cases where MuSiCal’s data-driven results seemingly contradict prior biological knowledge (see response to Reviewer’s Comment 4 and 6). We believe that these additional results have strengthened our study in terms of the ground truth elements.

9. “[...] once the *de novo* extraction does not produce the correct signatures, it is difficult to recover in subsequent steps, no matter how much filtering and optimization are carried out [...]”. I think that this statement is quite against most of my previous suggestions. I do believe that there is a large margin for improvements in the post-extraction phase. Several labs currently use % of the contribution, the confidence of interval (CI) from fitting methods, samples’ clinical history, presence of distinct drivers, integration with other genomic features, clonal/subclonal deconvolution, strand bias, and clustered/unclustered approaches. For example, SBS7 is often called by SigProfiler in non-UV light-exposed cancer with high APOBEC SBS2. However, the contribution is usually low, CIs are usually spanning zero, and SBS7 always disappears after a proper fitting. This is a clear false positive that is easily corrected in the post-extraction phase.

In our manuscript, we showed exactly an example of the confounding effect between SBS2 and SBS7a during *de novo* signature discovery (**Extended Data Figure 3d**, copied below). The overlapping features between these two signatures often resulted in artifactual distortions in NMF solutions, which were mitigated by mvNMF. In fact, this confounding interaction rendered SBS7a one of the top

signatures that NMF specifically had problem with, despite SBS7a itself being a sparse signature, while mvNMF performed significantly better (see Figure 2c). We agree with the reviewer that these confounding effects could potentially be partially rescued by downstream analysis, and we apologize that we might have sounded extreme in our claim referenced by the Reviewer. However, we do not think these downstream corrections are guaranteed to work. On the one hand, it requires extensive downstream analysis where it is unclear which methods are more suitable to deal with the introduced artifacts. For example, we have shown that the most popular thresholding approach (cutoff by percent of contribution) performed badly (Figure 4). On the other hand, flat signatures in particular, tend to draw large exposure attributions once artificially introduced in *de novo* signature discovery, because they could overlap with multiple signatures. These large artifactual contributions are especially difficult to correct for in downstream analysis, leading to over-assignments such as what we see in SBS40 from PCAWG results.

Extended Data Figure 3d. An example comparing the performance of NMF and mvNMF on identifying SBS7a. The NMF solution of SBS7a receives a large cosine error. The error spectrum indicates interference from SBS2 coexisting in the dataset. By comparison, mvNMF does not suffer from the SBS2 interference and is able to discover SBS7a accurately.

10. “[...]In fact, we recommended in our tutorial that the signature catalog should be restricted to tumor type-specific signatures during the matching and refitting steps (<https://github.com/parklab/MuSiCal/tree/main/examples>). However, such prior knowledge needs to be established in the first place through an unbiased approach [...]”. If the scope of this paper is to create a more accurate and reliable catalog of signatures per cancer type, then a larger number of samples should be included for each tumor type (as suggested by Reviewer #3). Some of the tumor types in PCAWG are not well represented, and therefore, to create a robust catalog for each tumor type more samples should be added. I also explained above that some of the assignments are likely incorrect, and this could affect the creation definitive catalog per cancer type. Because we have now published mutational signatures studies with >10.000 WGS, the competition is hard. Maybe it would be more useful for the community to have a tool with prior knowledge and weight, including the knowledge derived from these recent large seminal papers.

In the response to Reviewer’s previous comments above, we have elaborated on the importance of an unbiased data-driven approach and provided additional evidence to support MuSiCal’s assignments that the Reviewer was concerned with. The aim of our manuscript was to provide an improved signature analysis tool and demonstrate its effectiveness. We acknowledge that MuSiCal’s results could still contain errors, as no method is perfect, but we have shown extensively that MuSiCal already outperforms existing methods significantly through both simulation studies and ground truth analysis. We have toned down the language on providing a comprehensive reference signature catalog in the manuscript during the first revision.

11. Overall, I do believe that MuSiCal has improvements in its multi-step process compared to SigProfiler, but it is hard to quantify them without a proper and robust "ground truth" analysis. While the math and stats are superior in the simulation setting, the biology and clinical history must be taken into account, otherwise, the impact of this tool might be limited in a field where the bar is very high.

We have responded to the Reviewer’s comments on ground truth analysis and data-driven vs. integrative approaches in the responses above.

Reviewer #2:

Remarks to the Author:

I would like to express our appreciation for the extra steps and changes made to the manuscript, especially regarding the benchmarking routine. The inclusion of a better metric (PRC and auPRC) and more extensive simulations, including noise and a wider range of tools to compare to, has significantly strengthened the paper and claims about superiority of the method. I commend your thoroughness and attention to detail in this regard.

I am pleased to see that all minor and most of the major points have been addressed in the manuscript. However, I do have some additional comments and questions following up on some of the major points I raised previously. I have outlined these comments and questions below (the numbers correspond to those from my initial review) and I look forward to your response.

We thank the Reviewer for the insightful comments during the initial review. We are glad to see that the additions (especially the better metric) suggested by the Reviewer had significantly strengthened our manuscript. We have addressed the Reviewer's further comments below.

1. Major Point 5

I appreciate the efforts you have made in conducting extra noise simulations in your study. It is important to assess the robustness of your results to noise, and I commend your attention to this issue.

Two additional points:

Random noise: While I appreciate that the effect of random noise on the performance on MuSiCal appears to be limited, the introduced noise levels seem to be on the low side. At what noise level does the accuracy / precision-recall substantially fall off? Surely there must be a noise level at which MuSiCal is not able to faithfully reconstruct the underlying signatures anymore. Is there any estimate of the amount of noise that is present in real data and how it would relate to the simulated noise levels?

Spike-in noise: Furthermore, I noticed that the selection of SBS48 signatures in the simulations could potentially be perceived as cherry-picking, although I understand the reasoning behind it. It appears that this selection has a stronger effect than random noise. To provide additional context for the analysis, I would recommend using a few spiked-in signatures, such as a flat one and one with a unique profile.

Additionally, it would have been helpful to include multiple levels of such spike-in noise rather than just 5% in the simulations, similar to the case of random noise. Same as before, I would like to see at which noise level MuSiCal is not able to faithfully reconstruct the underlying signatures.

For random noise simulations, we completely followed the SigProfilerExtractor paper [6] (Islam et al. Cell Genomics 2022). The rationale for the noise level selection in [6] is the following, and we think it is a reasonable argument. Precisions of modern somatic variant calling approaches were above 95% (see, e.g., the PCAWG marker paper [7]), implying that at most 5% of the identified mutations could be attributed to noise. The authors in [6] thus picked 5% as the representative noise level, and considered 10% as a high level of noise. We believe that this range of noise levels covered most datasets we are dealing with nowadays. The Reviewer is right that there is definitely a point where MuSiCal and other

methods will fail at higher noise levels. However, we think that those noise ranges represent unrealistic scenarios, and thus identifying such a failing point might not add much practical value to the manuscript. Also, at the noise levels already benchmarked, we did not see a trend where MuSiCal might fail earlier compared to other tools (Extended Data Figure 8).

For spike-in noise, we did not cherry-pick SBS48. In fact, SBS48 was among the artifact signatures that were more similar to other signatures in the catalog, i.e., those that were more difficult to deal with (**Figure R23**). Of note, we had to exclude artifact signatures already present in the PCAWG samples, which our simulations were based on. Regarding noise levels, similar to the reasoning above on random noise, we think that 5% is a typical level. We did not benchmark more noise levels because each noise level represents a time-consuming analysis of >22,000 samples with multiple tools. In real-world applications, sequencing artifacts (such as those from library preparation or sequencing machine) usually stand out as a distinct signature. For example, we have utilized such artifact signatures in single-cell whole-genome sequencing data for more accurate variant calling [8] (Luquette et al. Nature Genetics 2022). This allows us to get around the scenario we are trying to simulate with spurious spike-in noises. Overall, we believe that the benchmarks presented in our current manuscript were sufficient to demonstrate the robustness of MuSiCal. We therefore refrained from performing more exhaustive benchmarks as they may not add much practical value to the manuscript.

Figure R23. Similarity of artifact signatures to other signatures in the COSMIC catalog. This plot is the same as Extended Data Figure 2c, except that only artifact signatures were shown.

2. Major Point 6

Up to the authors: I think it's worth mentioning that the SD is lower for the MuSiCal results especially since the SD is shown in Fig 2c. This might not be obvious to all readers.

We have added the following clarification in the caption of Figure 2c:

Note that the apparent similar or larger spread of cosine errors for mvNMF is due to the log scale on y-axis. Solutions from mvNMF are in fact more stable, producing smaller standard deviations in the cosine errors overall.

3. Major Point 8 Up to the authors: I think Fig R14 could be included in the main manuscript as an Extended Figure as other readers might also wonder about the effect of the mvNMF on the reconstruction error. Could be included with a single sentence like “While mvNMF increases the uniqueness of solutions, it did not significantly reduce the reconstruction error (Ext Data Fig XX).”

We have added Figure R14 to the manuscript as the new Extended Data Figure 25. We referred to this figure with the following sentence in the Methods section where we described the procedure of selecting the regularization parameter in mvNMF.

This procedure ensures that the regularization does not significantly sacrifice reconstruction accuracy, while still promotes uniqueness of the solutions (Extended Data Fig. 25).

4. Major Point 10

Why have the MuSiCal and SigProfileExtractor *de novo* signatures changed from the first submission? I thought they are still chosen based on the ideal F1 value?

In the first submission, we did not choose the result with the best F1 value. Instead, we used the Decomposition module from SigProfilerExtractor to match the *de novo* signatures identified by both MuSiCal and SigProfilerExtractor to the COSMIC catalog to obtain F1 values, and presented them as they were (see the response to Reviewer’s Comment 10 of the initial review for more details). The matching result by SigProfilerExtractor-Decomposition was shown in the heatmap of the old Figure 3a. During the first revision, we switched to provide the full PRC and used auPRC as the performance metric. In the revised heatmap of Figure 3a, the matching result with the best F1 score was shown as an example.

Therefore, differences in the heatmap of Figure 3a were expected. Of note, the *de novo* signatures themselves were not affected by the matching step. The minor differences in the *de novo* signature spectra compared to the first submission were because of a different replicate (among the 10 replicates per tumor type) being used by chance when remaking the plot.

5. Major Point 11

I appreciate the extra analysis performed by the authors and acknowledge that MuSiCal still outperforms SigProfileExtractor even when forced to the same number of signatures. However, it also becomes clear that the selection of the right number of signatures has a larger effect on the accuracy of the results (0.893 -> 0.914 = 0.024) than the added benefit of using minimum volume NMF and the improved COSMIC matching/refitting routines which increase accuracy from (0.914 -> 0.929 = 0.015), at least in this example. I understand that choosing the right number of signatures is in itself a difficult task and one at which MuSiCal shines. However, I think it would be good to explicitly state the strength of this effect in the main text as otherwise the reader doesn't know which aspect of MuSiCal contributes how much to the improved accuracy.

Indeed, selecting the right number of signatures is one aspect that MuSiCal excels but was not emphasized in our manuscript. We agree with the Reviewer that clarifications need to be made in terms of where the overall improvement of MuSiCal comes from. We have made the following modifications in the main text, from:

MuSiCal outperforms SigProfilerExtractor even when the built-in input normalization of SigProfilerExtractor is turned on or when SigProfilerExtractor is forced to select the same number of signatures as MuSiCal (Extended Data Fig. 7), suggesting that the improved performance of MuSiCal relies on the reduced algorithmic bias powered by mvNMF, as indicated by reduced cosine reconstruction errors of the *de novo* signatures when decomposed using the true signatures (Fig. 3f).

to:

MuSiCal outperforms SigProfilerExtractor even when the built-in input normalization of SigProfilerExtractor is turned on. SigProfilerExtractor performs better when it is forced to select the same number of signatures as MuSiCal, although still underperforms MuSiCal itself (Extended Data Fig. 7). This observation suggests that the improved performance of MuSiCal relies on both reduced algorithmic bias powered by mvNMF as indicated by reduced cosine reconstruction errors of the *de novo* signatures when decomposed using the true signatures (Fig. 3f), and better choice of the number of signatures, which in turn benefits from the uniqueness and stability of mvNMF solutions.

6. On Figure R17

There is a very clear increase in reconstruction error for $r=10, 12, 14$ that really stumped me. Reconstruction error for these values of r is higher than for all other r 's (excluding $r=1$), but this does not seem to affect the "neighboring" odd values 11, 13 and 15 to the same degree. This very much makes me suspicious about potential numerical issues or a bug in the code.

Do the authors have an explanation for this effect? I understand that this behavior does not affect the points raised in Major Point 11 but I do believe that this behavior might be reminiscent of a larger error in the MuSiCal code.

The pattern in even vs. odd values in Figure R17 (copied below for convenience) indeed looked suspicious. However, it was not observed in mvNMF runs on other replicates (in total 10) of the same Skin-Melanoma based synthetic datasets (see **Figure R24** below for an example of another replicate). Therefore, this pattern was simply coincidental and does not indicate any systematic issue.

The increase in reconstruction error itself was also nonintuitive (plot reproduced in **Figure R25** left panel below). But it is because the plotted “reconstruction error” was actually not the “mvNMF reconstruction error” *per se*. We explain this subtle point in more detail below. For each dataset, MuSiCal runs mvNMF (or NMF) multiple times – in this case 20 – on bootstrapped data and then clusters the discovered signatures to obtain consensus signatures (i.e., cluster means). Reconstruction errors with these consensus signatures were plotted in Figure R17 and Figure R25 left panel, following what is often plotted by SigProfilerExtractor. In Figure R25 right panel, we plotted the actual reconstruction errors with raw mvNMF outputs. Now we see the expected monotonic decreasing behavior with respect to number of signatures (r). The large increase in “consensus reconstruction error” in Figure R17 and Figure R25 left panel simply reflects that mvNMF solutions become unstable when r is over-specified. This can be seen from that “consensus reconstruction error” and silhouette score started to become worse from around the same r (Figure R17). When the algorithm becomes unstable, discovered signatures from different runs tend to differ more. As a result, the consensus signatures may deteriorate due to difficulties in clustering, leading to large reconstruction errors. This is part of the reason why we did not rely on Figure R17 type of plots to select r , as is commonly done by other tools. Instead, we investigate the clustering structure itself to obtain more accurate information about solutions from different runs (see Methods and Extended Data Figure 6 for more details).

We have put substantial efforts in making sure that our codes are free of bugs. The third author B. Geiger (added during the first revision) performed an extensive independent review and module test of the code base after our initial submission and found no issues.

Figure R17. Copied from responses to the initial review.

Figure R24. Same as Figure R17, but for another replicate of the simulated Skin-Melanoma dataset.

Figure R25. Left panel: same as reconstruction errors plotted in Figure R17. Consensus signatures were used to calculate the reconstruction errors. Right panel: reconstruction errors calculated with raw mvNMF or NMF outputs. Mean values over 20 mvNMF or NMF runs were plotted for each number of signatures.

7. Major Point 14

I acknowledge the extra work performed by the authors and found the new Ext Data Fig 24 quite informative. Additionally, I think the runtime should be stated in the main text or at least in the Online Methods. Ideally, the authors could include a small section termed “Computational cost of MuSiCal” under section 4.3 as this is where readers are likely going to search for this information.

For Ext Data Fig 24a: Definitely show the y-axis in hours and maybe show the y-axis as a log-scale.

We have changed the y-axis of Extended Data Figure 24a to hours in log scale. To be consistent, the y-axis of panel b is now plotted in log scale as well. We have added the following section in Methods under “4.3 The MuSiCal algorithm”:

4.3.5 Computational cost. For *de novo* signature discovery, MuSiCal (with mvNMF) requires similar but slightly more computational time compared to SigProfilerExtractor (with NMF), and considerably less memory (Extended Data Fig. 24). The computational costs of matching and refitting steps in MuSiCal are negligible compared to *de novo* signature discovery. The *in silico* parameter optimization step requires rerunning *de novo* signature discovery for a grid of thresholds. But during this grid search, there is no need to select the regularization parameter in mvNMF or the number of signatures (as they are both fixed), which are the most time-consuming calculations. Thus, the computational cost for *in silico* optimization is also small compared to *de novo* signature discovery when parallelized properly.

8. Major Point 17

Maybe I misunderstand the analysis done here but both simulations (PCAWG and MuSiCal) are compared to W_{data} and H_{data} gathered from mvNMF, right?. But isn't W_{data}/H_{data} created with mvNMF and therefore directly linked to the simulated MuSiCal data? So MuSiCal uses W_{data}/H_{data} to match and refit W_s/H_s and in turn uses these matrices to simulate W_{simul}/H_{simul} . The difference (simul - data) therefore points towards errors introduced in the matching and refitting stages. On the other hand, PCAWG used a different set of H_{data}/W_{data} that is not available to the authors from which they assigned their W_s/H_s matrices. So for the PCAWG assignments, the difference (simul - data) is a sum of the errors introduced in the PCAWG matching and refitting stages as well as the difference in $data_{mvNMF}$ and $data_{PCAWG}$!

Therefore the fact that the simulated MuSiCal data outperforms the simulated PCAWG data is very much expected from the way this comparison is set up. A more fair comparison would be to use the original W_{data}/H_{data} matrices from the PCAWG analysis (though they are likely not publicly available).

If my assumptions are correct, then this analysis just produces what you put in and is therefore not very informative.

We understand the Reviewer's concern that the *in silico* validation results seem to be biased towards MuSiCal. Both $W_{simul}^{MuSiCal}$ and W_{simul}^{PCAWG} were compared to W_{data} , which was generated by MuSiCal. Therefore the difference between $|W_{simul}^{MuSiCal} - W_{data}|$ and $|W_{simul}^{PCAWG} - W_{data}|$ should be interpreted as the combined effects of differences in both *de novo* signature discovery and matching/refitting between MuSiCal and PCAWG. But that is exactly what we aimed to quantify. More specifically, we aimed to quantify the consistency between the final signature assignment results and the original data, and thus the effects of all intermediate steps that went from raw data to final assignments should be considered. In particular, for PCAWG results, we believe that the NMF non-uniqueness related artifacts during *de novo* signature discovery (as illustrated in Figure 2) contributed to issues in the final signature assignments. With the comparison suggested by the Reviewer using W_{data} from PCAWG – which is not practically doable in the first place as mentioned by the Reviewer – the NMF-related issues will not be captured. Finally, it is not the case that this analysis simply produces what we put in. For example, if we had picked bad thresholds in MuSiCal's matching/refitting steps, the resulted signature assignments would have performed badly during *in silico* validation and easily worse than PCAWG assignments.

9. Major Point 21 After rereading the relevant sections I do believe that the MMRD/MMRP/POLE split is not sufficiently described in the main text given its strong effect on the discovery of new SBS signatures. Currently section 2.5 states: "In automatic cohort stratification, heterogeneous datasets are stratified into subsets". While this is true for individual stratification in Kidney.RCC, Liver.HCC, and Skin.Melanoma it is incorrect and feels disingenuous for the MMRD/MMRP/POLE stratification which in section 4.5.1 is very clearly described as a manual process.

Since this manual split has a strong effect on the outcome and is not performed using the MuSiCal preprocessing module it needs to be included in section 2.5. Something along the lines of "After an initial manual separation of MMRD / MMRP samples, tumor type-specific *de novo* signature discovery is performed ...".

Additionally, I would mention this again when the results are presented: "... in MMRD tumors and therefore benefitting from our manual stratification process, likely representing nonlinear interactions between DNA damage ..."

We have added the following sentence in the first paragraph of section 2.5:

Of note, samples with MMRD and/or polymerase epsilon exonuclease (POLE-exo) domain mutations are isolated with a dedicated procedure beforehand and analyzed separately in a tissue-independent manner (Methods)

We have also modified the sentence when the results were presented:

The 6 new SBS signatures with relatively poor matches are mainly (4 of the 6) discovered in MMRD tumors (potentially benefiting from a separate analysis for them), likely representing nonlinear interactions between DNA damage and repair

10. Small side point: Fig 6b is missing a "*" next to the black bar besides SBS95-100 (like it is present in 6c for ID19-26)

The missing "*" is now added.

11. Major Point 22

I thank the authors for the extensive extra analysis in the case of the newly discovered ID11b signature. Up to the authors: Include half a sentence before the description of the ID11b smoking correlations: "Unlike the original ID11 signature, ID11b correlates with both ..."

This sentence is now modified as "ID11b – unlike COSMIC ID11 – correlates with both ..."

Reviewer #3:

Remarks to the Author:

In the revised manuscript the authors present additional bioinformatics benchmarking of their algorithm MuSiCal for mutational signature extraction. In brief, the algorithm uses a regularised form of NMF termed minimal volume NMF based on trinucleotide representation of single base substitutions or alternatively indels. The additional analyses entail further simulations of base substitution spectra with added Gaussian noise, a spike-in of a confounding signature as well as a comparison to a broader series of tools. These analyses reveal a consistent improvement of MuSiCal over 4 other tools which are used widely in the field.

In this revision, we have further strengthened the conclusion on the improved performance of MuSiCal with additional analyses based on known biological ground truths – one based on platinum-associated SBS31/35 (**Figure R20**) and another based on HRD-associated SBS3 (included in the first revision but presented in a more straightforward manner in this revision; **Figure R19**). MuSiCal outperformed existing methods in both analyses, producing results that were more consistent with ground truth labels of platinum treatment or HRD status. Together with the extensive simulation studies already included in the manuscript, we are confident that MuSiCal represents a consistent improvement over existing methods for signature analysis.

1. The implemented changes addressed my first two major concerns (performance of MuSiCal in relation to other tools). Based on the available data it can be concluded that MuSiCal outperforms other algorithms run out of the box and on simulated data. Yet I do maintain that such performance gains need to be taken with a pinch of salt because (i) the data is simulated according to the author's modelling assumptions and also because (ii) there is often an unintended bias because as an author one knows one's algorithm's ideal tuning parameters, whereas other software is much harder to run. Given that all methods are essentially working on the same NMF core algorithm I'm not convinced that the same performance cannot be achieved by other tools.

During the first revision, we have explained in detail that our simulation was not biased towards our own model, and in fact inherently favored SigProfilerExtractor and other tools based on plain NMF (see response to Reviewer's major comment 2 in the initial review). As to implementation bias, we have had many years of experience working with mutational signatures and using different tools ourselves, in particular SigProfiler/SigProfilerExtractor. We believe that we have the expertise in running these tools properly. We disagree with the Reviewer that MuSiCal is based on the same NMF core algorithm. We would like to draw an analogy to plain linear regression and regularized linear regressions such as LASSO. We do not think that one should consider them as essentially the same algorithm. The advantage of LASSO compared to plain linear regression is also widely accepted and nontrivial. Similarly, MuSiCal is based on mvNMF, which is a regularized version of NMF. The performance gain of mvNMF over NMF is clearly demonstrated by the extensive analysis in our manuscript.

2. I have tested the available code from github though and can assert that one can run the provided examples and also my own analyses. Yet it is worth mentioning that the run time easily exceeds 48h.

We are aware that MuSiCal can require more computational time for larger datasets. In the first revision, we have added Extended Data Figure 24 (copied below) comparing the computational cost of MuSiCal with SigProfilerExtractor. In this revision, we further added the following section in Methods under "4.3 The MuSiCal algorithm" on computational cost:

4.3.5 Computational cost. For *de novo* signature discovery, MuSiCal (with mvNMF) requires similar but slightly more computational time compared to SigProfilerExtractor (with NMF), and considerably less memory (Extended Data Fig. 24). The computational costs of matching and refitting steps in MuSiCal are negligible compared to *de novo* signature discovery. The *in silico* parameter optimization step requires rerunning *de novo* signature discovery for a grid of thresholds. But during this grid search, there is no need to select the regularization parameter in mvNMF or the number of signatures (as they are both fixed), which are the most time-consuming calculations. Thus, the computational cost for *in silico* optimization is also small compared to *de novo* signature discovery when parallelized properly.

Extended Data Figure 24. Computational cost of MuSiCal in comparison to SigProfilerExtractor. MuSiCal (in both mvNMF and NMF modes) and SigProfilerExtractor (based on NMF) were run on each PCAWG tumor type separately for *de novo* signature discovery, and the corresponding computation time and memory usage were shown. MuSiCal was run with 10 CPUs on a high-performance cluster, and SigProfilerExtractor was run with 12 CPUs. **(a)** Computation time (in hours) for *de novo* signature discovery was plotted against the number of samples. Each dot represents a PCAWG tumor type. Solid lines represent linear fits in the log space, i.e., $\log(t) \sim \log(n)$, where t denotes computation time, and n number of samples. MuSiCal with mvNMF and SigProfilerExtractor were comparable in computation time for PCAWG tumor types, although MuSiCal with mvNMF scaled slightly worse ($t \propto n^{0.65}$ for MuSiCal with mvNMF and $t \propto n^{0.57}$ for SigProfilerExtractor). MuSiCal with NMF was much faster and scaled the best ($t \propto n^{0.39}$). **(b)** Same as panel (a) but for memory usage (in GB). MuSiCal with either mvNMF or NMF required considerably less memory than SigProfilerExtractor.

3. Unfortunately, the authors chose to mostly dismiss my other three comments. These were related to the extent to which MuSiCal gives insights into signatures with unknown etiologies, resolves issues with 'flat' signatures and whether the authors provide an improved reference catalogue.

In major comment 3 of the initial review ("give insights into signatures with unknown etiologies"), Reviewer suggested to show evidence for better correlation of chemotherapeutic exposures or DNA repair deficiency conditions. In the first revision, we provided an analysis to show that MuSiCal's SBS3 assignments were more consistent with ground truth HRD status. In this revision, we have presented the same analysis in a more straightforward manner (please see the response to Reviewer 1's Comment 1 above and **Figure R19** therein). In addition, we performed another analysis based on platinum treatment-associated SBS31/35, demonstrating that MuSiCal's SBS31/35 assignments were more consistent with the clinical history of platinum treatment compared to other tools (please see

the response to Reviewer 1's Comment 2 above and **Figure R20** therein). With these additional analyses, we hope that we have fully addressed Reviewer's major comment 3.

For Reviewer's major comment 4 on flat signatures, we have explained in detail the multiple pieces of evidence already included in our manuscript during the first revision. For Reviewer's major comment 5 on the reference catalog, we have also explained that deriving a comprehensive reference catalog from an even larger number of samples is out of scope of the current study, and toned down the language throughout our manuscript. Specifically, we have refrained from using the phrase "reference catalog" and focused on demonstrating the effectiveness of MuSiCal on PCAWG data.

4. My overall verdict is thus that MuSiCal may provide an improvement over other widely-used tools for calculating mutational signatures based on 96-class single base substitutions and when evaluated on simulated data. This could be a useful, albeit conceptually incremental advance. Yet in the absence of proper experimental validation data, which link the data to true mutational processes, it remains questionable whether the reported differences truly provide new insights or whether they just confirm subjective assumptions about how mutational signatures should look like.

So it remains questionable whether there's an actual improvement and the methodological advance appears incremental. MuSiCal doesn't tackle any fundamental questions in the field.

We appreciate the Reviewer's agreement that MuSiCal represents an improvement over other widely used tools in the field. We have provided multiple examples of new insights derived from reanalyzing PCAWG data with MuSiCal (new smoking-related ID11b, more accurate TOP1-related ID4, flat signatures, etc.). In the response to Reviewer 1's Comment 4 and 6 above, we provided two additional examples where MuSiCal's results could point to new biology. We believe that MuSiCal will lead to many more discoveries in the future when applied to broader datasets.

References

1. Degasperi, Andrea, et al. "A practical framework and online tool for mutational signature analyses show intertissue variation and driver dependencies." *Nature cancer* 1.2 (2020): 249-263.
2. Alexandrov, Ludmil B., et al. "The repertoire of mutational signatures in human cancer." *Nature* 578.7793 (2020): 94-101.
3. Degasperi, Andrea, et al. "Substitution mutational signatures in whole-genome–sequenced cancers in the UK population." *Science* 376.6591 (2022): abI9283.
4. Martínez-Jiménez, Francisco, et al. "Pan-cancer whole-genome comparison of primary and metastatic solid tumours." *Nature* (2023): 1-9.
5. Kucab, Jill E., et al. "A compendium of mutational signatures of environmental agents." *Cell* 177.4 (2019): 821-836.
6. Islam, SM Ashiqul, et al. "Uncovering novel mutational signatures by de novo extraction with SigProfilerExtractor." *Cell Genomics* 2.11 (2022): 100179.

7. The ICGC/TCGA Pan-Cancer Analysis of Whole Genomes Consortium. Pan-cancer analysis of whole genomes. *Nature* 578, 82–93 (2020).
8. Luquette, Lovelace J., et al. "Single-cell genome sequencing of human neurons identifies somatic point mutation and indel enrichment in regulatory elements." *Nature Genetics* 54.10 (2022): 1564-1571.

Decision Letter, second revision:

27th Jul 2023

Dear Peter,

Your Technical Report, "Accurate and sensitive mutational signature analysis with MuSiCal" has now been seen by the original Referees #1 and #2.

You will see from their comments below that while there is support for publication, there are still some important points remaining. We continue to be interested in the possibility of publishing your study in *Nature Genetics*, but would like to consider your response to these concerns in the form of a revised manuscript before we make a final decision on publication.

Briefly - and potentially entirely predictably! - the two reports present opinions in agreement with their past reviews.

Reviewer #1 appreciates the efforts made in revision, but remains unconvinced that MuSiCal offers a sufficient broad-appeal advance.

Referee #2, conversely, is completely supportive, and provides rebuttals to some of the criticisms from #1 and #3.

While we think Reviewer #1's points on the overall advance are not invalid, our editorial assessment is that the opinion of the field is likely closer to Reviewer #2's position - i.e., that MuSiCal offers some points of distinction to e.g. SigProfilerExtractor that are indeed of value to those performing mutational signature analysis. Given, as noted by Referee #2, having a variety of tools is generally a boon for analysis, we are prepared to overrule Reviewer #1's remaining concerns on the overall advance; however, we would nonetheless direct you to respond to all the remaining reviewer comments, as well as moderating your claims in light of Reviewer #1's specific points.

To guide the scope of the revisions, the editors discuss the referee reports in detail within the team, including with the chief editor, with a view to identifying key priorities that should be addressed in revision and sometimes overruling referee requests that are deemed beyond the scope of the current study. We hope that you will find the prioritized set of referee points to be useful when revising your study. Please do not hesitate to get in touch if you would like to discuss these issues further.

We therefore invite you to revise your manuscript taking into account all reviewer and editor comments. Please highlight all changes in the manuscript text file. At this stage we will need you to

upload a copy of the manuscript in MS Word .docx or similar editable format.

*2) If you have not done so already please begin to revise your manuscript so that it conforms to our Technical Report format instructions, available [here](http://www.nature.com/ng/authors/article_types/index.html). Refer also to any guidelines provided in this letter.

[redacted]

We hope to receive your revised manuscript within four to eight weeks. If you cannot send it within this time, please let us know.

Sincerely,

Michael Fletcher, PhD
Senior Editor, Nature Genetics

ORCID: 0000-0003-1589-7087

Reviewers' Comments:

Reviewer #1:

Remarks to the Author:

Jin et al.'s rebuttal and updated paper showcases commendable efforts in addressing my questions/criticisms by providing additional data and conducting further analysis to support their claims.

However, certain points raised in my two revisions were not fully addressed. Additionally, there remain disagreements regarding some of the conclusions and concepts.

Here are my major points:

I respect the Authors' response about the nature of SBS9. Nevertheless, it is important to consider the available data from multiple papers that indicate the absence of SBS9 in myeloid tumors. Based on this, upon checking the GEL database it becomes evident that these SBS9 positive myeloid tumors exhibit a significantly higher mutational burden (5-10 times higher) compared to what has been traditionally reported in myeloid tumors. Furthermore, the presence of APOBEC, which is typically absent in myeloid tumors, is also observed in these cases. Interestingly, among the most frequent hematological tumors, there are only two known to have both SBS9 and APOBEC: multiple myeloma and aggressive lymphomas. Upon reviewing the supplementary table 5 of the original paper (Degasperi et al., Science 2022), it becomes apparent that the multiple myeloma tumors were mistakenly included within the myeloid tumor group, despite their lymphoid nature. This misclassification led to the identification of SBS9 within the myeloid group. This serves as a compelling example highlighting the importance of integrating de novo extraction techniques with clinical and biological knowledge specific to each tumor type.

I am pleased to note that the Authors have responded to my comments regarding HRD/BRACness by demonstrating improved performance in the recognition of SBS3 in HRD+ cases. It is worth emphasizing that SBS3 is an integral component of HRD and should not be utilized as an isolated biomarker for HRD assessment. In light of this, I previously encouraged the authors to conduct HRDetect using SBS3 extracted via both Sigorofiler and Musical methodologies. This additional analysis would shed light on whether the results of HRDetect are influenced by the choice of SBS3 extraction method. While it is undoubtedly important to enhance the accuracy in defining SBS3 signature, it is equally critical to evaluate the potential clinical and translational impact of such improvements.

I appreciate the Authors' comment regarding the potential similarity between SBS31 and SBS137. However, it appears that claiming a direct association between these signatures may be overstretched. As a matter of fact, Degasperis et al. Science 2022 have pointed out that the specific signature SBS137 remains unknown in its etiology and exhibits a closer resemblance to UV light exposure rather than platinum exposure. This finding raises doubts about the direct attribution of SBS137 to platinum-induced mutagenesis.

Regarding SBS84, I acknowledge the presence of one colon case (281278) reported in the GEL dataset with this signature. The original study by Kasar et al. from the Getz lab (Nat Comm 2016) has demonstrated that SBS84 operates in clusters, suggesting that its activity should ideally be tested on kataegis or somatic hypermutation loci, and not genome-wide. I am not fully sure about the specific methodology used to extract SBS84 in GEL dataset, but claiming a different etiology for this well-defined signature requires more substantial evidence. It is also important to note that AID/SBS84 signatures are usually observed in conjunction with SBS9, which serves as evidence of post-germinal center involvement. Consequently, it is challenging to find tumors with one signature and not the other.

We appreciate the Reviewer's perspective regarding clock-like signatures in myeloid tumors. While Mitchell et al. (Nature 2022) and Williams et al. (Nature 2022) may have focused on small cohorts of patients or normal cells, it is indeed well-established from other studies, including large-scale projects like TCGA or the BEAT-AML, that myeloid tumors predominantly exhibit clock-like mutational, with few notable exceptions. For example, Griffin et al. (Nature 2023) identified dendritic cell leukemia as an exception to the clock-like signatures observed in myeloid tumors, however the UV light signature reported is in line with the skin localization of that tumor and its biology.

The sentence provided by the Authors in their rebuttal letter states that "another unknown mutational process could produce a signature similar to SBS31, resulting in SBS31 being assigned to some samples without platinum treatment". While I don't fundamentally disagree with this notion, I believe it is a significant claim that can have significant implications for data analysis and interpretation. To truly be convinced of its validity, particularly for well-established signatures such as SBS31, SBS84, or SBS9, a more comprehensive and robust validation is necessary. Presenting a few cases with these signatures does not provide sufficiently convincing evidence.

Reviewer #2:

Remarks to the Author:

I would like to thank the authors for the extra analyses and clarifications that were included in the second revisions step. The changes and clarifications were to my fullest satisfaction and I do not have any additional points to raise. I believe MuSiCal and this article is a valuable addition to the cancer genomics field.

I do however acknowledge that Reviewers 1 and 3 are less favorable towards the manuscript and I would like to comment on some of their concerns below.

Reviewer 1 firstly criticized the lack of ground truth data to test the algorithm, which the authors addressed by analyzing platinum-associated signatures SBS31/35 and HRD-associated SBS3. While

these signatures may not be a perfect ground truth, they are well rooted in biology and their use as a reference is well justified. Due to the general difficulty of finding ground truth data for mutational signature analysis, I believe that the combination of these additional analyses as well as the simulation studies are sufficient to show a general improvement over existing tools and I suggest adding these two new analyses to the paper as Extended Data Figures.

The second and seemingly larger concern of Reviewer 1 was centered around individual signatures that were detected in some samples but that do not conform with our current understanding of those signatures. While I somewhat share the Reviewer's concern, I also agree with the authors' statement that "any inconsistency between the results and prior biological knowledge could represent an error on the one hand – which does not undermine MuSiCal's improvement over existing methods (Figure R19 and R20) – or suggest new biology on the other hand (Figure R21 and R22)". The authors support their latter point with other studies that have independently found the same signatures (e.g. SBS137 in Degasperi et al. Science 2022). I would assume that a similar analysis with widely used tools such as SigProfilerExtractor, signature.tools.lib, SignatureAnalyzer or others would produce a similar range of equally contradictory signatures.

Reviewer 1 also states: "Maybe it would be more useful for the community to have a tool with prior knowledge and weight, including the knowledge derived from these recent large seminal papers". While I do agree that a tool that incorporates both data-driven signature discovery as well as prior biological / clinical knowledge would be valuable, such an approach is conceptually quite different from the algorithm presented here and I would consider it out of scope of the presented manuscript.

Reviewer 3's main concerns are that, although Musical outperforms the other tools, the benchmarking on simulations could be biased 1) because the simulations were set up according to the authors "modeling assumptions" and 2) because of the "unintended bias" based on the author knowing their algorithm best. While these concerns have some merit, they are so general that they also apply to most other method papers. By using a variety of simulations and ground truth analyses, I do believe that the authors have done their best to show the improvements of mvNMF and Musical for a range of tasks.

Reviewer 3 further criticizes the absence of experimental data with "true" signatures to validate the algorithm. While again this is indeed a valid criticism I do not believe it is one that can be addressed by the authors simply because such data does not exist or only exists for isolated instances or signatures with very clear etiologies (UV, certain chemicals). Taking this criticism at heart would prevent publication of a vast majority of novel and potentially interesting algorithms.

Finally, Reviewer 3 questions the novelty of the approach. While using mvNMF might not be a groundbreaking novelty, I do believe that it contains sufficient novelty to warrant its publication and that it provides sufficient methodological advancement to move the field forward.

Both Reviewer 1 and 3 acknowledge the improvements of Musical over existing tools (Reviewer 1: "the Authors did a good job of showing the statistical advantages that MuSiCal [sic] compared to SigProfiler" - Reviewer 3: "These analyses reveal a consistent improvement of MuSiCal over 4 other tools which are used widely in the field."). I believe that the authors have gone to sufficient lengths to demonstrate that their algorithm can outperform competing methods. I generally believe that the community benefits from the availability of a multitude of tools for mutational signature analysis and I

believe that Musical offers enough improvements in features and accuracy to make it a good addition to the repertoire of signature calling methods and very interesting to the readers of Nature Genetics.

Author Rebuttal, second revision:

Detailed responses to reviewers

Reviewer #1:

Remarks to the Author:

Jin et al.'s rebuttal and updated paper showcases commendable efforts in addressing my questions/criticisms by providing additional data and conducting further analysis to support their claims. However, certain points raised in my two revisions were not fully addressed. Additionally, there remain disagreements regarding some of the conclusions and concepts.

We are grateful to the Reviewer for another round of detailed and thoughtful comments! In this revision, we have performed a more detailed HRDetect analysis as requested, using SBS3 assignments calculated by different methods. As detailed in the following pages, HRDetect with MuSiCal's SBS3 assignment achieved a lower false positive rate at the same sensitivity compared to the original HRDetect.

Regarding the comments on specific signatures, we understand the Reviewer's concerns. We believe that we have provided some orthogonal evidence from public data to support our results and that, to completely settle these issues, a significant amount of experimental investigations are needed. In light of the Reviewer's concerns, we have now included additional comments and Extended Data Figures to note the caveats and uncertainties in the interpretation of some signature assignment results.

Other changes to the manuscript include the addition of three Extended Data Figures (two at the request of Reviewer 2 and one on SBS9-positive samples) as well as textual changes to Discussion related to those figures. We also removed a sentence related to the clinical implications of MuSiCal.

Here are my major points:

1. I respect the Authors' response about the nature of SBS9. Nevertheless, it is important to consider the available data from multiple papers that indicate the absence of SBS9 in myeloid tumors. Based on this, upon checking the GEL database it becomes evident that these SBS9 positive myeloid tumors exhibit a significantly higher mutational burden (5-10 times higher) compared to what has been traditionally reported in myeloid tumors. Furthermore, the presence of APOBEC, which is typically absent in myeloid tumors, is also observed in these cases. Interestingly, among the most frequent hematological tumors, there are only two known to have both SBS9 and APOBEC: multiple myeloma and aggressive lymphomas. Upon reviewing the supplementary table 5 of the original paper (Degasperi et al., *Science* 2022), it becomes apparent that the multiple myeloma tumors were mistakenly included within the myeloid tumor group, despite their lymphoid nature. This misclassification led to the identification of SBS9 within the myeloid group. This serves as a compelling example highlighting the importance of integrating de novo extraction techniques with clinical and biological knowledge specific to each tumor type.

We thank the Reviewer for pointing out the misclassification of multiple myelomas (MMs) as myeloid tumors in the GEL signature paper (Degasperi et al., *Science* 2022). We agree that it appears some samples have been misclassified. Of the 91 samples in the myeloid group, 6 have both SBS9 and APOBEC. However, there are additional 23 samples that are SBS9-positive but without APOBEC. For these 23 samples, it is not clear to us whether they should be classified as MM or non-MM, using the

prior knowledge the Reviewer cites. Perhaps it is not unreasonable to think that some non-MM myeloid samples could in fact have some SBS9. If so, our initial result that 2 out of 56 Myeloid-MPN samples we analyzed have SBS9 are not necessarily mistakes by MuSiCal. Since detailed tumor type annotation is missing in the GEL paper, we are not able to resolve this question, but it does seem clear that the classification based on signatures may not be a simple binary one.

As a specific example, we present evidence that the GEL myeloid sample GEL-2186446-11 has a nonzero exposure of SBS9 but still is highly likely to be of myeloid origin rather than a misclassified MM as the Reviewer suggested. First of all, this sample has a low mutation burden of 2186, similar to SBS9-negative GEL myeloid tumors (mean mutation burden = 1836 vs mean mutation burden of 6889 for the SBS9- positive samples). Second, it does not show any appearance of APOBEC signatures either as reported in the exposure table by Degasperi et al. or visually in the mutational spectrum (see bottom of **Figure R26**). Its mutational spectrum has the characteristic peaks of SBS9 at T>A and T>G (Figure R26), which is probably what led to a relative SBS9 exposure of 0.18 by Degasperi et al. Note that the mutational spectrum of this sample is also highly similar to the two SBS9-positive Myeloid-MPN samples we analyzed using MuSiCal (cosine similarity > 0.96, bottom three rows of Figure R26). Our two samples were annotated as myeloid and did not have any APOBEC components. We think these observations provide strong support that SBS9-positivity

does not necessarily imply that the sample is MM, and also lend credibility to the MuSiCal's SBS9 assignment for these two samples.

We are aware that all three samples mentioned here showed a stronger enrichment of T[T>G]A mutations compared to COSMIC SBS9, as pointed out by the Reviewer previously. However, we did notice that the SBS9 extracted by the GEL signature paper was more enriched at the same T[T>G]A mutation channel (compare the top two rows of Figure R26), suggesting that SBS9 might have a larger fraction of T[T>G]A mutations than previously appreciated. We think that the issue boils down to whether one believes that the T>A and T>G mutations in these three samples come from SBS9. Although one could have removed SBS9 assignment entirely from myeloid samples based on prior biological knowledge, the T>A and T>G mutations in these samples would be left unexplained by doing so. We thus believe it is reasonable to take an unbiased data-driven approach and still assign SBS9 to these samples, as was done by both MuSiCal and Degasperi et al.

Nevertheless, we modified the Discussion to note that the SBS9 assignment in these myeloid samples needs to be interpreted cautiously and now show these myeloid samples in Extended Data Figure 27 (copied below).

Figure R26. Spectra of COSMIC SBS9, SBS9 from the GEL signature paper (Degasperri et al., *Science* 2022), two PCAWG Myeloid-MPN samples (SP116883 and SP116885) with nonzero SBS9 exposures according to MuSiCal, and an example of a GEL myeloid sample (GEL-2186446-11) with nonzero SBS9 exposure according to Degasperri et al. MuSiCal assigned SBS1, SBS5, and SBS9 to SP116883 (mutation burden = 2002) with relative exposures of 0.17, 0.60, and 0.23, respectively. MuSiCal assigned SBS1, SBS5, and SBS9 to SP116885 (mutation burden = 2260) with relative exposures of 0.18, 0.57, and 0.24, respectively. Degasperri et al. assigned SBS1, SBS5, SBS9, and SBS124 to GEL-2186446-11 (mutation burden = 2186) with relative exposures of 0.39, 0.27, 0.18, and 0.16, respectively. Cosine similarities among these three samples were all > 0.96.

Extended Data Figure 27. Assignment of SBS9 in two Myeloid-MPN samples. MuSiCal assigned SBS9 to 2 of the 56 Myeloid-MPN samples from PCAWG. The spectra of these two samples (SP116883 and SP116885) were shown together with SBS9. Both samples demonstrated characteristic peaks of SBS9 at T>A and T>G channels. However, SBS9 is known to be associated with polymerase eta activity during somatic hypermutation in lymphoid cells and is usually observed in post-germinal center lymphoid tumors. The assignment of SBS9 in these two myeloid tumors may thus represent a distinct mutational process that produces a similar signature in the trinucleotide context. Indeed, compared to SBS9, the T>G mutations in these two samples were specifically enriched at T[T>G]A.

2. I am pleased to note that the Authors have responded to my comments regarding HRD/BRCAness by demonstrating improved performance in the recognition of SBS3 in HRD+ cases. It is worth emphasizing that SBS3 is an integral component of HRD and should not be utilized as an isolated biomarker for HRD assessment. In light of this, I previously encouraged the authors to conduct HRDetect using SBS3 extracted via both SigProfiler and Musical methodologies. This additional analysis would shed light on whether the results of HRDetect are influenced by the choice of SBS3 extraction method. While it is undoubtedly important to enhance the accuracy in defining SBS3 signature, it is equally critical to evaluate the potential clinical and translational impact of such improvements.

In our last revision, we had demonstrated the improved performance of MuSiCal using ground truth data of HRD-associated SBS3 and platinum-associated SBS31/35. We did not perform a more detailed HRDetect analysis, as we believed that HRD detection represents a separate problem that is only marginally relevant to our manuscript. In this revision, we have now performed this analysis in response to the Reviewer's request.

We closely followed the training and evaluation procedure used in Figure 4a of the HRDetect paper (Davies et al., *Nature Medicine* 2017). Specifically, we ran both MuSiCal and SigProfilerExtractor on the 560 breast cancer dataset to obtain SBS3 assignments. We replaced

the SBS3 exposures used in HRDetect by the MuSiCal- and SigProfilerExtractor-derived results separately (while keeping the other variables in the model). We then retrained HRDetect and compared its performance with the original HRDetect. The results were summarized in **Figure R27**. Although HRDetect with SBS3 assignments calculated from different methods had a similar AUC of 0.984, with MuSiCal-derived SBS3 assignments, the model achieved the lowest false positive rate (FPR) at higher sensitivity ranges that are more

clinically relevant. For example, HRDetect with MuSiCal-derived SBS3 assignments achieved FPR of 4.1% at 100% sensitivity, compared to the FPR of 8.5% for the original HRDetect at the same sensitivity.

We also compared the performance of models with only SBS3 (without the other variables) calculated by different methods. Importantly, MuSiCal SBS3 achieved an AUC of 0.958, better than SBS3 exposures used in the original HRDetect (AUC = 0.939) and SigProfilerExtractor SBS3 (AUC=0.877). This performance is even better than using HRDetect's most important feature del.mh (deletions with microhomology) alone (AUC = 0.946). This result suggests that SBS3 alone – when assigned accurately – can already achieve good performance in HRD detection. This could have important clinical implications. In particular, although better performance is achieved when integrating SBS3 with other variables, features such as structural variation signatures and deletions with microhomology can only be obtained from whole-genome sequencing (WGS) data. However, WGS is still not routinely used in the clinics currently, which is an important limitation of HRDetect's applications. By comparison, SBS3 can be accurately estimated from whole-exome or even panel sequencing data (Gulhan et al., *Nature Genetics* 2019), which are more widely accessible in the clinics. Our results thus support the use of SBS3 alone for HRD detection, which could benefit many more patients and clinics without access to WGS.

Nevertheless, we agree with the Reviewer that we should be careful not to overstate the clinical impact of improved SBS3 assignment. We have therefore removed the following sentence in the main text:

~~In fact, difficulties in signature assignment may have undermined the performance of previous signature-based classifiers of patient subgroups and limited their clinical applications [29, 30].~~

Figure R27. Performance of different variations of HRDetect and models based on individual mutational signatures. See texts above for more details.

3. I appreciate the Authors' comment regarding the potential similarity between SBS31 and SBS137. However, it appears that claiming a direct association between these signatures may be overstretched. As a matter of fact, Degasperri et al. *Science* 2022 have pointed out that the specific signature SBS137 remains unknown in its etiology and exhibits a closer resemblance to UV light exposure rather than platinum exposure. This finding raises doubts about the direct attribution of SBS137 to platinum-induced mutagenesis.

We point out that Degasperri et al. *Science* 2022 did not claim a closer resemblance of SBS137 to the UV signature than to the platinum signature. In fact, Degasperri et al. demonstrated that 1) tumors with SBS137 do not have the CC>TT DBS signature characteristic of UV light exposure; 2) SBS137 has transcriptional strand bias (TSB) in the opposite direction of that of the UV signature. Specifically, SBS137 has an excess of C>T mutations on the transcribed strand, which is exactly the same as the platinum signature SBS31. By comparison, UV signatures SBS7a/b have an excess of C>T mutations on the untranscribed strand. These observations together point to the resemblance of SBS137 to SBS31 rather than to SBS7a/b.

4. Regarding SBS84, I acknowledge the presence of one colon case (281278) reported in the GEL dataset with this signature. The original study by Kasar et al. from the Getz lab (Nat Comm 2016) has demonstrated that SBS84 operates in clusters, suggesting that its activity should ideally be tested on kataegis or somatic hypermutation loci, and not genome-wide. I am not fully sure about the specific methodology used to extract SBS84 in GEL dataset, but claiming a different etiology for this well-defined signature requires more substantial evidence. It is also important to note that AID/SBS84 signatures are usually observed in conjunction with SBS9, which serves as evidence of post-germinal center involvement. Consequently, it is challenging to find tumors with one signature and not the other.

Similar to the Reviewer's previous comments on SBS9 and SBS31, we believe that this is again a case that requires significant future work to settle but is minor to the overall presentation of our manuscript. We have provided visual and orthogonal evidence supporting our SBS84 assignment in these small number of samples that were suspicious to the Reviewer and noted that they could otherwise be errors, as no algorithm is perfect. Either way, we do not think it undermines MuSiCal's overall improvement over existing methods. To note the caveats and uncertainties in the interpretation of some signature assignment results, we have added a comment in the Discussion section in the main text (see response to Reviewer's Comment 1 above).

5. We appreciate the Authors' perspective regarding clock-like signatures in myeloid tumors. While Mitchell et al. (Nature 2022) and Williams et al. (Nature 2022) may have focused on small cohorts of patients or normal cells, it is indeed well-established from other studies, including large-scale projects like TCGA or the BEAT-AML, that myeloid tumors predominantly exhibit clock-like mutational, with few notable exceptions. For example, Griffin et al. (Nature 2023) identified dendritic cell leukemia as an exception to the clock-like signatures observed in myeloid tumors, however the UV light signature reported is in line with the skin localization of that tumor and its biology.

Please see our response to Comment 1.

The sentence provided by the Authors in their rebuttal letter states that "another unknown mutational process could produce a signature similar to SBS31, resulting in SBS31 being assigned to some samples without platinum treatment". While I don't fundamentally disagree with this notion, I believe it is a significant claim that can have significant implications for data analysis and interpretation. To truly be convinced of its validity, particularly for well-established signatures such as SBS31, SBS84, or SBS9, a more comprehensive and robust validation is necessary. Presenting a few cases with these signatures does not provide sufficiently convincing evidence.

To settle these issues, experimental validations will be needed. However, we think those are out of scope of for this manuscript. In light of the Reviewer's concerns, we have modified the main text to note the uncertainties in the interpretation of some signature assignment results (see Comment 1 response).

Reviewer #2:

Remarks to the Author:

I would like to thank the authors for the extra analyses and clarifications that were included in the second revisions step. The changes and clarifications were to my fullest satisfaction and I do not have any additional points to raise. I believe MuSiCal and this article is a valuable addition to the cancer genomics field.

We thank the Reviewer for the positive feedback. We also thank the Reviewer for the insightful comments during the previous two rounds of reviews that had significantly strengthened our manuscript.

I do however acknowledge that Reviewers 1 and 3 are less favorable towards the manuscript and I would like to comment on some of their concerns below.

Reviewer 1 firstly criticized the lack of ground truth data to test the algorithm, which the authors addressed by analyzing platinum-associated signatures SBS31/35 and HRD-associated SBS3. While these signatures may not be a perfect ground truth, they are well rooted in biology and their use as a reference is well justified. Due to the general difficulty of finding ground truth data for mutational signature analysis, I believe that the combination of these additional analyses as well as the simulation studies are sufficient to show a general improvement over existing tools and I suggest adding these two new analyses to the paper as Extended Data Figures.

We thank the Reviewer for the clarifications regarding the ground truth analysis. We have added Figures R19 and R20 on HRD-associated SBS3 and platinum-associated SBS31/35 as Extended Data Figures 24 and 25, respectively.

The second and seemingly larger concern of Reviewer 1 was centered around individual signatures that were detected in some samples but that do not conform with our current understanding of those signatures. While I somewhat share the Reviewer's concern, I also agree with the authors' statement that "any inconsistency between the results and prior biological knowledge could represent an error on the one hand – which does not undermine MuSiCal's improvement over existing methods (Figure R19 and R20) – or suggest new biology on the other hand (Figure R21 and R22)". The authors support their latter point with other studies that have independently found the same signatures (e.g. SBS137 in Degasperi et al. Science 2022). I would assume that a similar analysis with widely used tools such as SigProfilerExtractor, signature.tools.lib, SignatureAnalyzer or others would produce a similar range of equally contradictory signatures. Reviewer 1 also states: "Maybe it would be more useful for the community to have a tool with prior knowledge and weight, including the knowledge derived from these recent large seminal papers". While I do agree that a tool that incorporates both data-driven signature discovery as well as prior biological / clinical knowledge would be valuable, such an approach is conceptually quite different from the algorithm presented here and I would consider it out of scope of the presented manuscript. Reviewer 3's main concerns are that, although Musical outperforms the other tools, the benchmarking on simulations could be biased 1) because the simulations were set up according to the authors "modeling assumptions" and 2) because of the "unintended bias" based on the author knowing their algorithm best. While these concerns have some merit, they are so general that they also apply to most other

method papers. By using a variety of simulations and ground truth analyses, I do believe that the authors have done their best to show the improvements of mvNMF and Musical for a range of tasks. Reviewer 3 further criticizes the absence of experimental data with “true” signatures to validate the algorithm. While again this is indeed a valid criticism I do not believe it is one that can be addressed by the authors simply because such data does not exist or only exists for isolated instances or signatures with very clear etiologies (UV, certain chemicals). Taking this criticism at heart would prevent publication of a vast majority of novel and potentially interesting algorithms. Finally, Reviewer 3 questions the novelty of the approach. While using mvNMF might not be a groundbreaking novelty, I do believe that it contains sufficient novelty to warrant its publication and that it provides sufficient methodological advancement to move the field forward. Both Reviewer 1 and 3 acknowledge the improvements of Musical over existing tools (Reviewer 1: “the Authors did a good job of showing the statistical advantages that MuSiCal [sic] compared to SigProfiler” - Reviewer 3: “These analyses reveal a consistent improvement of MuSiCal over 4 other tools which are used widely in the field.”). I believe that the authors have gone to sufficient lengths to demonstrate that their algorithm can outperform competing methods. I generally believe that the community benefits from the availability of a multitude of tools for mutational signature analysis and I believe that Musical offers enough improvements in features and accuracy to make it a good addition to the repertoire of signature calling methods and very interesting to the readers of Nature Genetics.

We thank the Reviewer for the comments regarding the other two Reviewers’ criticisms.

Decision Letter, third revision:

4th Oct 2023

Dear Peter,

Thank you for submitting your revised manuscript "Accurate and sensitive mutational signature analysis with MuSiCal" (NG-TR59859R4). We've made an editorial check of your changes and we think they are satisfactory to the point of not requiring further review, and therefore we'll be happy in principle to publish it in Nature Genetics, pending minor revisions to satisfy the referees' final requests and to comply with our editorial and formatting guidelines.

****As the current version of your manuscript is in a PDF format, please email us a copy of the file in an editable format (Microsoft Word or LaTeX)-- we cannot proceed with PDFs at this stage.****

Sincerely,

Michael Fletcher, PhD
Senior Editor, Nature Genetics

ORCID: 0000-0003-1589-7087

Final Decision Letter:

8th Jan 2024

Dear Peter,

I am delighted to say that your manuscript "Accurate and sensitive mutational signature analysis with MuSiCal" has been accepted for publication in an upcoming issue of Nature Genetics.

Your paper will be published online after we receive your corrections and will appear in print in the next available issue. You can find out your date of online publication by contacting the Nature Press Office (press@nature.com) after sending your e-proof corrections.

Please note that *Nature Genetics* is a Transformative Journal (TJ). Authors may publish their research with us through the traditional subscription access route or make their paper immediately open access through payment of an article-processing charge (APC). Authors will not be required to make a final decision about access to their article until it has been accepted. [Find out more about Transformative Journals](https://www.springernature.com/gp/open-research/transformative-journals)

Authors may need to take specific actions to achieve [compliance](https://www.springernature.com/gp/open-research/funding/policy-compliance-faqs) with funder and institutional open access mandates. If your research is supported by a funder that requires immediate open access (e.g. according to [Plan S principles](https://www.springernature.com/gp/open-research/plan-s-compliance)) then you should select the gold OA route, and we will direct you to the compliant route where possible. For authors selecting the subscription publication route, the journal's standard licensing terms will need to be accepted, including [self-archiving-and-license-to-publish](https://www.nature.com/nature-portfolio/editorial-policies/self-archiving-and-license-to-publish). Those licensing terms will supersede any other terms that the author or any third party may assert apply to any version of the manuscript.

If you have not already done so, we invite you to upload the step-by-step protocols used in this

manuscript to the Protocols Exchange, part of our on-line web resource, natureprotocols.com. If you complete the upload by the time you receive your manuscript proofs, we can insert links in your article that lead directly to the protocol details. Your protocol will be made freely available upon publication of your paper. By participating in natureprotocols.com, you are enabling researchers to more readily reproduce or adapt the methodology you use. Natureprotocols.com is fully searchable, providing your protocols and paper with increased utility and visibility. Please submit your protocol to <https://protocolexchange.researchsquare.com/>. After entering your nature.com username and password you will need to enter your manuscript number (NG-TR59859R5). Further information can be found at <https://www.nature.com/nature-portfolio/editorial-policies/reporting-standards#protocols>

Sincerely,

Michael Fletcher, PhD
Senior Editor, Nature Genetics

ORCID: 0000-0003-1589-7087